# Learning Sparse and Low-Rank Priors for Image Recovery via Iterative Reweighted Least Squares Minimization

**Stamatios Lefkimmiatis, Iaroslav Koshelev**
Huawei Noah's Ark Lab
{stamatios.lefkimmiatis, koshelev.iaroslav}@huawei.com

## ABSTRACT

We introduce a novel optimization algorithm for image recovery under learned sparse and low-rank constraints, which we parameterize as weighted extensions of the $\ell_p^p$-vector and $\mathcal{S}_p^p$ Schatten-matrix quasi-norms for $0 < p \le 1$, respectively. Our proposed algorithm generalizes the Iteratively Reweighted Least Squares (IRLS) method, used for signal recovery under $\ell_1$ and nuclear-norm constrained minimization. Further, we interpret our overall minimization approach as a recurrent network that we then employ to deal with inverse low-level computer vision problems. Thanks to the convergence guarantees that our IRLS strategy offers, we are able to train the derived reconstruction networks using a memory-efficient implicit back-propagation scheme, which does not pose any restrictions on their effective depth. To assess our networks' performance, we compare them against other existing reconstruction methods on several inverse problems, namely image deblurring, super-resolution, demosaicking and sparse recovery. Our reconstruction results are shown to be very competitive and in many cases outperform those of existing unrolled networks, whose number of parameters is orders of magnitude higher than that of our learned models. The code is available at this link.

## 1 INTRODUCTION

With the advent of modern imaging techniques, we are witnessing a significant rise of interest in inverse problems, which appear increasingly in a host of applications ranging from microscopy and medical imaging to digital photography, 2D&3D computer vision, and astronomy (Bertero & Boccacci, 1998). An inverse imaging problem amounts to estimating a latent image from a set of possibly incomplete and distorted indirect measurements. In practice, such problems are typical ill-posed (Tikhonov, 1963; Vogel, 2002), which implies that the equations relating the underlying image with the measurements (image formation model) are not adequate by themselves to uniquely characterize the solution. Therefore, in order to recover approximate solutions, which are meaningful in a statistical or physical sense, from the set of solutions that are consistent with the image formation model, it is imperative to exploit prior knowledge about certain properties of the underlying image.

Among the key approaches for solving ill-posed inverse problems are variational methods (Benning & Burger, 2018), which entail the minimization of an objective function. A crucial part of such an objective function is the *regularization* term, whose role is to promote those solutions that fit best our prior knowledge about the latent image. Variational methods have also direct links to Bayesian methods and can be interpreted as seeking the penalized maximum likelihood or the maximum *a posteriori* (MAP) estimator (Figueiredo et al., 2007), with the regularizer matching the negative log-prior. Due to the great impact of the regularizer in the reconsturction quality, significant research effort has been put in the design of suitable priors. Among the overwhelming number of existing priors in the literature, sparsity and low-rank (spectral-domain sparsity) promoting priors have received considerable attention. This is mainly due to their solid mathematical foundation and the competitive results they achieve (Bruckstein et al., 2009; Mairal et al., 2014).

Nowdays, thanks to the advancements of deep learning there is a plethora of networks dedicated to image reconstruction problems, which significantly outperform conventional approaches. Nevertheless, they are mostly specialized and applicable to a single task. Further, they are difficult to analyze and interpret since they do not explicitly model any of the well-studied image properties,

successfully utilized in the past (Monga et al., 2021). In this work, we aim to harness the power of supervised learning but at the same time rely on the rich body of modeling and algorithmic ideas that have been developed in the past for dealing with inverse problems. To this end our contributions are: (**1**) We introduce a generalization of the Iterative Reweighted Least Squares (IRLS) method based on novel tight upper-bounds that we derive. (**2**) We design a recurrent network architecture that explicitly models sparsity-promoting image priors and is applicable to a wide range of reconstruction problems. (**3**) We propose a memory efficient training strategy based on implicit back-propagation that does not restrict in any way our network's effective depth.

## 2 IMAGE RECONSTRUCTION

Let us first focus on how one typically deals with inverse imaging problems of the form:

$$y = Ax + n, \tag{1}$$

where $x \in \mathbb{R}^{n \cdot c}$ is the multichannel latent image of $c$ channels, that we seek to recover, $A : \mathbb{R}^{n \cdot c} \to \mathbb{R}^{m \cdot c'}$ is a linear operator that models the impulse response of the sensing device, $y \in \mathbb{R}^{m \cdot c'}$ is the observation vector, and $n \in \mathbb{R}^{m \cdot c'}$ is a noise vector that models all approximation errors of the forward model and measurement noise. Hereafter, we will assume that $n$ consists of i.i.d samples drawn from a Gaussian distribution of zero mean and variance $\sigma_n^2$. Note that despite of the seeming simplicity of this observation model, it is widely used in the literature, since it can accurately enough describe a plethora of practical problems. Specifically, by varying the form of $A$, Eq. (1) can cover many different inverse imaging problems such as denoising, deblurring, demosaicking, inpainting, super-resolution, MRI reconstruction, etc. If we further define the objective function:

$$\mathcal{J}(x) = \frac{1}{2\sigma_n^2} \|y - Ax\|_2^2 + \mathcal{R}(x), \tag{2}$$

where $\mathcal{R} : \mathbb{R}^{n \cdot c} \to \mathbb{R}_+ = \{x \in \mathbb{R} | x \geq 0\}$ is the regularizer (image prior), we can recover an estimate of the latent image $x^*$ as the minimizer of the optimization problem: $x^* = \arg\min_x \mathcal{J}(x)$. Since the type of the regularizer $\mathcal{R}(x)$ can significantly affect the reconstruction quality, it is of the utmost importance to employ a proper regularizer for the reconstruction task at hand.

### 2.1 SPARSE AND LOW-RANK IMAGE PRIORS

Most of the existing image regularizers in the literature can be written in the generic form:

$$\mathcal{R}(x) = \sum_{i=1}^{\ell} \phi(G_i x), \tag{3}$$

where $G : \mathbb{R}^{n \cdot c} \to \mathbb{R}^{\ell \cdot d}$ is the *regularization operator* that transforms $x$, $G_i = M_i G$, $M_i = I_d \otimes e_i^\mathsf{T}$ with $\otimes$ denoting the Kronecker product, and $e_i$ is the unit vector of the standard $\mathbb{R}^\ell$ basis. Further, $\phi : \mathbb{R}^d \to \mathbb{R}_+$ is a potential function that penalizes the response of the $d$-dimensional transform-domain feature vector, $z_i = G_i x \in \mathbb{R}^d$. Among such regularizers, widely used are those that promote sparse and low-rank responses by utilizing as their potential functions the $\ell_1$ and nuclear norms (Rudin et al., 1992; Figueiredo et al., 2007; Lefkimmiatis et al., 2013; 2015). Enforcing sparsity of the solution in some transform-domain has been studied in-depth and is supported both by solid mathematical theory (Donoho, 2006; Candes & Wakin, 2008; Elad, 2010) as well as strong empirical results, which indicate that distorted images do not typically exhibit sparse or low-rank representations, as opposed to their clean counterparts. More recently it has also been advocated that non-convex penalties such as the $\ell_p^p$ vector and $\mathcal{S}_p^p$ Schatten-matrix quasi-norms enforce sparsity better and lead to improved image reconstruction results (Chartrand, 2007; Lai et al., 2013; Candes et al., 2008; Gu et al., 2014; Liu et al., 2014; Xie et al., 2016; Kümmerle & Verdun, 2021).

Based on the above, we consider two expressive parametric forms for the potential function $\phi(\cdot)$, which correspond to weighted and smooth extensions of the $\ell_p^p$ and the Schatten matrix $\mathcal{S}_p^p$ quasi-norms with $0 < p \leq 1$, respectively. The first one is a sparsity-promoting penalty, defined as:

$$\phi_{sp}(z; w, p) = \sum_{j=1}^{d} w_j \left(z_j^2 + \gamma\right)^{\frac{p}{2}}, \ z, w \in \mathbb{R}^d, \tag{4}$$

while the second one is a low-rank (spectral-domain sparsity) promoting penalty, defined as:

$$\phi_{lr}(Z; w, p) = \sum_{j=1}^{r} w_j \left(\sigma_j^2(Z) + \gamma\right)^{\frac{p}{2}}, \ Z \in \mathbb{R}^{m \times n}, w \in \mathbb{R}_+^r, \text{ with } r = \min(m, n). \tag{5}$$

In both definitions $\gamma$ is a small fixed constant that ensures the smoothness of the penalty functions. Moreover, in Eq. (5) $\boldsymbol{\sigma}(\boldsymbol{Z})$ denotes the vector with the singular values of $\boldsymbol{Z}$ sorted in decreasing order, while the weights $\boldsymbol{w}$ are sorted in increasing order. The reason for the latter order is that to better promote low-rank solutions we need to penalize more the smaller singular values of the matrix than its larger ones. Next, we define our proposed sparse and low-rank promoting image priors as:

$$\mathcal{R}_{sp}(\boldsymbol{x}) = \sum_{i=1}^{\ell} \phi_{sp}(\boldsymbol{z}_i; \boldsymbol{w}_i, p), \quad \text{(6a)} \qquad\qquad \mathcal{R}_{lr}(\boldsymbol{x}) = \sum_{i=1}^{\ell} \phi_{lr}(\boldsymbol{Z}_i; \boldsymbol{w}_i, p), \quad \text{(6b)}$$

where in Eq. (6a) $\boldsymbol{z}_i = \boldsymbol{G}_i \boldsymbol{x} \in \mathbb{R}^d$, while in Eq. (6b) $\boldsymbol{Z}_i \in \mathbb{R}^{c \times q}$ is a matrix whose dependence on $\boldsymbol{x}$ is expressed as: $\mathrm{vec}(\boldsymbol{Z}_i) = \boldsymbol{G}_i \boldsymbol{x} \in \mathbb{R}^d$, with $d = c \cdot q$. In words, the $j$-th row of $\boldsymbol{Z}_i$ is formed by the $q$-dimensional feature vector $\boldsymbol{Z}_i^{(j,:)}$ extracted from the image channel $\boldsymbol{x}^j, j = 1, \ldots, c$. The motivation for enforcing the matrices $\boldsymbol{Z}_i$ to have low-rank is that the channels of natural images are typically highly correlated. Thus, it is reasonable to expect that the features stored in the rows of $\boldsymbol{Z}_i$, would be dependent to each other. We note that a similar low-rank enforcing regularization strategy is typically followed by regularizers whose goal is to model the non-local similarity property of natural images Gu et al. (2014); Xie et al. (2016). In these cases the matrices $\boldsymbol{Z}_i$ are formed in such a way so that each of their rows holds the elements of a patch extracted from the image, while the entire matrix consists of groups of structurally similar patches.

## 2.2 Majorization-Minimization Strategy

One of the challenges in the minimization of the overall objective function in Eq. (2) is that the image priors introduced in Eq. (6) are generally non-convex w.r.t $\boldsymbol{x}$. This precludes any guarantees of reaching a global minimum, and we can only opt for a stationary point. One way to handle the minimization task would be to employ the gradient descent method. Potential problems in such case are the slow convergence as well as the need to adjust the step-size in every iteration of the algorithm, so as to avoid divergence from the solution. Other possible minimization strategies are variable splitting (VS) techniques such as the Alternating Method of Multipliers (Boyd et al., 2011) and Half-Quadratic splitting (Nikolova & Ng, 2005), or the Fast Iterative Shrinkage Algorithm (FISTA) (Beck & Teboulle, 2009). The underlying idea of such methods is to transform the original problem in easier to solve sub-problems. However, VS techniques require finetuning of additional algorithmic parameters, to ensure that a satisfactory convergence rate is achieved, while FISTA works-well under the assumption that the proximal map of the regularizer (Boyd & Vandenberghe, 2004) is not hard to compute. Unfortunately, this is not the case for the regularizers considered in this work.

For all the above reasons, here we pursue a majorization-minimization (MM) approach (Hunter & Lange, 2004), which does not pose such strict requirements. Under the MM approach, instead of trying to minimize the objective function $\mathcal{J}(\boldsymbol{x})$ directly, we follow an iterative procedure where each iteration consists of two steps: (**a**) selection of a surrogate function that serves as a tight upper-bound of the original objective (*majorization-step*) and (**b**) computation of a current estimate of the solution by minimizing the surrogate function (*minimization-step*). Specifically, the iterative algorithm for solving the minimization problem $\boldsymbol{x}^* = \arg\min_{\boldsymbol{x}} \mathcal{J}(\boldsymbol{x})$ takes the form: $\boldsymbol{x}^{k+1} = \arg\min_{\boldsymbol{x}} \mathcal{Q}(\boldsymbol{x}; \boldsymbol{x}^k)$, where $\mathcal{Q}(\boldsymbol{x}; \boldsymbol{x}^k)$ is the majorizer of the objective function $\mathcal{J}(\boldsymbol{x})$ at some point $\boldsymbol{x}^k$, satisfying the two conditions:

$$\mathcal{Q}(\boldsymbol{x}; \boldsymbol{x}^k) \geq \mathcal{J}(\boldsymbol{x}), \forall \boldsymbol{x}, \boldsymbol{x}^k \quad \text{and} \quad \mathcal{Q}(\boldsymbol{x}^k; \boldsymbol{x}^k) = \mathcal{J}(\boldsymbol{x}^k). \quad (7)$$

Given these two properties of the majorizer, it can be easily shown that iteratively minimizing $\mathcal{Q}(\boldsymbol{x}; \boldsymbol{x}^k)$ also monotonically decreases the objective function $\mathcal{J}(\boldsymbol{x})$ (Hunter & Lange, 2004). In fact, to ensure this, it only suffices to find a $\boldsymbol{x}^{k+1}$ that decreases the value of the majorizer, *i.e.*, $\mathcal{Q}(\boldsymbol{x}^{k+1}; \boldsymbol{x}^k) \leq \mathcal{Q}(\boldsymbol{x}^k; \boldsymbol{x}^k)$. Moreover, given that both $\mathcal{Q}(\boldsymbol{x}; \boldsymbol{x}^k)$ and $\mathcal{J}(\boldsymbol{x})$ are bounded from below, we can safely state that upon convergence we will reach a stationary point.

The success of the described iterative strategy solely relies on our ability to efficiently minimize the chosen majorizer. Noting that the data fidelity term of the objective is quadratic, we proceed by seeking a quadratic majorizer for the image prior $\mathcal{R}(\boldsymbol{x})$. This way the overall majorizer will be of quadratic form, which is amenable to efficient minimization by solving the corresponding normal equations. Below we provide two results that allow us to design such tight quadratic upper bounds both for the sparsity-promoting (6a) and the low-rank promoting (6b) regularizers. Their proofs are

provided in Appendix A.1. We note that, to the best of our knowledge, the derived upper-bound presented in Lemma 2 is novel and it can find use in a wider range of applications, which also utilize low-rank inducing penalties (Hu et al., 2021), than the ones we focus here.

**Lemma 1.** *Let $\boldsymbol{x}, \boldsymbol{y} \in \mathbb{R}^n$ and $\boldsymbol{w} \in \mathbb{R}_+^n$. The weighted-$\ell_p^p$ function $\phi_{sp}\left(\boldsymbol{x}; \boldsymbol{w}, p\right)$ defined in Eq. (4) can be upper-bounded as:*

$$\phi_{sp}\left(\boldsymbol{x}; \boldsymbol{w}, p\right) \leq \frac{p}{2} \boldsymbol{x}^\mathsf{T} \boldsymbol{W_y} \boldsymbol{x} + \frac{p\gamma}{2} \operatorname{tr}\left(\boldsymbol{W_y}\right) + \frac{2-p}{2} \phi_{sp}\left(\boldsymbol{y}; \boldsymbol{w}, p\right), \, \forall \, \boldsymbol{x}, \boldsymbol{y} \tag{8}$$

*where $\boldsymbol{W_y} = \operatorname{diag}\left(\boldsymbol{w}\right) \left[\boldsymbol{I} \circ \left(\boldsymbol{y}\boldsymbol{y}^\mathsf{T} + \gamma\boldsymbol{I}\right)\right]^{\frac{p-2}{2}}$ and $\circ$ denotes the Hadamard product. The equality in (8) is attained when $\boldsymbol{x} = \boldsymbol{y}$.*

**Lemma 2.** *Let $\boldsymbol{X}, \boldsymbol{Y} \in \mathbb{R}^{m \times n}$ and $\boldsymbol{\sigma}\left(\boldsymbol{Y}\right), \boldsymbol{w} \in \mathbb{R}_+^r$ with $r = \min\left(m, n\right)$. The vector $\boldsymbol{\sigma}\left(\boldsymbol{Y}\right)$ holds the singular values of $\boldsymbol{Y}$ in decreasing order while the elements of $\boldsymbol{w}$ are sorted in increasing order. The weighted-Schatten-matrix function $\phi_{lr}\left(\boldsymbol{X}; \boldsymbol{w}, p\right)$ defined in Eq. (5) can be upper-bounded as:*

$$\phi_{lr}\left(\boldsymbol{X}; \boldsymbol{w}, p\right) \leq \frac{p}{2} \operatorname{tr}\left(\boldsymbol{W_Y} \boldsymbol{X} \boldsymbol{X}^\mathsf{T}\right) + \frac{p\gamma}{2} \operatorname{tr}\left(\boldsymbol{W_Y}\right) + \frac{2-p}{2} \phi_{lr}\left(\boldsymbol{Y}; \boldsymbol{w}, p\right), \, \forall \, \boldsymbol{X}, \boldsymbol{Y} \tag{9}$$

*where $\boldsymbol{W_Y} = \boldsymbol{U} \operatorname{diag}\left(\boldsymbol{w}\right) \boldsymbol{U}^\mathsf{T} \left(\boldsymbol{Y}\boldsymbol{Y}^\mathsf{T} + \gamma\boldsymbol{I}\right)^{\frac{p-2}{2}}$ and $\boldsymbol{Y} = \boldsymbol{U} \operatorname{diag}\left(\boldsymbol{\sigma}\left(\boldsymbol{Y}\right)\right) \boldsymbol{V}^\mathsf{T}$, with $\boldsymbol{U} \in \mathbb{R}^{m \times r}$ and $\boldsymbol{V} \in \mathbb{R}^{n \times r}$. The equality in (9) is attained when $\boldsymbol{X} = \boldsymbol{Y}$.*

Next, with the help of the derived tight upper-bounds we obtain the quadratic majorizers for both of our regularizers as:

$$\mathcal{Q}_{sp}\left(\boldsymbol{x}; \boldsymbol{x}^k\right) = \frac{p}{2} \sum_{i=1}^{\ell} \boldsymbol{z}_i^\mathsf{T} \boldsymbol{W}_{\boldsymbol{z}_i^k} \boldsymbol{z}_i, \quad \text{(10a)} \qquad \mathcal{Q}_{lr}\left(\boldsymbol{x}; \boldsymbol{x}^k\right) = \frac{p}{2} \sum_{i=1}^{\ell} \operatorname{tr}\left(\boldsymbol{Z}_i^\mathsf{T} \boldsymbol{W}_{\boldsymbol{Z}_i^k} \boldsymbol{Z}_i\right), \quad \text{(10b)}$$

where $\boldsymbol{z}_i$ and $\boldsymbol{Z}_i$ are defined as in Eq. (6), $\boldsymbol{W}_{\boldsymbol{z}_i^k}, \boldsymbol{W}_{\boldsymbol{Z}_i^k}$ are defined in Lemmas 1 and 2, respectively, and we have ignored all the constant terms that do not depend on $\boldsymbol{x}$. In both cases, by adding the majorizer of the regularizer, $\mathcal{Q}_{reg}\left(\boldsymbol{x}; \boldsymbol{x}^k\right)$, to the quadratic data fidelity term we obtain the overall majorizer $\mathcal{Q}\left(\boldsymbol{x}; \boldsymbol{x}^k\right)$, of the objective function $\mathcal{J}\left(\boldsymbol{x}\right)$. Since this majorizer is quadratic, it is now possible to obtain the $(k+1)$-th update of the MM iteration by solving the normal equations:

$$\boldsymbol{x}^{k+1} = \left(\boldsymbol{A}^\mathsf{T}\boldsymbol{A} + p \cdot \sigma_{\boldsymbol{n}}^2 \sum_{i=1}^{\ell} \boldsymbol{G}_i^\mathsf{T} \boldsymbol{W}_i^k \boldsymbol{G}_i + \alpha\boldsymbol{I}\right)^{-1} \left(\boldsymbol{A}^\mathsf{T}\boldsymbol{y} + \alpha\boldsymbol{x}^k\right) = \left(\boldsymbol{S}^k + \alpha\boldsymbol{I}\right)^{-1} \boldsymbol{b}^k, \tag{11}$$

where $\alpha = \delta\sigma_{\boldsymbol{n}}^2$, $\boldsymbol{W}_i^k = \boldsymbol{W}_{\boldsymbol{z}_i^k} \in \mathbb{R}^{d \times d}$ for $\mathcal{R}_{sp}\left(\boldsymbol{x}\right)$, and $\boldsymbol{W}_i^k = \boldsymbol{I}_q \otimes \boldsymbol{W}_{\boldsymbol{Z}_i^k} \in \mathbb{R}^{c \cdot q \times c \cdot q}$ for $\mathcal{R}_{lr}\left(\boldsymbol{x}\right)$. We note that, to ensure that the system matrix in Eq. (11) is invertible, we use an augmented majorizer that includes the additional term $\frac{\delta}{2} \left\| \boldsymbol{x} - \boldsymbol{x}^k \right\|_2^2$, with $\delta > 0$ being a fixed small constant (we refer to Appendix A.1.1 for a justification of the validity of this strategy). This leads to the presence of the extra term $\alpha\boldsymbol{I}$ additionally to the system matrix $\boldsymbol{S}^k$ and of $\alpha\boldsymbol{x}^k$ in $\boldsymbol{b}^k$. Based on the above, the minimization of $\mathcal{J}\left(\boldsymbol{x}\right)$, incorporating any of the two regularizers of Eq. (6), boils down to solving a sequence of re-weighted least squares problems, where the weights $\boldsymbol{W}_i^k$ of the current iteration are updated using the solution of the previous one. Given that our regularizers in Eq. (6) include the weights $\boldsymbol{w}_i \neq \boldsymbol{1}$, our proposed algorithm generalizes the IRLS methods introduced by Daubechies et al. (2010) and Mohan & Fazel (2012), which only consider the case where $\boldsymbol{w}_i = \boldsymbol{1}$ and have been successfully applied in the past on sparse and low-rank recovery problems, respectively.

## 3 LEARNING PRIORS USING IRLS RECONSTRUCTION NETWORKS

To deploy the proposed IRLS algorithm, we have to specify both the regularization operator $\boldsymbol{G}$ and the parameters $\boldsymbol{w} = \{\boldsymbol{w}_i\}_{i=1}^{\ell}, p$ of the potential functions in Eqs. (4) and (5), which constitute the image regularizers of Eq. (6). Manually selecting their values, with the goal of achieving satisfactory reconstructions, can be a cumbersome task. Thus, instead of explicitly defining them, we pursue the idea of implementing IRLS as a recurrent network, and learn their values in a supervised way. Under this approach, our learned IRLS-based reconstruction network (LIRLS) can be described as: $\boldsymbol{x}^{k+1} = f_{\boldsymbol{\theta}}\left(\boldsymbol{x}^k; \boldsymbol{y}\right)$, with $\boldsymbol{\theta} = \{\boldsymbol{G}, \boldsymbol{w}, p\}$ denoting its parameters. The network itself consists of three main components: (**a**) A feature extraction layer that accepts as input the current reconstruction estimate, $\boldsymbol{x}^k$, and outputs the feature maps $\{\boldsymbol{z}_i^k = \boldsymbol{G}_i \boldsymbol{x}^k\}_{i=1}^{\ell}$. (**b**) The weight module that acts on $\boldsymbol{z}_i^k$ and the parameters $\boldsymbol{w}_i$ to construct the weight matrix $\boldsymbol{W}_i^k$, which is part of the system matrix $\boldsymbol{S}^k$ in Eq. (11). (**c**) The least-squares (LS) layer whose role is to produce a refined reconstruction estimate, $\boldsymbol{x}^{k+1}$, as the solution of Eq. (11). The overall architecture of LIRLS is shown in Fig. 1.

## 3.1 NETWORK TRAINING

A common training strategy for recurrent networks is to unroll the network using a fixed number of iterations and update its parameters either by means of back-propagation through time (BPTT) or by its truncated version (TBPTT) (Robinson & Fallside, 1987; Kokkinos & Lefkim-

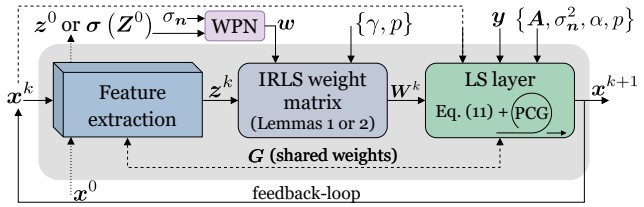

Figure 1: The proposed LIRLS recurrent architecture.

miatis, 2019). However, this strategy cannot be efficiently applied to LIRLS. The reason is that the system matrix $\boldsymbol{S}^k$ in Eq. (11) typically is of huge dimensions and its direct inversion is generally infeasible. Thus, to compute $\boldsymbol{x}^{k+1}$ via Eq. (11) we need to rely on a matrix-free iterative linear solver such as the conjugate gradient method (CG) (Shewchuk, 1994). This means that apart from the IRLS iterations we need also to take into account the internal iterations required by the linear solver. Therefore, unrolling both types of iterations would result in a very deep network, whose efficient training would be extremely challenging for two main reasons. The first is the required amount of memory, which would be prohibitively large. The second one is the problem of vanishing/exploding gradients that appears in recurrent architectures (Pascanu et al., 2013), which would be even more pronounced in the case of LIRLS due to its very deep structure.

To overcome these problems, we rely on the fact that our IRLS strategy guarantees the convergence to a fixed point $\boldsymbol{x}^*$. Practically, this means that upon convergence of IRLS, it will hold $\boldsymbol{x}^* = f_{\boldsymbol{\theta}}\left(\boldsymbol{x}^*; \boldsymbol{y}\right)$. Considering the form of our IRLS iterations, as shown in Eq. (11), this translates to:

$$g\left(\boldsymbol{x}^*, \boldsymbol{\theta}\right) \equiv \boldsymbol{x}^* - f_{\boldsymbol{\theta}}\left(\boldsymbol{x}^*; \boldsymbol{y}\right) = \boldsymbol{S}^*\left(\boldsymbol{x}^*, \boldsymbol{\theta}\right)\boldsymbol{x}^* - \boldsymbol{A}^\top\boldsymbol{y} = \boldsymbol{0}, \tag{12}$$

where we explicitly indicate the dependency of $\boldsymbol{S}^*$ on $\boldsymbol{x}^*$ and $\boldsymbol{\theta}$. To update the network's parameters during training, we have to compute the gradients $\nabla_{\boldsymbol{\theta}}\mathcal{L}\left(\boldsymbol{x}^*\right) = \nabla_{\boldsymbol{\theta}}\boldsymbol{x}^* \nabla_{\boldsymbol{x}^*}\mathcal{L}\left(\boldsymbol{x}^*\right)$, where $\mathcal{L}$ is a loss function. Now, if we differentiate both sides of Eq. (12) w.r.t $\boldsymbol{\theta}$, then we get:

$$\frac{\partial g\left(\boldsymbol{x}^*, \boldsymbol{\theta}\right)}{\partial \boldsymbol{\theta}} + \frac{\partial g\left(\boldsymbol{x}^*, \boldsymbol{\theta}\right)}{\partial \boldsymbol{x}^*}\frac{\partial \boldsymbol{x}^*}{\partial \boldsymbol{\theta}} = \boldsymbol{0} \Rightarrow \nabla_{\boldsymbol{\theta}}\boldsymbol{x}^* = -\nabla_{\boldsymbol{\theta}}g\left(\boldsymbol{x}^*, \boldsymbol{\theta}\right)\left(\nabla_{\boldsymbol{x}^*}g\left(\boldsymbol{x}^*, \boldsymbol{\theta}\right)\right)^{-1}. \tag{13}$$

Thus, we can now compute the gradient of the loss function w.r.t the network's parameters as:

$$\nabla_{\boldsymbol{\theta}}\mathcal{L}\left(\boldsymbol{x}^*\right) = -\nabla_{\boldsymbol{\theta}}g\left(\boldsymbol{x}^*, \boldsymbol{\theta}\right)\boldsymbol{v}, \tag{14}$$

where $\boldsymbol{v}$ is obtained as the solution of the linear problem $\nabla_{\boldsymbol{x}^*}g\left(\boldsymbol{x}^*, \boldsymbol{\theta}\right)\boldsymbol{v} = \nabla_{\boldsymbol{x}^*}\mathcal{L}\left(\boldsymbol{x}^*\right)$ and all the necessary auxiliary gradients can be computed via automatic differentiation. Based on the above, we can train the LIRLS network without having to restrict its overall effective depth or save any intermediate results that would significantly increase the memory requirements. The implementation details of our network for both its forward and backward passes, are described in Algorithm 1 in Sec. A.4. Finally, while our training strategy is similar in spirit with the one used for training Deep Equilibrium Models (DEQ) (Bai et al., 2019), an important difference is that our recurrent networks are guaranteed to converge to a fixed point, while in general this is not true for DEQ models.

## 3.2 LIRLS NETWORK IMPLEMENTATION

In this section we discuss implementation details for the LIRLS variants whose performance we will assess on different grayscale/color image reconstruction tasks. As mentioned earlier, among the parameters that we aim to learn is the regularization operator $\boldsymbol{G}$. For all the different networks we parameterize $\boldsymbol{G}$ with a valid convolution layer that consists of 24 filters of size $5 \times 5$. In the case of color images these filters are shared across channels. Further, depending on the particular LIRLS instance that we utilize, we either fix the values of the parameters $\boldsymbol{w}, p$ or we learn them during training. Hereafter, we will use the notations $\ell_p^{p,\boldsymbol{w}}$ and $\mathcal{S}_p^{p,\boldsymbol{w}}$ to refer to the networks that employ a learned sparse-promoting prior and a learned low-rank promoting prior, respectively. The different LIRLS instances that we consider in this work are listed below:

1. $\ell_1/\mathcal{S}_1$ (nuclear): fixed $p = 1$, fixed $\boldsymbol{w} = \boldsymbol{1}$, where $\boldsymbol{1}$ is a vector of ones.

2. Weighted $\ell_1^{\boldsymbol{w}}/\mathcal{S}_1^{\boldsymbol{w}}$ (weighted nuclear): fixed $p = 1$, weights $\boldsymbol{w}$ are computed by a weight prediction neural network (WPN). WPN accepts as inputs either the features $\boldsymbol{z}_0 = \hat{\boldsymbol{G}}\boldsymbol{x}_0$ for the grayscale case or their singluar values for the color case, as well as the noise standard deviation $\sigma_{\boldsymbol{n}}$. The

Table 1: Comparisons on grayscale image deblurring.

| Dataset | | TV-$\ell_1$ | TV | $\ell_1$ | $\ell_p^p$ | $\ell_1^w$ | $\ell_p^{p,w}$ | RED | IRCNN | FDN |
|---|---|---|---|---|---|---|---|---|---|---|
| Sun et al. (2013) | PSNR | 30.81 | 30.96 | 31.98 | 32.36 | **32.70** | 32.67 | 31.68 | 32.55 | 32.63 |
| | SSIM | 0.8272 | 0.8351 | 0.8759 | 0.8849 | **0.8965** | 0.8960 | 0.8573 | 0.8860 | 0.8894 |
| Levin et al. (2009) | PSNR | 34.74 | 34.92 | 35.19 | 35.33 | **35.75** | 35.60 | 35.19 | 32.86 | 35.08 |
| | SSIM | 0.9552 | 0.9568 | 0.9600 | 0.9617 | **0.9647** | 0.9628 | 0.9584 | 0.9118 | 0.9609 |

Table 2: Comparisons on color image deblurring.

| | TVN | VTV | $\mathcal{S}_1$ | $\mathcal{S}_p^p$ | $\mathcal{S}_1^w$ | $\mathcal{S}_p^{p,w}$ | RED | IRCNN | FDN | DWDN |
|---|---|---|---|---|---|---|---|---|---|---|
| PSNR | 31.71 | 31.38 | 33.09 | 34.17 | 33.95 | **34.24** | 31.38 | 33.19 | 32.51 | 34.09 |
| SSIM | 0.8506 | 0.8440 | 0.8995 | 0.9227 | 0.9179 | **0.9234** | 0.8233 | 0.9123 | 0.8857 | 0.9197 |

vector $\boldsymbol{x}_0$ is the estimate obtained after 5 IRLS steps of the pretrained $\ell_1/\mathcal{S}_1$ networks, while $\hat{\boldsymbol{G}}$ is their learned regularization operator. For all studied problems except of MRI reconstruction, we use a lightweight RFDN architecture proposed by Liu et al. (2020) to predict the weights, while for MRI reconstruction we use a lightweight UNet from Ronneberger et al. (2015), which we have found to be more suitable for this task. For the $\mathcal{S}_1^{\boldsymbol{w}}$ network, in order to enforce the predicted weights to be sorted in increasing order, we apply across channels of the ouput of WPN a cumulative sum. In all cases, the number of parameters of WPN does not exceed 0.3M.

3. $\ell_p^p/\mathcal{S}_p^p$: learned $p \in [0.4, 0.9]$, fixed $\boldsymbol{w} = \boldsymbol{1}$.

4. $\ell_p^{p,\boldsymbol{w}}/S_p^{p,\boldsymbol{w}}$: learned $p \in [0.4, 0.9]$, weights $\boldsymbol{w}$ are computed as described in item 2, with the only difference being that both $\boldsymbol{x}_0$ and $\hat{\boldsymbol{G}}$ are now obtained from the pretarained $\ell_p^p/\mathcal{S}_p^p$ networks.

We note that the output of the LS layer, which corresponds to the solution of the normal equations in Eq. (11), is computed by utilizing a preconditioned version of CG (PCG) Hestenes & Stiefel (1952). This allows for an improved convergence of the linear solver. Details about our adopted preconditioning strategy are provided in Appendix A.2. Finally, in all the reported cases, the constant $\delta$ related to the parameter $\alpha$ in Eq. (11) is set to $8e^{-4}$, while as initial solution for the linear solver we use the output of an independently trained fast FFT-based Wiener filter.

### 3.3 TRAINING DETAILS

The convergence of the forward pass of LIRLS to a fixed point $\boldsymbol{x}^*$ is determined according to the criterion: $||\boldsymbol{S}^*(\boldsymbol{x}^*, \boldsymbol{\theta}) \boldsymbol{x}^* - \boldsymbol{A}^\top \boldsymbol{y}||_2 / ||\boldsymbol{A}^\top \boldsymbol{y}||_2 < 1e^{-4}$, which needs to be satisfied for 3 consecutive IRLS steps. If this criterion is not satisfied, the forward pass is terminated after 400 IRLS steps during training and 15 steps during inference. In the LS layers we use at most 150 CG iterations during training and perform an early exit if the relative tolerance of the residual falls below $1e^{-6}$, while during inference the maximum amount of CG iterations is reduced to 50. When training LIRLS, in the backward pass, as shown in Eq. (14), we are required to solve a linear problem whose system matrix corresponds to the Hessian of the objective function (2). This symmetric matrix is positive definite when $\mathcal{J}(\boldsymbol{x})$ is convex and indefinite otherwise. In the former case we utilize the CG algorithm to solve the linear problem, while in the latter one we use the Minimal Residual Method (MINRES) (Paige & Saunders, 1975). We perform early exit if the relative tolerance of the residual is below $1e^{-2}$ and limit the maximum amount of iterations to 2000. It should be noted that we have experimentally found these values to be adequate for achieving stable training.

All our models are trained using as loss function the negative peak-to-signal-noise-ratio (PSNR) between the ground truth and the network's output. We use the Adam optimizer (Kingma & Ba, 2015) with a learning rate $5e^{-3}$ for all the models that do not involve a WPN and $1e^{-4}$ otherwise. The learning rate is decreased by a factor of 0.98 after each epoch. On average we set the batch size to 8 and train our models for 100 epochs, where each epoch consists of 500 batch passes.

## 4 EXPERIMENTS

In this section we assess the performance of all the LIRLS instances described in Sec. 3.2 on four different image reconstruction tasks, namely image deblurring, super-resolution, demosaicking and MRI reconstruction. In all these cases the only difference in the objective function $\mathcal{J}(\boldsymbol{x})$ is the form of the degradation operator $\boldsymbol{A}$. Specifically, the operator $\boldsymbol{A}$ has one of the following forms: (**a**) low-pass valid convolutional operator (*deblurring*), (**b**) composition of a low-pass valid convolutional operator and a decimation operator (*super-resolution*), (**c**) color filter array (CFA) operator (*demosaicking*), and (**d**) sub-sampled Fourier operator (MRI *reconstruction*). The first two recovery tasks are related to either grayscale or color images, demosaicking is related to color images, and MRI reconstruction is related to single-channel images.

Table 3: Comparisons on grayscale image super-resolution.

| Scale | Noise | | TV-$\ell_1$ | TV | $\ell_1$ | $\ell_p^p$ | $\ell_1^w$ | $\ell_p^{p,w}$ | Bicubic | RED | IRCNN | USRNet |
|---|---|---|---|---|---|---|---|---|---|---|---|---|
| x2 | 0% | PSNR | 26.23 | 28.04 | 28.20 | 27.98 | **28.76** | 28.31 | 24.74 | 28.34 | 28.48 | 28.58 |
| | | SSIM | 0.6937 | 0.8028 | 0.8064 | 0.7953 | 0.8186 | 0.8067 | 0.6699 | 0.8213 | 0.8324 | **0.8188** |
| | 1% | PSNR | 17.37 | 26.64 | 26.81 | 26.64 | **27.32** | 26.94 | 24.63 | 26.88 | 26.22 | 27.26 |
| | | SSIM | 0.2739 | 0.7320 | 0.7379 | 0.7292 | **0.7610** | 0.7470 | 0.6521 | 0.7275 | 0.6799 | 0.7607 |
| x3 | 0% | PSNR | 24.50 | 25.55 | 25.63 | 25.44 | 26.02 | 25.67 | 23.34 | 25.70 | 25.91 | **26.06** |
| | | SSIM | 0.5875 | 0.6825 | 0.6857 | 0.6741 | 0.6975 | 0.6832 | 0.5872 | 0.7006 | 0.7144 | **0.7101** |
| | 1% | PSNR | 14.26 | 24.62 | 24.67 | 24.48 | 25.03 | 24.79 | 23.26 | 24.81 | 24.70 | **25.07** |
| | | SSIM | 0.1812 | 0.6276 | 0.6306 | 0.6195 | 0.6472 | 0.6353 | 0.5739 | 0.6417 | 0.6219 | **0.6586** |
| x4 | 0% | PSNR | 22.51 | 24.07 | 24.09 | 23.92 | **24.16** | 24.10 | 22.27 | 24.15 | 23.72 | 23.30 |
| | | SSIM | 0.4773 | 0.6031 | 0.6052 | 0.5960 | 0.6061 | 0.6010 | 0.5342 | 0.6164 | 0.6168 | **0.6203** |
| | 1% | PSNR | 11.81 | 23.31 | 23.32 | 23.13 | 23.35 | **23.38** | 22.21 | 23.35 | 23.44 | 21.41 |
| | | SSIM | 0.1228 | 0.5627 | 0.5643 | 0.5551 | **0.5661** | 0.5648 | 0.5247 | 0.5731 | 0.5713 | 0.4945 |

Table 4: Comparisons on color image super-resolution.

| Scale | Noise | | TVN | VTV | $\mathcal{S}_1$ | $\mathcal{S}_p^p$ | $\mathcal{S}_1^w$ | $\mathcal{S}_p^{p,w}$ | Bicubic | RED | IRCNN | USRNet |
|---|---|---|---|---|---|---|---|---|---|---|---|---|
| x2 | 0% | PSNR | 28.01 | 28.13 | 28.21 | 28.62 | 28.42 | 28.78 | 24.73 | 28.68 | 28.33 | **28.86** |
| | | SSIM | 0.8035 | 0.8042 | 0.8088 | 0.8203 | 0.8212 | 0.8217 | 0.6661 | 0.8210 | 0.8331 | **0.8359** |
| | 1% | PSNR | 26.87 | 26.86 | 27.06 | 27.43 | 27.46 | 27.48 | 24.62 | 25.74 | 26.37 | **27.97** |
| | | SSIM | 0.7404 | 0.7392 | 0.7469 | 0.7646 | 0.7640 | 0.7687 | 0.6486 | 0.6215 | 0.6843 | **0.7924** |
| x3 | 0% | PSNR | 25.46 | 25.62 | 25.50 | 25.60 | 25.35 | 25.90 | 23.33 | 25.90 | 25.74 | **26.17** |
| | | SSIM | 0.6812 | 0.6823 | 0.6828 | 0.6904 | 0.6877 | 0.6928 | 0.5833 | 0.7030 | 0.7092 | **0.7203** |
| | 1% | PSNR | 24.76 | 24.76 | 24.83 | 25.09 | 25.08 | 25.11 | 23.25 | 24.72 | 24.77 | **25.65** |
| | | SSIM | 0.6322 | 0.6314 | 0.6348 | 0.6468 | 0.6452 | 0.6497 | 0.5702 | 0.6075 | 0.6203 | **0.6857** |
| x4 | 0% | PSNR | 24.05 | 24.13 | 24.15 | 24.28 | 24.22 | 24.27 | 22.25 | **24.33** | 23.46 | 23.30 |
| | | SSIM | 0.6018 | 0.6024 | 0.6045 | 0.6111 | 0.6079 | 0.6101 | 0.5305 | 0.6191 | 0.6073 | **0.6238** |
| | 1% | PSNR | 23.43 | 23.42 | 23.46 | **23.66** | 23.53 | **23.66** | 22.19 | 23.64 | 23.40 | 23.57 |
| | | SSIM | 0.5655 | 0.5645 | 0.5666 | 0.5752 | 0.5698 | 0.5774 | 0.5210 | 0.5681 | 0.5643 | **0.6028** |

## 4.1 TRAIN AND TEST DATA

To train our models for the first three recovery tasks, we use random crops of size $64 \times 64$ taken from the *train* and *val* subsets of the BSD500 dataset provided by Martin et al. (2001), while for MRI reconstruction we use $320 \times 320$ single-coil 3T images from the NYU fastMRI knee dataset (Knoll et al. (2020)). In order to train the deblurring networks we use synthetically created blur kernels varying in size from 13 to 35 pixels according to the procedure described by Boracchi & Foi (2012). For the super-resolution task we use scale factors 2, 3 and 4 with $25 \times 25$ kernels randomly synthesized using the algorithm provided by Bell-Kligler et al. (2019). For accelerated MRI we consider $\times 4$ and $\times 8$ undersampling factors in k-space using conjugate-symmetric masks so that training and inference is performed in the real domain. For the demosaicking problem we use the RGGB CFA pattern. All our models are trained and evaluated on a range of noise levels that are typical in the literature for each considered problem. Based on this, we consider $\sigma_n$ to be up to 1% of the maximum image intensity for deblurring, super-resolution and MRI reconstruction tasks, while for demosaicing we use a wider range of noise levels, with $\sigma_n$ being up to 3%.

For our evaluation purposes we use common benchmarks proposed for each recovery task. In particular, for deblurring we use the benchmarks proposed by Sun et al. (2013) and Levin et al. (2009). For super-resolution we use the BSD100RK dataset, which consists of 100 test images from the BSD500 dataset, and the degradation model proposed by Bell-Kligler et al. (2019). For demosaicking we use the images from the McMaster (Zhang et al., 2011) dataset. For MRI reconstruction we use the 3T scans from the *val* subset of the NYU fastMRI knee dataset, where from each scan we take the central slice and two more slices located 8 slices apart from the central one. None of our training data intersect with the data we use for evaluation purposes. We should further note, that all benchmarks we use for evaluation contain diverse sets of images with relatively large resolutions. This strategy is not common for other general purpose methods, which typically report results on a limited set of small images, due to their need to manually fine-tune certain parameters. In our case all network parameters are learned via training and remain fixed during inference. In order to provide a fair comparison, for the methods that do require manual tuning of parameters, we perform a grid search on a smaller subset of images to find the values that lead to the best results. Then we use these values fixed during the evaluation for the entire set of images per each benchmark. Finally, note that for all TV-based regularizers we compute their solutions using an IRLS strategy.

## 4.2 RESULTS

**Deblurring.** In Table 1 we report the average results in terms of PSNR and structure-similarity index measure (SSIM) for several of our LIRLS instances, utilizing different sparsity-promoting priors, and few competing methods, namely anisotropic (TV-$\ell_1$) (Zach et al., 2007) and isotropic

Table 5: Comparisons on image demosaicking.

| Dataset | Noise | | TVN | VTV | $\mathcal{S}_1$ | $\mathcal{S}_p^p$ | $\mathcal{S}_1^{\boldsymbol{w}}$ | $\mathcal{S}_p^{p,\boldsymbol{w}}$ | Bilinear | RED | IRCNN |
|---|---|---|---|---|---|---|---|---|---|---|---|
| | 0% | PSNR | 33.33 | 32.64 | 36.27 | **37.66** | 36.60 | 37.62 | 32.27 | 34.53 | 37.52 |
| | | SSIM | 0.9377 | 0.9285 | 0.9605 | **0.9649** | 0.9606 | 0.9632 | 0.9259 | 0.9338 | 0.9612 |
| McMaster | 1% | PSNR | 33.32 | 32.16 | 35.14 | **36.43** | 35.49 | 36.37 | 31.76 | 34.40 | 36.12 |
| Zhang et al. (2011) | | SSIM | 0.9198 | 0.9076 | 0.9416 | **0.9498** | 0.9451 | 0.9490 | 0.9014 | 0.9208 | 0.9403 |
| | 2% | PSNR | 32.21 | 31.16 | 33.37 | 34.55 | 33.95 | **34.80** | 30.64 | 33.08 | 34.42 |
| | | SSIM | 0.8843 | 0.8638 | 0.9151 | 0.9224 | 0.9226 | **0.9281** | 0.8443 | 0.8821 | 0.9178 |
| | 3% | PSNR | 31.19 | 30.13 | 31.39 | 32.44 | 32.60 | **33.34** | 29.34 | 31.59 | 33.23 |
| | | SSIM | 0.8539 | 0.8164 | 0.8808 | 0.8835 | 0.8997 | **0.9050** | 0.7745 | 0.8308 | 0.8931 |

Table 6: Comparisons on MRI reconstruction.

| Acceleration | | TV-$\ell_1$ | TV | $\ell_1$ | $\ell_p^p$ | $\ell_1^{\boldsymbol{w}}$ | $\ell_p^{p,\boldsymbol{w}}$ | FBP |
|---|---|---|---|---|---|---|---|---|
| x4 | PSNR | 26.31 | 26.40 | 28.53 | 28.99 | **30.11** | 29.75 | 25.53 |
| | SSIM | 0.5883 | 0.5921 | 0.6585 | 0.6701 | **0.7081** | 0.6958 | 0.6014 |
| x8 | PSNR | 24.43 | 24.51 | 25.86 | 26.19 | **28.02** | 27.30 | 24.34 |
| | SSIM | 0.5092 | 0.5120 | 0.5518 | 0.5620 | **0.6168** | 0.5973 | 0.5237 |

Total Variation (Rudin et al., 1992), RED (Romano et al., 2017), IRCNN (Zhang et al., 2017), and FDN (Kruse et al., 2017). From these results we observe that our LIRLS models lead to very competitive performance, despite the relatively small number of learned parameters. In fact, the $\ell_1^{\boldsymbol{w}}$ and $\ell_p^{p,\boldsymbol{w}}$ variants obtain the best results among all methods, including FDN, which is the current grayscale sota method, and IRCNN that involves 4.7M parameters in total (for its 25 denoising networks). Similar comparisons for color images are provided in Table 2, where we report the reconstruction results of our LIRLS variants, which utilize different learned low-rank promoting priors. For these comparisons we further consider the vector-valued TV (VTV) (Blomgren & Chan, 1998), Nuclear TV (TVN) Lefkimmiatis et al. (2015) and DWDN (Dong et al., 2020). From these results we again observe that the LIRLS models perform very well. The most interesting observation though, is that our $\mathcal{S}_p^p$ model with a total of 601 learned parameters manages to outperform the DWD network which involves 7M learned parameters. It is also better than $\mathcal{S}_1^{\boldsymbol{w}}$ and very close to $\mathcal{S}_p^{p,\boldsymbol{w}}$, which indicates that the norm order $p$ plays a more crucial role in this task than the weights $\boldsymbol{w}$.

**Super-resolution.** Similarly to the image deblurring problem, we provide comparisons among competing methods both for grayscale and color images. For these comparisons we still consider the RED and IRCNN methods as before and we further include results from bicubic upsampling and the USRNet network Zhang et al. (2020). The latter network, unlike RED and IRCNN, is specifically designed to deal with the problem of super-resolution and involves 17M of learned parameters. From the reported results we can see that for both the grayscale and color cases the results we obtain with our LIRLS instances are very competitive and they are only slightly inferior than the specialized USRNet network. For a visual assessment of the reconstructions quality we refer to Fig. 3.

**Demosaicking.** For this recovery task we compare our low-rank promoting LIRLS models against the same general-purpose classical and deep learning approaches as in color deblurring, plus bicubic upsamplping. The average results of all the competing methods are reported in Table 5. From these results we come to a similar conclusion as in the debluring case. Specifically, the best performance is achieved by the $\mathcal{S}_p^p$ and $\mathcal{S}_p^{p,\boldsymbol{w}}$ LIRLS models. The first one performs best for lower noise levels while the latter achieves the best results for the highest ones. Visual comparisons among the different methods are provided in Fig. 4.

**MRI Reconstruction.** In Table 6 we compare the performance of our LIRLS models with anisotropic and isotropic TV reconstruction and the Fourier back-projection (FBP) method. While

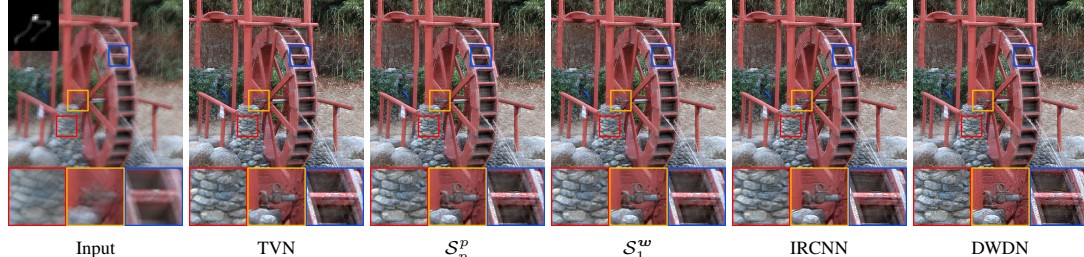

| Input | TVN | $\mathcal{S}_p^p$ | $\mathcal{S}_1^{\boldsymbol{w}}$ | IRCNN | DWDN |

Figure 2: Visual comparisons among several methods on a real blurred color image. Image and blur kernel were taken from Pan et al. (2016). For more visual examples please refer to Appendix A.7.

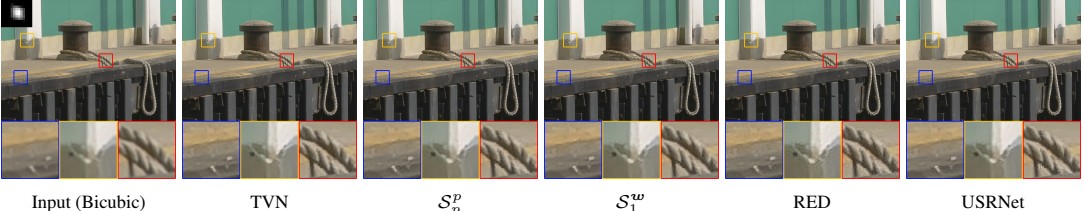

| Input (Bicubic) | TVN | $\mathcal{S}_p^p$ | $\mathcal{S}_1^{\boldsymbol{w}}$ | RED | USRNet |
|---|---|---|---|---|---|

Figure 3: Visual comparisons among several methods on a real low-resolution image enlarged by a scale factor 2. Image from Cai et al. (2019), downscaling kernel was obtained using the method by Bell-Kligler et al. (2019). For more visual examples please refer to Appendix A.7.

these three methods are conventional ones, they can be quite competitive on this task and are still relevant due to their interpretability, which is very important in medical applications. Based on the reported results and the visual comparisons provided in Fig. 5 we can conclude that the LIRLS models do a very good job in terms of reconstruction quality. It is also worth noting that similarly to the other two gray-scale recovery tasks, we observe that the best performance among the LIRLS models is achieved by $\ell_1^{\boldsymbol{w}}$. This can be attributed to the fact that the learned prior of this model while being adaptive, thanks to the presence of the weights, is still convex, unlike $\ell_p^p$ and $\ell_p^{p,\boldsymbol{w}}$. Therefore, its output is not significantly affected by the initial solution. Additional discussions about our networks reconstruction performance related to the choice of $p$ and the weights $\boldsymbol{w}$ is provided in Appendix A.3, along with some possible extensions that we plan to explore in the future.

## 5 CONCLUSIONS

In this work we have demonstrated that our proposed IRLS method, which covers a rich family of sparsity and low-rank promoting priors, can be successfully applied to deal with a wide range of practical inverse problems. In addition, thanks to its convergence guarantees we have managed to use it in the context of supervised learning and efficiently train recurrent networks that involve only a small number of parameters, but can still lead to very competitive reconstructions. Given that most of the studied image priors are non-convex, an interesting open research topic is to analyze how the initial solution affects the output of the LIRLS models and how we can exploit this knowledge to further improve their reconstruction performance. Finally, it would be interesting to explore whether learned priors with spatially adaptive norm orders $p$, can lead to additional improvements.

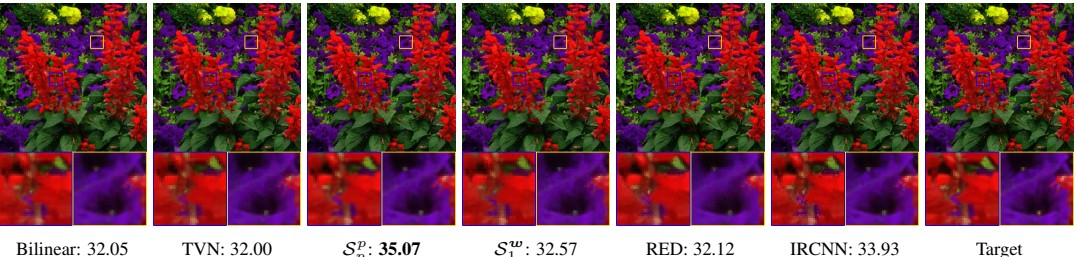

| Bilinear: 32.05 | TVN: 32.00 | $\mathcal{S}_p^p$: **35.07** | $\mathcal{S}_1^{\boldsymbol{w}}$: 32.57 | RED: 32.12 | IRCNN: 33.93 | Target |
|---|---|---|---|---|---|---|

Figure 4: Visual comparisons among several methods on a mosaicked image with 1% noise. For each image its PSNR value is provided in dB. For more visual examples please refer to Appendix A.7.

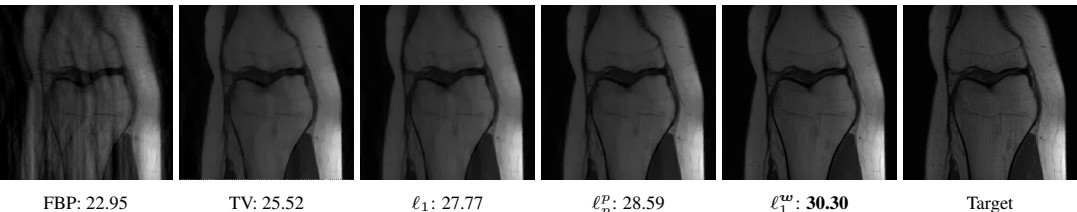

| FBP: 22.95 | TV: 25.52 | $\ell_1$: 27.77 | $\ell_p^p$: 28.59 | $\ell_1^{\boldsymbol{w}}$: **30.30** | Target |
|---|---|---|---|---|---|

Figure 5: Visual comparisons among several methods on a simulated MRI with x4 acceleration. For each image its PSNR value is provided in dB. For more visual examples please refer to Appendix A.7.

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

## A APPENDIX

### A.1 PROOFS

In this section we derive the proofs for the tight upper-bounds presented in Lemma 1 and Lemma 2. For our proofs we use the following inequality for a function $|x|^p, 0 < p \leq 2$, which is given by Sun et al. (2017):

$$|x|^p \leq \frac{p}{2}|y|^{p-2}x^2 + \frac{2-p}{2}|y|^p, \ \forall x \in \mathbb{R} \text{ and } y \in \mathbb{R} \setminus \{0\} \tag{15}$$

and the Ruhe's trace inequality (Marshall et al., 1979), which is stated in the following theorem:

**Theorem 1.** *Let $\boldsymbol{A}$ and $\boldsymbol{B}$ be $n \times n$ positive semidefinite Hermittian matrices. Then it holds that:*

$$\operatorname{tr}(\boldsymbol{AB}) \geq \sum_{i=1}^{n} \sigma_i(\boldsymbol{A}) \sigma_{n-i+1}(\boldsymbol{B}), \tag{16}$$

*where $\sigma_i(\boldsymbol{A})$ denotes the $i$-th singular value of $\boldsymbol{A}$ and the singular values are sorted in a decreasing order, that is $\sigma_i(\boldsymbol{A}) > \sigma_{i+1}(\boldsymbol{A})$.*

*Proof of Lemma 1.* The proof is straightforward and relies on the inequality of Eq. (15). Specifically, let us consider the positive scalars $\sqrt{\boldsymbol{x}_i^2 + \gamma}$, $\sqrt{\boldsymbol{y}_i^2 + \gamma}$, with $\gamma > 0$. If we plug them in (15) we get:

$$\left(\boldsymbol{x}_i^2 + \gamma\right)^{\frac{p}{2}} \leq \frac{p}{2}\left(\boldsymbol{y}_i^2 + \gamma\right)^{\frac{p-2}{2}}\left(\boldsymbol{x}_i^2 + \gamma\right) + \frac{2-p}{2}\left(\boldsymbol{y}_i^2 + \gamma\right)^{\frac{p}{2}}. \tag{17}$$

Multiplying both sides by a non-negative scalar $\boldsymbol{w}_i$ leads to:

$$\boldsymbol{w}_i\left(\boldsymbol{x}_i^2 + \gamma\right)^{\frac{p}{2}} \leq \frac{p}{2}\left(\boldsymbol{y}_i^2 + \gamma\right)^{\frac{p-2}{2}}\boldsymbol{w}_i\left(\boldsymbol{x}_i^2 + \gamma\right) + \frac{2-p}{2}\boldsymbol{w}_i\left(\boldsymbol{y}_i^2 + \gamma\right)^{\frac{p}{2}}. \tag{18}$$

The above inequality is closed under summation and, thus, it further holds that:

$$\phi_{sp}(\boldsymbol{x}; \boldsymbol{w}, p) = \sum_{i=1}^{n} \boldsymbol{w}_i\left(\boldsymbol{x}_i^2 + \gamma\right)^{\frac{p}{2}} \tag{19}$$

$$\leq \frac{p}{2}\sum_{i=1}^{n}\boldsymbol{w}_i\left(\boldsymbol{y}_i^2 + \gamma\right)^{\frac{p-2}{2}}\left(\boldsymbol{x}_i^2 + \gamma\right) + \frac{2-p}{2}\sum_{i=1}^{n}\boldsymbol{w}_i\left(\boldsymbol{y}_i^2 + \gamma\right)^{\frac{p}{2}}$$

$$= \frac{p}{2}\sum_{i=1}^{n}\boldsymbol{w}_i\left(\boldsymbol{y}_i^2 + \gamma\right)^{\frac{p-2}{2}}\boldsymbol{x}_i^2 + \frac{p\gamma}{2}\sum_{i=1}^{n}\boldsymbol{w}_i\left(\boldsymbol{y}_i^2 + \gamma\right)^{\frac{p-2}{2}} + \frac{2-p}{2}\sum_{i=1}^{n}\boldsymbol{w}_i\left(\boldsymbol{y}_i^2 + \gamma\right)^{\frac{p}{2}}$$

$$= \frac{p}{2}\boldsymbol{x}^{\mathsf{T}}\boldsymbol{W_y}\boldsymbol{x} + \frac{p\gamma}{2}\operatorname{tr}(\boldsymbol{W_y}) + \frac{2-p}{2}\phi_{sp}(\boldsymbol{y}; \boldsymbol{w}, p), \ \forall \boldsymbol{x}, \boldsymbol{y} \tag{20}$$

with $\boldsymbol{W_y} = \operatorname{diag}\left(\boldsymbol{w}_1\left(\boldsymbol{y}_1^2 + \gamma\right)^{\frac{p-2}{2}}, \ldots, \boldsymbol{w}_n\left(\boldsymbol{y}_n^2 + \gamma\right)^{\frac{p-2}{2}}\right) = \operatorname{diag}(\boldsymbol{w})\left[\boldsymbol{I} \circ \left(\boldsymbol{yy}^{\mathsf{T}} + \gamma\boldsymbol{I}\right)\right]^{\frac{p-2}{2}}$.

By substitution and carrying over the algebraic operations on the r.h.s of Eq. (20), we can show that when $\boldsymbol{x} = \boldsymbol{y}$, the inequality reduces to equality. $\qquad\square$

We note that it is possible to derive the IRLS algorithm that minimizes $\mathcal{J}(\boldsymbol{x})$ of Eq. (2) under the weighted $\ell_p^p$ regularizers, without relying on the MM framework. In particular, we can redefine the regularizer $\phi_{sp}(\cdot)$ as:

$$\tilde{\phi}_{sp}(\boldsymbol{x}; \boldsymbol{w}, p) = \sum_{i=1}^{n} \boldsymbol{w}_i\left(\boldsymbol{x}_i^2 + \gamma\right)^{\frac{p-2}{2}}\boldsymbol{x}_i^2 \tag{21}$$

from where the weights $\boldsymbol{W_y}$ of Lemma 1 can be inferred. Then, the convergence of the IRLS strategy to a stationary point can be proven according to (Daubechies et al., 2010).

Unfortunately, this strategy doesn't seem to apply for the weights $\boldsymbol{W_Y}$ of Lemma 2. The reason is that the weighted $\mathcal{S}_p^p$ regularizers don't apply directly on the matrix $\boldsymbol{X}$ but instead on its singular values. Specifically, it holds that:

$$\phi_{lr}\left(\boldsymbol{X};\boldsymbol{w},p\right) = \sum_{i=1}^{n} \boldsymbol{w}_i \left(\boldsymbol{\sigma}_i^2\left(\boldsymbol{X}\right)+\gamma\right)^{\frac{p}{2}} = \operatorname{tr}\left(\boldsymbol{W}\left(\boldsymbol{X}\right)\left(\boldsymbol{XX}^\top+\gamma\boldsymbol{I}\right)^{\frac{p}{2}}\right). \tag{22}$$

Unlike the previous case, here the weights $\boldsymbol{w}$ are included in the matrix $\boldsymbol{W}\left(\boldsymbol{X}\right)$, which directly depends on $\boldsymbol{X}$ as: $\boldsymbol{W}\left(\boldsymbol{X}\right) = \boldsymbol{U}\left(\boldsymbol{X}\right)\operatorname{diag}\left(\boldsymbol{w}\right)\boldsymbol{U}^\top\left(\boldsymbol{X}\right)$, with $\boldsymbol{U}\left(\boldsymbol{X}\right)$ being the left singular vectors of $\boldsymbol{X}$. Therefore it is unclear how the approach by Mohan & Fazel (2012) would apply in this case and how the convergence of IRLS can be established.

*Proof of Lemma 2.* Let us consider the positive scalars $\sqrt{\boldsymbol{\sigma}_i^2\left(\boldsymbol{X}\right)+\gamma}$, $\sqrt{\boldsymbol{\sigma}_i^2\left(\boldsymbol{Y}\right)+\gamma}$. If we plug them in (15) we get:

$$\left(\boldsymbol{\sigma}_i^2\left(\boldsymbol{X}\right)+\gamma\right)^{\frac{p}{2}} \leq \frac{p}{2}\left(\boldsymbol{\sigma}_i^2\left(\boldsymbol{Y}\right)+\gamma\right)^{\frac{p-2}{2}}\left(\boldsymbol{\sigma}_i^2\left(\boldsymbol{X}\right)+\gamma\right) + \frac{2-p}{2}\left(\boldsymbol{\sigma}_i^2\left(\boldsymbol{Y}\right)+\gamma\right)^{\frac{p}{2}}. \tag{23}$$

Multiplying both sides by a non-negative scalar $\boldsymbol{w}_i$ leads to:

$$\boldsymbol{w}_i\left(\boldsymbol{\sigma}_i^2\left(\boldsymbol{X}\right)+\gamma\right)^{\frac{p}{2}} \leq \frac{p}{2}\boldsymbol{w}_i\left(\boldsymbol{\sigma}_i^2\left(\boldsymbol{Y}\right)+\gamma\right)^{\frac{p-2}{2}}\left(\boldsymbol{\sigma}_i^2\left(\boldsymbol{X}\right)+\gamma\right) + \frac{2-p}{2}\boldsymbol{w}_i\left(\boldsymbol{\sigma}_i^2\left(\boldsymbol{Y}\right)+\gamma\right)^{\frac{p}{2}}. \tag{24}$$

The above inequality is closed under summation and, thus, it further holds that:

$$\phi_{lr}\left(\boldsymbol{X};\boldsymbol{w},p\right) = \sum_{i=1}^{r}\boldsymbol{w}_i\left(\boldsymbol{\sigma}_i^2\left(\boldsymbol{X}\right)+\gamma\right)^{\frac{p}{2}}$$

$$\leq \frac{p}{2}\sum_{i=1}^{r}\boldsymbol{w}_i\left(\boldsymbol{\sigma}_i^2\left(\boldsymbol{Y}\right)+\gamma\right)^{\frac{p-2}{2}}\left(\boldsymbol{\sigma}_i^2\left(\boldsymbol{X}\right)+\gamma\right) + \frac{2-p}{2}\sum_{i=1}^{r}\boldsymbol{w}_i\left(\boldsymbol{\sigma}_i^2\left(\boldsymbol{Y}\right)+\gamma\right)^{\frac{p}{2}}$$

$$= \frac{p}{2}\sum_{i=1}^{r}\boldsymbol{w}_i\left(\boldsymbol{\sigma}_i^2\left(\boldsymbol{Y}\right)+\gamma\right)^{\frac{p-2}{2}}\boldsymbol{\sigma}_i^2\left(\boldsymbol{X}\right) + \frac{p\gamma}{2}\sum_{i=1}^{r}\boldsymbol{w}_i\left(\boldsymbol{\sigma}_i^2\left(\boldsymbol{Y}\right)+\gamma\right)^{\frac{p-2}{2}} + \frac{2-p}{2}\phi_{lr}\left(\boldsymbol{Y};\boldsymbol{w},p\right)$$

$$= \frac{p}{2}\sum_{i=1}^{r}\boldsymbol{\sigma}_{r-i+1}\left(\boldsymbol{W_Y}\right)\boldsymbol{\sigma}_i\left(\boldsymbol{XX}^T\right) + \frac{p\gamma}{2}\operatorname{tr}\left(\boldsymbol{W_Y}\right) + \frac{2-p}{2}\phi_{lr}\left(\boldsymbol{Y};\boldsymbol{w},p\right), \tag{25}$$

where $\boldsymbol{W_Y} = \boldsymbol{U}\operatorname{diag}\left(\boldsymbol{w}\right)\boldsymbol{U}^\top\left(\boldsymbol{YY}^\top+\gamma\boldsymbol{I}\right)^{\frac{p-2}{2}}$ and $\boldsymbol{Y}$ admits the singular value decomposition $\boldsymbol{Y} = \boldsymbol{U}\operatorname{diag}\left(\boldsymbol{\sigma}\left(\boldsymbol{Y}\right)\right)\boldsymbol{V}^\top$ with $\boldsymbol{U}\in\mathbb{R}^{m\times r}$, $\boldsymbol{V}\in\mathbb{R}^{n\times r}$, and $r=\min\left(m,n\right)$. Further, we show that it holds:

$$\boldsymbol{W_Y} = \boldsymbol{U}\operatorname{diag}\left(\boldsymbol{w}\right)\boldsymbol{U}^\top\left(\boldsymbol{YY}^\top+\gamma\boldsymbol{I}\right)^{\frac{p-2}{2}}$$

$$= \boldsymbol{U}\begin{bmatrix} \boldsymbol{w}_1\left(\boldsymbol{\sigma}_1^2\left(\boldsymbol{Y}\right)+\gamma\right)^{\frac{p-2}{2}} & \cdots & 0 \\ & \ddots & \\ 0 & \cdots & \boldsymbol{w}_r\left(\boldsymbol{\sigma}_r^2\left(\boldsymbol{Y}\right)+\gamma\right)^{\frac{p-2}{2}} \end{bmatrix}\boldsymbol{U}^\top$$

$$= \hat{\boldsymbol{U}}\begin{bmatrix} \boldsymbol{w}_r\left(\boldsymbol{\sigma}_r^2\left(\boldsymbol{Y}\right)+\gamma\right)^{\frac{p-2}{2}} & \cdots & 0 \\ & \ddots & \\ 0 & \cdots & \boldsymbol{w}_1\left(\boldsymbol{\sigma}_1^2\left(\boldsymbol{Y}\right)+\gamma\right)^{\frac{p-2}{2}} \end{bmatrix}\hat{\boldsymbol{U}}^\top \tag{26}$$

$$= \hat{\boldsymbol{U}}\operatorname{diag}\left(\boldsymbol{\sigma}\left(\boldsymbol{W_Y}\right)\right)\hat{\boldsymbol{U}}^\top \in \mathbb{R}^{m\times m},$$

where $\hat{\boldsymbol{U}} = \boldsymbol{UJ}$, with $\boldsymbol{J}$ denoting the exchange matrix (row-reversed identity matrix). We note that the vector $\boldsymbol{\sigma}\left(\boldsymbol{W_Y}\right)\in\mathbb{R}_+^r$, similarly to $\boldsymbol{\sigma}\left(\boldsymbol{X}\right)$ and $\boldsymbol{\sigma}\left(\boldsymbol{Y}\right)$, holds the singular values of $\boldsymbol{W_Y}$ in decreasing order, given that

$$\boldsymbol{w}_{i+1}\left(\boldsymbol{\sigma}_{i+1}^2\left(\boldsymbol{Y}\right)+\gamma\right)^{\frac{p-2}{2}} \geq \boldsymbol{w}_i\left(\boldsymbol{\sigma}_i^2\left(\boldsymbol{Y}\right)+\gamma\right)^{\frac{p-2}{2}} \quad \forall i. \tag{27}$$

This is true because according to the definition of $\boldsymbol{w}$, it holds $\boldsymbol{w}_{i+1} \geq \boldsymbol{w}_i \, \forall i$, while it also holds $\left(\boldsymbol{\sigma}_{i+1}^2\left(\boldsymbol{Y}\right) + \gamma\right)^{\frac{p-2}{2}} \geq \left(\boldsymbol{\sigma}_i^2\left(\boldsymbol{Y}\right) + \gamma\right)^{\frac{p-2}{2}} \, \forall i$, since $\boldsymbol{\sigma}_{i+1}\left(\boldsymbol{Y}\right) \leq \boldsymbol{\sigma}_i\left(\boldsymbol{Y}\right)$ and $\frac{p-2}{2} \leq 0$.

Finally, given that both $\boldsymbol{W_Y}$ and $\boldsymbol{XX}^{\mathsf{T}}$ are positive semidefinite symmetric matrices, we can invoke Ruhe's trace inequality from Theorem 1 and combine it with Eq. (25) to get:

$$\phi_{lr}\left(\boldsymbol{X}; \boldsymbol{w}, p\right) \leq \frac{p}{2} \operatorname{tr}\left(\boldsymbol{W_Y XX}^{\mathsf{T}}\right) + \frac{p\gamma}{2} \operatorname{tr}\left(\boldsymbol{W_Y}\right) + \frac{2-p}{2}\phi_{lr}\left(\boldsymbol{Y}; \boldsymbol{w}, p\right) \, \forall \boldsymbol{X}, \boldsymbol{Y}. \qquad (28)$$

By substitution and carrying over the algebraic operations on the r.h.s of Eq. (28), we can show that when $\boldsymbol{X} = \boldsymbol{Y}$ the inequality reduces to equality. □

### A.1.1 THEORETICAL JUSTIFICATION FOR USING AN AUGMENTED MAJORIZER

In Section 2.2 where we consider the solution of the normal equations in Eq. (11), instead of the majorizers that we derived in Eqs. (10), $\mathcal{Q}_{reg}$, we consider their augmented counterparts which are of the form:

$$\tilde{\mathcal{Q}}\left(\boldsymbol{x}; \boldsymbol{x}^k\right) = \mathcal{Q}_{reg}\left(\boldsymbol{x}; \boldsymbol{x}^k\right) + \frac{\alpha}{2}\left\|\boldsymbol{x} - \boldsymbol{x}^k\right\|_2^2. \qquad (29)$$

The reason is that, under this choice the system matrix of Eq. (11) is guaranteed to be non-singular and, thus, a unique solution of the linear system always exists. To verify that this choice doesn't compromise the convergence guarantees of our IRLS approach, we note that $\tilde{\mathcal{Q}}\left(\boldsymbol{x}; \boldsymbol{x}^k\right)$ is still a valid majorizer and satisfies both properties of Eq. (7), required by the MM framework. Specifically, it is straightforward to show that:

$$\tilde{\mathcal{Q}}\left(\boldsymbol{x}; \boldsymbol{x}\right) = \mathcal{Q}_{reg}\left(\boldsymbol{x}\right) \quad \text{and} \quad \tilde{\mathcal{Q}}\left(\boldsymbol{x}^k; \boldsymbol{x}\right) \geq \mathcal{Q}_{reg} \, \forall \boldsymbol{x}, \boldsymbol{x}^k. \qquad (30)$$

Finally, we note that the use of the augmented majorizer serves an additional purpose. In particular, due to the term $\frac{\alpha}{2}\left\|\boldsymbol{x} - \boldsymbol{x}^k\right\|_2^2$, the majorizer $\tilde{\mathcal{Q}}$ enforces the IRLS estimates between two successive IRLS iterations, $\boldsymbol{x}^k$ and $\boldsymbol{x}^{k+1}$, not to differ significantly. Both the unique solution of the linear system and the closeness of the the successive IRLS estimates play an important role for the stability of the training stage of our LIRLS networks.

### A.2 MATRIX EQUILIBRATION PRECONDITIONING

During the training and inference of LIRLS, both the network parameters as well as the samples in the input batches vary significantly. This results in a convergence behavior that is not consistent, which is mostly attributed to the varying convergence rate of the linear solver at each IRLS step. Indeed, it turns out that the main term $\boldsymbol{S}^k$ of system matrix, defined in Eq.(11), in certain cases can be poorly conditioned. To deal with this issue and improve the overall convergence of LIRLS we apply a preconditioning strategy. In particular, we employ a matrix equilibration (Duff & Koster, 2001) such that the resulting preconditioned matrix has a unit diagonal, while its off-diagonal entries are not greater than 1 in magnitude. In our case all the components that form the system matrix are given in operator form, and thus we do not have access to the individual matrix elements of $\boldsymbol{S} = \boldsymbol{S}^k + \alpha\boldsymbol{I}$. For this reason, we describe below the practical technique of forming a diagonal matrix preconditioner that equilibrates the matrix $\boldsymbol{S} \in \mathbb{R}^{n \cdot c \times n \cdot c}$.

We start by noting that such matrix can be decomposed as:

$$\boldsymbol{S} = \boldsymbol{A}^{\mathsf{T}}\boldsymbol{A} + p \cdot \sigma_n^2 \boldsymbol{G}^{\mathsf{T}}\boldsymbol{W}\boldsymbol{G} + \alpha\boldsymbol{I} = \begin{bmatrix} \boldsymbol{A}^{\mathsf{T}} & \sqrt{p} \cdot \sigma_n \boldsymbol{G}^{\mathsf{T}}\boldsymbol{W}^{1/2} & \sqrt{\alpha}\boldsymbol{I} \end{bmatrix} \begin{bmatrix} \boldsymbol{A} \\ \sqrt{p} \cdot \sigma_n \boldsymbol{W}^{1/2}\boldsymbol{G} \\ \sqrt{\alpha}\boldsymbol{I} \end{bmatrix} = \boldsymbol{B}^{\mathsf{T}}\boldsymbol{B}, \qquad (31)$$

where $\boldsymbol{B} \in \mathbb{R}^{(n_1+n_2+n_3) \times n_3}$, with $n_3 = n \cdot c$. We further note a simple fact, that any diagonal matrix $\boldsymbol{D} = \operatorname{diag}\left(\boldsymbol{d}\right)$ multiplied with $\boldsymbol{B}$ from the right $\left(\boldsymbol{BD}\right)$ equally scales all the elements of each column $\boldsymbol{B}_{:,j}$ with a corresponding diagonal element $d_j$, and the same diagonal matrix multiplied with $\boldsymbol{B}^{\mathsf{T}}$ from the left $\left(\boldsymbol{DB}^{\mathsf{T}}\right)$ scales the same way each matrix row of $\boldsymbol{B}^{\mathsf{T}}$. Let us now select the diagonal elements $d_j$ of matrix $\boldsymbol{D}$ to be of the form $d_j = 1/\|\boldsymbol{B}_{:,j}\|_2$, i.e. the inverse of the $\ell_2$ norm of the corresponding column of the matrix $\boldsymbol{B}$. In this case, the product $\boldsymbol{BD}$ results in a matrix

with normalized columns, while the product $\boldsymbol{D}\boldsymbol{B}^\mathsf{T}$ results in a matrix with normalized rows. The product of the two, *i.e.*, the preconditioned matrix $\boldsymbol{D}^\mathsf{T}\boldsymbol{B}^\mathsf{T}\boldsymbol{B}\boldsymbol{D}$, becomes equilibrated since it has a unit diagonal and all of its non-diagonal elements are smaller or equal to 1. The task then becomes to develop a simple way of calculating the vector $\boldsymbol{d}$ that holds the norms of the matrix $\boldsymbol{B}$ columns.

From the definition of matrix $\boldsymbol{B}$ in Eq (31), the squared norm of its $j$-th column can be computed as:

$$||\boldsymbol{B}_{:,j}||_2^2 = \sum_{i=1}^{n_1+n_2+n_3} \boldsymbol{B}_{i,j}^2 = \sum_{i=1}^{n_1} \boldsymbol{A}_{i,j}^2 + p \cdot \sigma_{\boldsymbol{n}}^2 \sum_{i=1}^{n_2} \left[ \boldsymbol{W}^{1/2}\boldsymbol{G} \right]_{i,j}^2 + \alpha. \tag{32}$$

All restoration problems under study are large-scale, meaning the matrices $\boldsymbol{A}$ and $\boldsymbol{G}$ and $\boldsymbol{W}$ are structured, but only available in an operator form. Depending on the problem at hand, $\boldsymbol{A}$ is either a valid convolution matrix (deblurring), strided valid convolution matrix (super-resolution), diagonal binary matrix (demosaicing) or orthonormal subsampled FFT matrix (MRI reconstruction), $\boldsymbol{G}$ is a block matrix with valid convolution matrices as blocks, while the matrix $\boldsymbol{W}$ is either a diagonal one, for the case of sparse promoting priors, or a block diagonal matrix with each block being a matrix of dimensions $c \times c$, for the case of low-rank promoting priors.

We utilize the following trick in order to calculate both terms $\sum_{i=1}^{n_1} \boldsymbol{A}_{i,j}^2$ and $\sum_{i=1}^{n_2} \left[ \boldsymbol{W}^{1/2}\boldsymbol{G} \right]_{i,j}^2$ appearing in Eq. (32). In particular, considering any arbitrary matrix $\boldsymbol{C}$ and a vector of ones $\boldsymbol{1}$ we note that $\sum_{i=1}^{n} \boldsymbol{C}_{i,j}^2 = \left( \left[ \boldsymbol{C}^{\circ 2} \right]^\mathsf{T} \boldsymbol{1} \right)_j$, where $\boldsymbol{C}^{\circ 2} \equiv \boldsymbol{C} \circ \boldsymbol{C}$ is the Hadamard square operation and $\circ$ denotes the Hadamard product. The computation of the Haramard square for operators $\boldsymbol{A}$ is straightforward: for possibly strided valid convolution matrices (the following also holds for convolutions with periodic and zero boundaries) used in deblurring and super-resolution problems the Hadamard square operator can be obtained by squaring element-wise all the elements of the convolution kernel, for the demosaicing problem we have $\boldsymbol{A}^{\circ 2} = \boldsymbol{A}$, and for the MRI reconstruction with subsampled orthonormal FFT matrix, we compute the squared norms of the columns directly, as it is equal to the corresponding acceleration (sampling) rate. Since the matrix $\boldsymbol{W}$ has a the specific structure described above, and the convolution filter bank operator $\boldsymbol{G}$ is applied independently for each color channel, it is straightforward to show, that for both sparse and low-rank cases the following holds:

$$\sum_{i=1}^{n_2} \left[ \boldsymbol{W}^{1/2}\boldsymbol{G} \right]_{i,j}^2 = \left( \left[ \left( \boldsymbol{W}^{1/2}\boldsymbol{G} \right)^{\circ 2} \right]^\mathsf{T} \boldsymbol{1} \right)_j = \left( \boldsymbol{G}^{\circ 2} \left( \boldsymbol{W}^{1/2} \right)^{\circ 2} \boldsymbol{1} \right)_j . \tag{33}$$

As already discussed above, $\boldsymbol{G}^{\circ 2}$ can be obtained by squaring element-wise all the elements of the convolution filter bank. The computation of $\left( \boldsymbol{W}^{1/2} \right)^{\circ 2}$ is trivial for the sparse case where $\boldsymbol{W}$ is a diagonal matrix for which $\left( \boldsymbol{W}^{1/2} \right)^{\circ 2} = \boldsymbol{W}$. For the low-rank case we construct $\boldsymbol{W}$ from its eigendecomposition, so all of its eigenvalues and eigenvectors are at hand, meaning that we can easily obtain $\boldsymbol{W}^{1/2}$ by computing the square root of eigenvalues of $\boldsymbol{W}$ and composing them with its eigenvectors. Then, $\left( \boldsymbol{W}^{1/2} \right)^{\circ 2}$ is computed easily by squaring element-wise all of the elements of $\boldsymbol{W}^{1/2}$.

## A.3 DISCUSSION ON THE PERFORMANCE OF THE LIRLS MODELS

In Section 4.2 we reported the reconstruction results that different LIRLS models have achieved for the studied reconstruction tasks both for grayscale/single-channel and color images. From these results we observe that for different recovery problems, the best performance is not always achieved by the same model. In particular, for the grayscale reconstruction tasks we can see that the $\ell_1^{\boldsymbol{w}}$ and $\ell_p^{p,\boldsymbol{w}}$ LIRLS models perform better on average than the $\ell_1$ and $\ell_p^p$ models. This is a strong indication that the presence of the learned weights $\boldsymbol{w}$ lead to more powerful sparsity-promoting regularizers and have a positive impact on the reconstruction quality. Moreover, we also observe that the best performance among all the models is accomplished by the $\ell_1^{\boldsymbol{w}}$, which can be somehow counter-intuitive since in theory the choice of $p < 1$ should promote sparse solutions better. A possible explanation for this is that the $\ell_1^{\boldsymbol{w}}$ regularizer is a convex one and, thus, is amenable to efficient minimization and LIRLS will converge to the global minimum of the objective function $\mathcal{J}(\boldsymbol{x})$ in

Eq. (2). On the other hand, the $\ell_p^{p,\boldsymbol{w}}$ regularizer is non-convex, which means that the stationary point reached by LIRLS can be sensitive to the initialization and might be far from the global minimum.

Regarding the color reconstruction tasks and the learned low-rank promoting regularizers that we have considered, we observe that the best performance on average is achieved by the $\mathcal{S}_p^p$ and $\mathcal{S}_p^{p,\boldsymbol{w}}$ LIRLS models with $p < 1$. To better interpret this results, we need to have in mind that the only convex regularizer out of this family is the $\mathcal{S}_1$ (nuclear norm). Given that the nuclear norm is not as expressive as the rest of the low-rank promoting regularizers, it is expected that this LIRLS model will be the least performing. Also note that the weighted nuclear norm, $\mathcal{S}_1^{\boldsymbol{w}}$, unlike to the $\ell_1^{\boldsymbol{w}}$, is non-convex and thus it does not benefit from the convex optimization guarantees. Based on the above, and having in mind the results we report in Section 4.2, it turns out that in this case the choice of a $p < 1$ plays a more important role in the reconstruction quality than the learned weights $\boldsymbol{w}$.

Another important issue that is worth of discussing, is the fact that in this work we have applied sparsity-promoting regularization on grayscale reconstruction tasks and low-rank promoting regularization on color recovery tasks. We note that by no means this is a strict requirement and it is possible to seek for low-rank solutions in some transform domain when dealing with grayscale images as in Lefkimmiatis et al. (2012; 2013; 2015), or sparse solutions when dealing with color images. Due to space limitations we have not explored this cases in this work, but we plan to include related results in an extended version of this paper.

---

**Algorithm 1:** Forward and backward passes of LIRLS networks.

---

**Inputs:** $\boldsymbol{x}_0$: initial solution, $\boldsymbol{y}$: degraded image, $\boldsymbol{A}$: degradation operator
**Input parameters:** $\boldsymbol{\theta} = \{\boldsymbol{G}, \boldsymbol{w}, p\}$: network parameters, $\sigma_{\boldsymbol{n}}^2, \alpha, \gamma$
**Forward Pass**

    Initialize: $k = 0$;

    **repeat**

        1. Compute the feature maps $\left\{\boldsymbol{z}_i^k = \boldsymbol{G}_i \boldsymbol{x}^k\right\}_{i=1}^{\ell}$, $\left(\boldsymbol{Z}_i^k = \text{vec}\left(\boldsymbol{z}_i^k\right) \text{for the low-rank case}\right)$.

        2. Compute the updated $\boldsymbol{W}_i^k$ weight matrices based on the current estimate $\boldsymbol{x}^k$:

            • Sparse case: $\boldsymbol{W}_i^k = \text{diag}\left(\boldsymbol{w}_i\right)\left[\boldsymbol{I} \circ \left(\boldsymbol{z}_i^k \boldsymbol{z}_i^{k\mathsf{T}} + \gamma \boldsymbol{I}\right)\right]^{\frac{p-2}{2}}$.

            • Low-rank case: $\boldsymbol{W}_i^k = \boldsymbol{I}_q \otimes \left[\boldsymbol{U}_i^k \text{diag}\left(\boldsymbol{w}_i\right) \boldsymbol{U}_i^{k\mathsf{T}} \left(\boldsymbol{Z}_i^k \boldsymbol{Z}_i^{k\mathsf{T}} + \gamma \boldsymbol{I}\right)^{\frac{p-2}{2}}\right]$, where

            $\boldsymbol{Z}_i^k = \boldsymbol{U}_i^k \text{diag}\left(\boldsymbol{\sigma}(\boldsymbol{Z}_i^k)\right) \boldsymbol{V}_i^{k\mathsf{T}}$.

        3. Find the updated solution $\boldsymbol{x}^{k+1}$ by solving the linear system:

$$\boldsymbol{x}^{k+1} = \left(\boldsymbol{A}^{\mathsf{T}}\boldsymbol{A} + p \cdot \sigma_{\boldsymbol{n}}^2 \sum_{i=1}^{\ell} \boldsymbol{G}_i^{\mathsf{T}} \boldsymbol{W}_i^k \boldsymbol{G}_i + \alpha \boldsymbol{I}\right)^{-1} \left(\boldsymbol{A}^{\mathsf{T}}\boldsymbol{y} + \alpha \boldsymbol{x}^k\right).$$

        4. $k = k + 1$.

    **until** *the convergence criterion is satisfied*;

    Return $\boldsymbol{x}^* = \boldsymbol{x}^k$;

**Backward Pass**

        1. Use $\boldsymbol{x}^*$ to compute $\boldsymbol{W}_i^* = \boldsymbol{W}_i^*\left(\boldsymbol{G}, \boldsymbol{x}^*\right)$ following steps 1 and 2 in the **Forward Pass**. Then use both to define the following auxiliary network with parameters $\boldsymbol{\theta}$:

$$\boldsymbol{g}\left(\boldsymbol{x}^*, \boldsymbol{\theta}\right) = \left(\boldsymbol{A}^{\mathsf{T}}\boldsymbol{A} + p \cdot \sigma_{\boldsymbol{n}}^2 \sum_{i=1}^{\ell} \boldsymbol{G}_i^{\mathsf{T}} \boldsymbol{W}_i^*\left(\boldsymbol{G}, \boldsymbol{x}^*\right) \boldsymbol{G}_i\right) \boldsymbol{x}^* - \boldsymbol{A}^{\mathsf{T}}\boldsymbol{y}.$$

        2. Compute $\boldsymbol{v} = \left(\nabla_{\boldsymbol{x}^*} \boldsymbol{g}\right)^{-1} \boldsymbol{\rho}$ by solving the linear system $\nabla_{\boldsymbol{x}^*} \boldsymbol{g} \cdot \boldsymbol{v} = \boldsymbol{\rho}$, where $\boldsymbol{\rho} = \nabla_{\boldsymbol{x}^*} \mathcal{L}$ and $\mathcal{L}$ is the training loss function.

        3. Obtain the gradient $\nabla_{\boldsymbol{\theta}} \mathcal{L}$ by computing the product $\nabla_{\boldsymbol{\theta}} \boldsymbol{g} \cdot \boldsymbol{v}$.

        4. Use $\nabla_{\boldsymbol{\theta}} \mathcal{L}$ to update the network's parameters $\boldsymbol{\theta}$ or backpropagate further into their parent leafs.

---

## A.4 Algorithmic implementation of LIRLS networks

In Algorithm 1 we provide the pseudo-code for the forward and backward passes of the LIRLS models, where we distinguish between the learned low-rank and sparsity promoting scenarios. The gradients in the backward pass can be easily computed using any of the existing autograd libraries.

## A.5 Empirical convergence of LIRLS to a fixed point

As we have explained in the manuscript, relying on Lemmas 1 and 2 we have managed to find valid quadratic majorizers for both the sparsity- and low-rank promoting regularizers defined in Eq. (6). Given that these majorizers satisfy all the necessary conditions required by the MM framework, we can safely conclude that the proposed IRLS strategy will converge to a fixed point. In this section we provide further empirical evidence which support our theoretical justification. In particular, we have conducted several evaluations of the trained LIRLS models. In the first scenario we run LIRLS models for 30 steps for color deblurring and simulated MRI with x4 acceleration on the corresponding datasets described in Subsection 4.1. After that, we calculate the mean PSNR and SSIM scores individually for each step across all images in each dataset. We provide the resulting plots in Fig. 6. As we can notice, after approximately 25 iterations, both PSNR and SSIM curves for all different learned regularizers have reached an equillibrium and their values do not change. For comparison reasons we also plot the evolution of PSNR and SSIM for standard TV-based regularizers that exist in the literature. These results provide a strong indication that our LIRLS models indeed reach a fixed point, which is well aligned with the theory.

Additionally to the previous averaged convergence results, we provide some representative examples of convergence to a fixed point per individual images and models. For this reason we have selected images from grayscale and color super-resolution benchmarks and provide the inference results of $\ell_p^{p,\boldsymbol{w}}$ and $\mathcal{S}_p^{p,\boldsymbol{w}}$, respectively in Figs.7,8. The plots in the top row of each figure depict the evolution of the relative tolerance rtol $= \frac{||\boldsymbol{x}^k - \boldsymbol{x}^{k-1}||}{\boldsymbol{x}^k}$, the value of the objective function $\mathcal{J}(\boldsymbol{x}) - $ const shifted by a constant value, and the PSNR score across the number of the performed IRLS iterations. The middle rows show the estimated solutions at specific steps, while the bottom rows include the corresponding relative error, i.e. the difference between the current latent estimate and the one from the previous step. The corresponding PSNR values and relative errors are provided for each image. For visualization purposes the images with the difference are normalized by the maximum relative error. From these figures it is clearly evident that the relative error between the current estimate and the one from the previous IRLS iteration gradually decreases and approaches zero. At the same time the value of the objective function approaches a stationary point, and the PSNR value starts saturating at the later iterations. Please note that in Fig. 8 we include results obtained by employing more than the 15 IRLS iterations that we used for the comparisons reported in Sec. 4.2. The reason for this, is that our main purpose here is to experimentally demonstrate that our $S_p^{p,\boldsymbol{w}}$ LIRLS model indeed converges to a fixed point and not to increase the computational efficiency of the model.

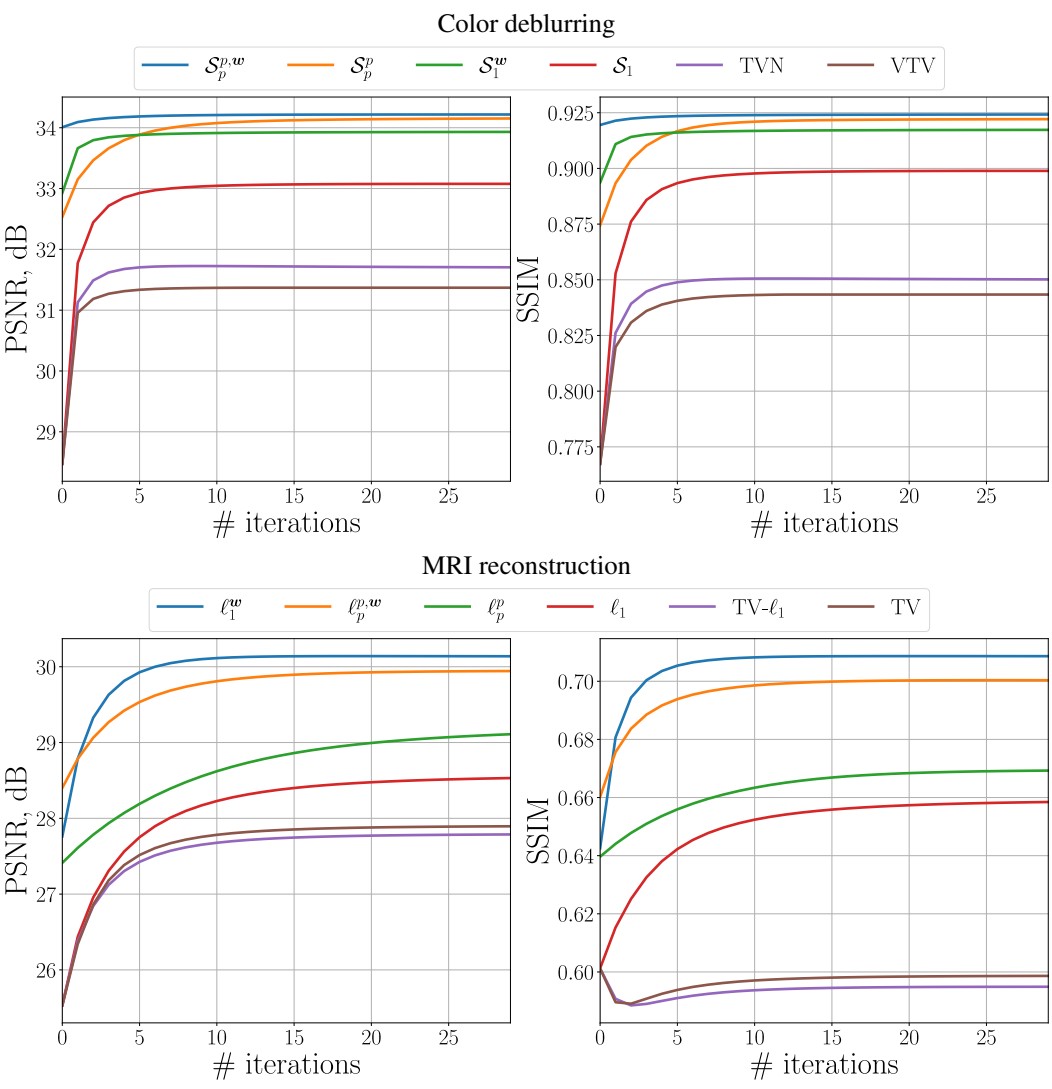

Figure 6: Convergence of LIRLS models to a fixed point. Top: color deblurring Sun et al. (2013) dataset with 1% noise, bottom: simulated MRI with x4 acceleration and 1% noise benchmark based on Knoll et al. (2020) and described in Subsection 4.1.

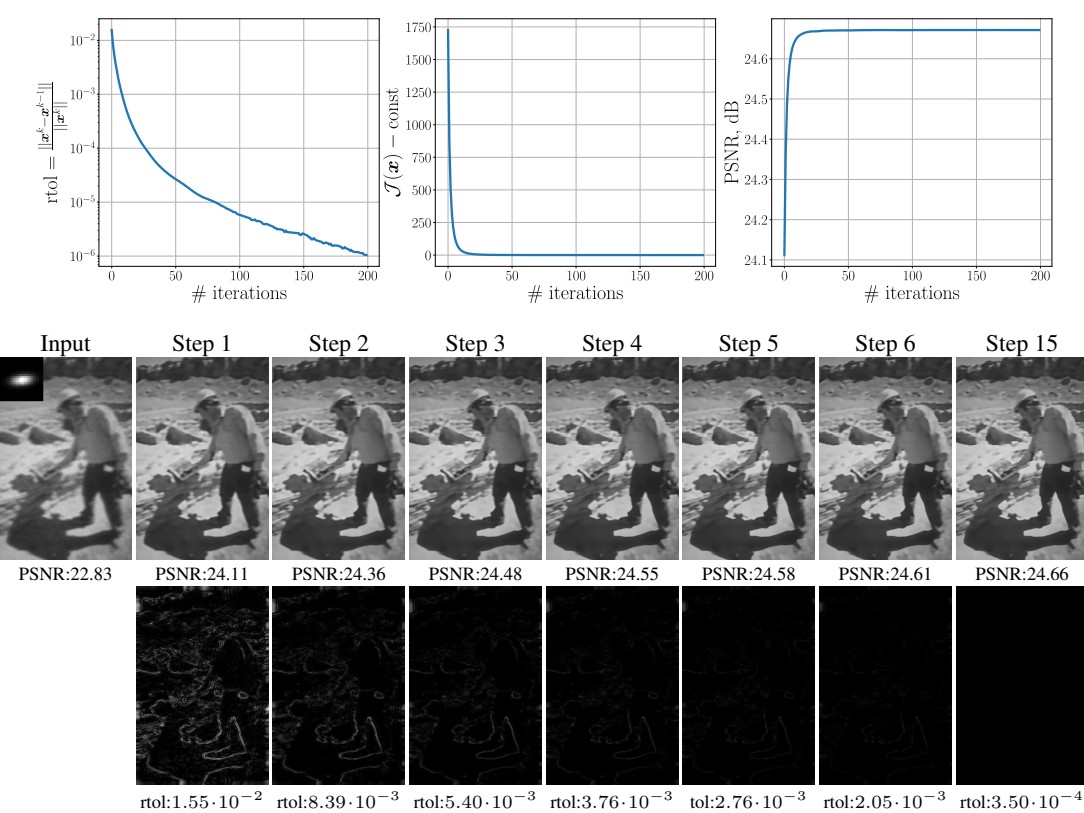

Figure 7: Demonstration of convergence to a fixed point of the the $\ell_p^{p,\boldsymbol{w}}$ LIRLS model for the task of grayscale super-resolution. The input corresponds to a synthetically downscaled image by a scale factor of 3 with 1% noise. The image is taken from the BSD100RK dataset.

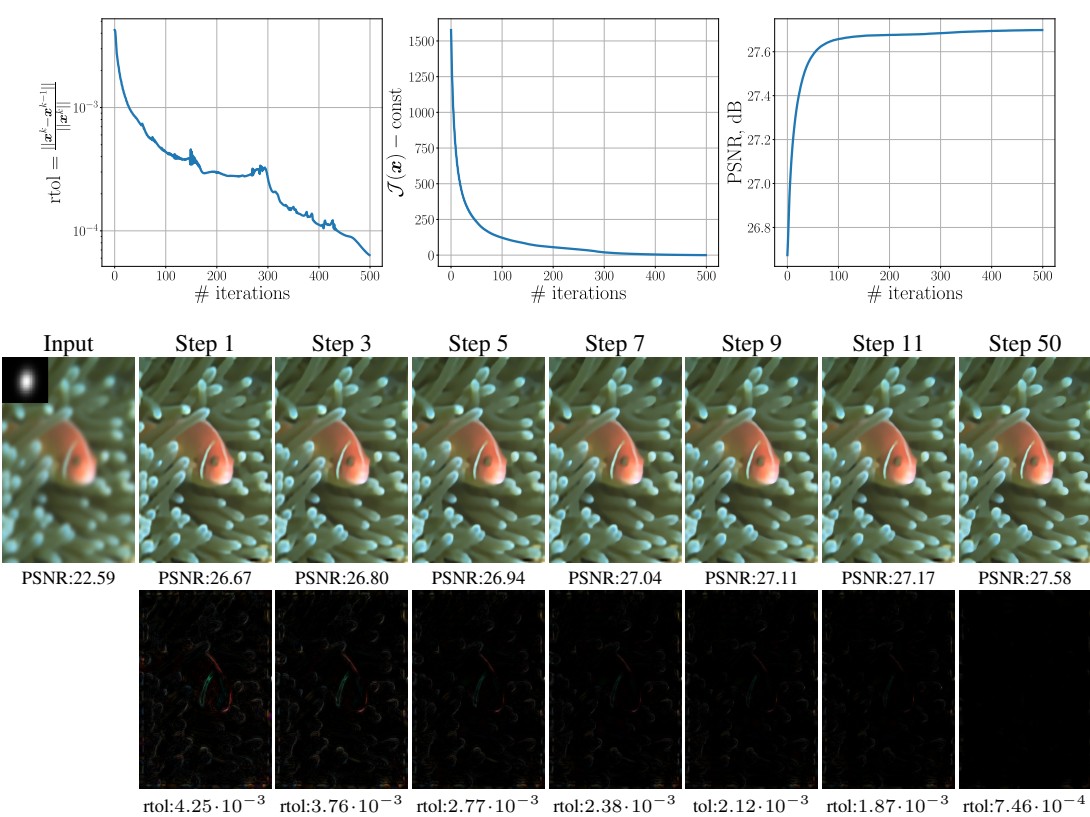

Figure 8: Demonstration of convergence to a fixed point of the the $\mathcal{S}_p^{p,\boldsymbol{w}}$ LIRLS model for the task of color super-resolution. The input corresponds to a synthetically downscaled image by a scale factor of 4 without noise. The image is taken from the BSD100RK dataset.

## A.6 Weight Prediction Networks Architectures

In the main manuscript we have specified a neural network, whose role is to predict the weights $\boldsymbol{w}$ from some initial solution $\boldsymbol{x}_0$, that will then be used in the weighted $\ell_p^p$ and $\mathcal{S}_p^p$ norms during the optimization stage (see Fig. 1). This weight prediction network is chosen to be either a lightweight RFDN architecture proposed by Liu et al. (2020) for the deblurring, super-resolution and demosaicing problems, or a lightweight UNet from Ronneberger et al. (2015) for MRI reconstruction. Below in Table 7 and Table 8 we present a detailed per-layer structure of both networks that we have used in our reported experiments.

Table 7: Detailed per-layer structure of the RFDN Liu et al. (2020) weight prediction network (WPN) that was used to predict the weights $\boldsymbol{w}$ of the weighted $\ell_p^p$ ($\ell_1^{\boldsymbol{w}}$, $\ell_p^{p,\boldsymbol{w}}$) and weighted $\mathcal{S}_p^p$ ($\mathcal{S}_1^{\boldsymbol{w}}$, $\mathcal{S}_p^{p,\boldsymbol{w}}$) norms for the deblurring, super-resolution and demosaicing problems. Here "conv" denotes a convolution layer, "relu" denotes a rectified linear unit function (ReLU), "lrelu" denotes a leaky ReLU function with negative slope of 0.05, "sigmoid" denotes a sigmoid function, "sc" denotes a skip-connection, "cat" denotes a concatenation along channels dimension, "maxpool" denotes a max-pooling operation, "interp" denotes a bilinear interpolation, "mul" denotes a point-wise multiplication. For sparsity promoting priors the number of input and output channels is 25 and 24 respectively, while for low-rank promoting priors it is 4 and 3, respectively. Blocks with repeated structures (but not shared weights) are denoted with —"—. For a more detailed architecture description we refer to the codes released by the authors of Liu et al. (2020), which we have used without any modifications: `https://github.com/njulj/RFDN`.

| Block | Layer | Kernel Size | Stride | Padding | Input Channels | Output Channels | Bias |
|---|---|---|---|---|---|---|---|
| | conv | $3 \times 3$ | $1 \times 1$ | $1 \times 1$ | 25/4 | 40 | True |
| Residual Feature Distillation Block | conv+lrelu | $3 \times 3$ | $1 \times 1$ | | 40 | 20 | True |
| | conv+sc+lrelu | $3 \times 3$ | $1 \times 1$ | $1 \times 1$ | 40 | 40 | True |
| | conv+lrelu | $3 \times 3$ | $1 \times 1$ | | 40 | 20 | True |
| | conv+sc+lrelu | $3 \times 3$ | $1 \times 1$ | $1 \times 1$ | 40 | 40 | True |
| | conv+lrelu | $3 \times 3$ | $1 \times 1$ | | 40 | 20 | True |
| | conv+sc+lrelu | $3 \times 3$ | $1 \times 1$ | $1 \times 1$ | 40 | 40 | True |
| | conv+lrelu | $3 \times 3$ | $1 \times 1$ | $1 \times 1$ | 40 | 20 | True |
| | cat+conv | $3 \times 3$ | $1 \times 1$ | | 80 | 40 | True |
| | conv | $1 \times 1$ | $1 \times 1$ | | 40 | 10 | True |
| | conv | $3 \times 3$ | $2 \times 2$ | | 10 | 10 | True |
| | maxpool | $7 \times 7$ | $3 \times 3$ | | | | |
| | conv+relu | $3 \times 3$ | $1 \times 1$ | $1 \times 1$ | 10 | 10 | True |
| | conv+relu | $3 \times 3$ | $1 \times 1$ | $1 \times 1$ | 10 | 10 | True |
| | conv+interp | $3 \times 3$ | $1 \times 1$ | $1 \times 1$ | 10 | 10 | True |
| | conv+sc | $1 \times 1$ | $1 \times 1$ | | 10 | 10 | True |
| | conv+sigmoid+mul | $1 \times 1$ | $1 \times 1$ | | 10 | 40 | True |
| Residual Feature Distillation Block | —"— | | | | | | |
| Residual Feature Distillation Block | —"— | | | | | | |
| Residual Feature Distillation Block | —"— | | | | | | |
| | cat+conv+lrelu | $1 \times 1$ | $1 \times 1$ | | 160 | 40 | True |
| | conv+sc | $3 \times 3$ | $1 \times 1$ | $1 \times 1$ | 40 | 40 | True |
| | conv | $3 \times 3$ | $1 \times 1$ | $1 \times 1$ | 40 | 24/3 | True |

Table 8: Detailed per-layer structure of the U-Net Ronneberger et al. (2015) weight prediction network (WPN) that was used to predict the weights $w$ of the weighted $\ell_p^p$ norm ($\ell_p^{p,w}$) for the MRI reconstruction problem. Here "conv" denotes a convolution layer, "up-conv" denotes a transpose convolution layer, "relu" denotes a rectified linear unit function (ReLU), "norm" denotes an instance normalization layer Ulyanov et al. (2016), "sc" denotes a skip-connection, "cat" denotes a concatenation along channels dimension, "maxpool" denotes a max-pooling operation.

| Block | Layer | Kernel Size | Stride | Input Channels | Output Channels | Bias |
|---|---|---|---|---|---|---|
| | conv+norm+relu | $3 \times 3$ | $1 \times 1$ | 25 | 25 | True |
| | conv+norm+relu | $3 \times 3$ | $1 \times 1$ | 25 | 25 | True |
| Down | maxpool | $2 \times 2$ | $2 \times 2$ | | | |
| | conv+norm+relu | $3 \times 3$ | $1 \times 1$ | 25 | 32 | True |
| | conv+norm+relu | $3 \times 3$ | $1 \times 1$ | 32 | 32 | True |
| Down | maxpool | $2 \times 2$ | $2 \times 2$ | | | |
| | conv+norm+relu | $3 \times 3$ | $1 \times 1$ | 32 | 64 | True |
| | conv+norm+relu | $3 \times 3$ | $1 \times 1$ | 64 | 64 | True |
| Down | maxpool | $2 \times 2$ | $2 \times 2$ | | | |
| | conv+norm+relu | $3 \times 3$ | $1 \times 1$ | 64 | 64 | True |
| | conv+norm+relu | $3 \times 3$ | $1 \times 1$ | 64 | 64 | True |
| Up | up-conv+cat | $2 \times 2$ | $2 \times 2$ | 64 | 64 | True |
| | conv+norm+relu | $3 \times 3$ | $1 \times 1$ | 128 | 32 | True |
| | conv+norm+relu | $3 \times 3$ | $1 \times 1$ | 32 | 32 | True |
| Up | up-conv+cat | $2 \times 2$ | $2 \times 2$ | 32 | 32 | True |
| | conv+norm+relu | $3 \times 3$ | $1 \times 1$ | 64 | 25 | True |
| | conv+norm+relu | $3 \times 3$ | $1 \times 1$ | 25 | 25 | True |
| Up | up-conv+cat | $2 \times 2$ | $2 \times 2$ | 25 | 25 | True |
| | conv+norm+relu | $3 \times 3$ | $1 \times 1$ | 50 | 25 | True |
| | conv+norm+relu | $3 \times 3$ | $1 \times 1$ | 25 | 25 | True |
| | conv | $1 \times 1$ | $1 \times 1$ | 25 | 24 | True |

## A.7 VISUAL RESULTS

In this section we provide additional visual comparisons among the competing methods for all the inverse problems under study.

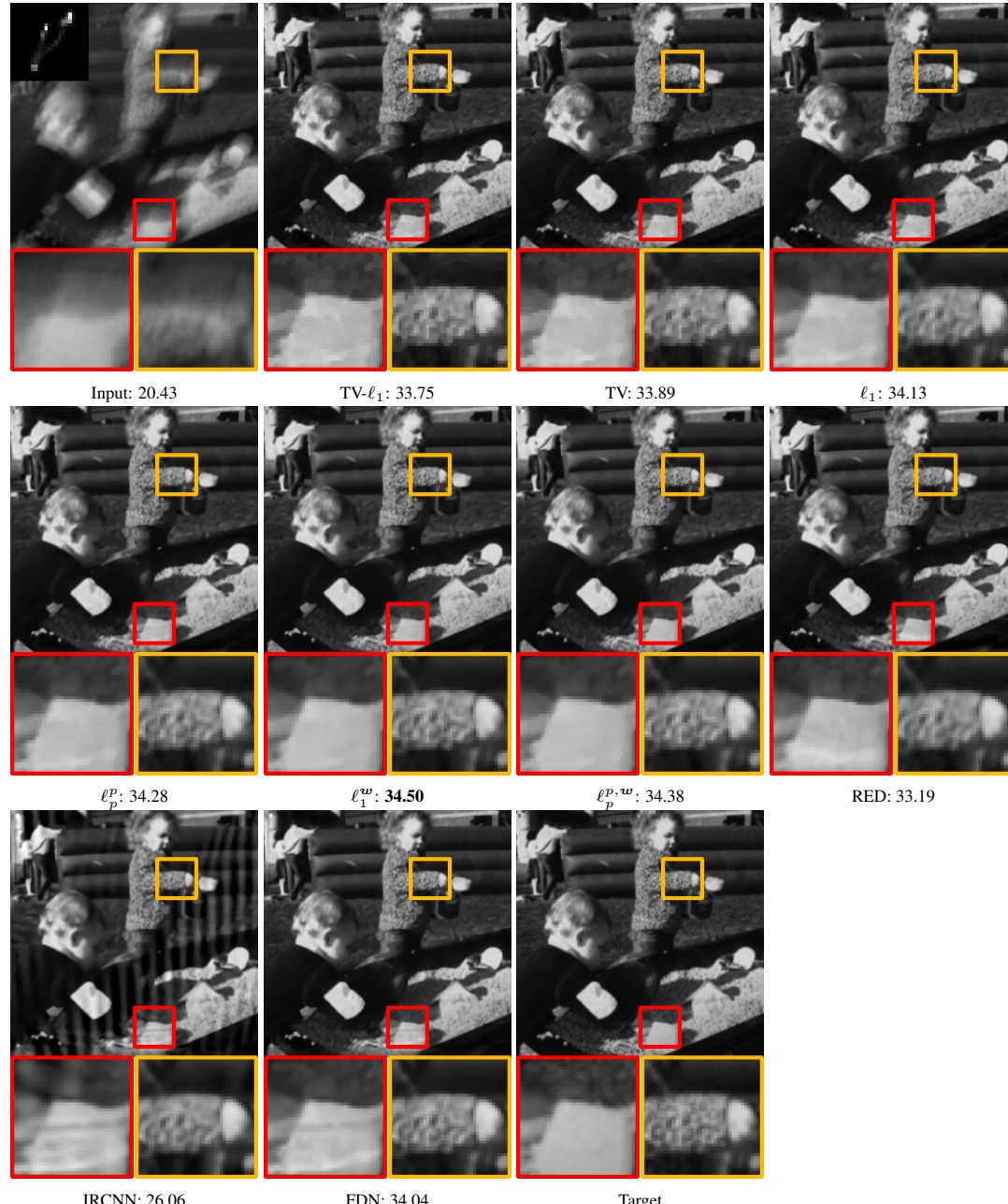

Figure 9: Visual comparisons among several methods on an optically blurred image from the Levin et al. (2009) dataset. For each reconstructed image its PSNR value is provided in dB.

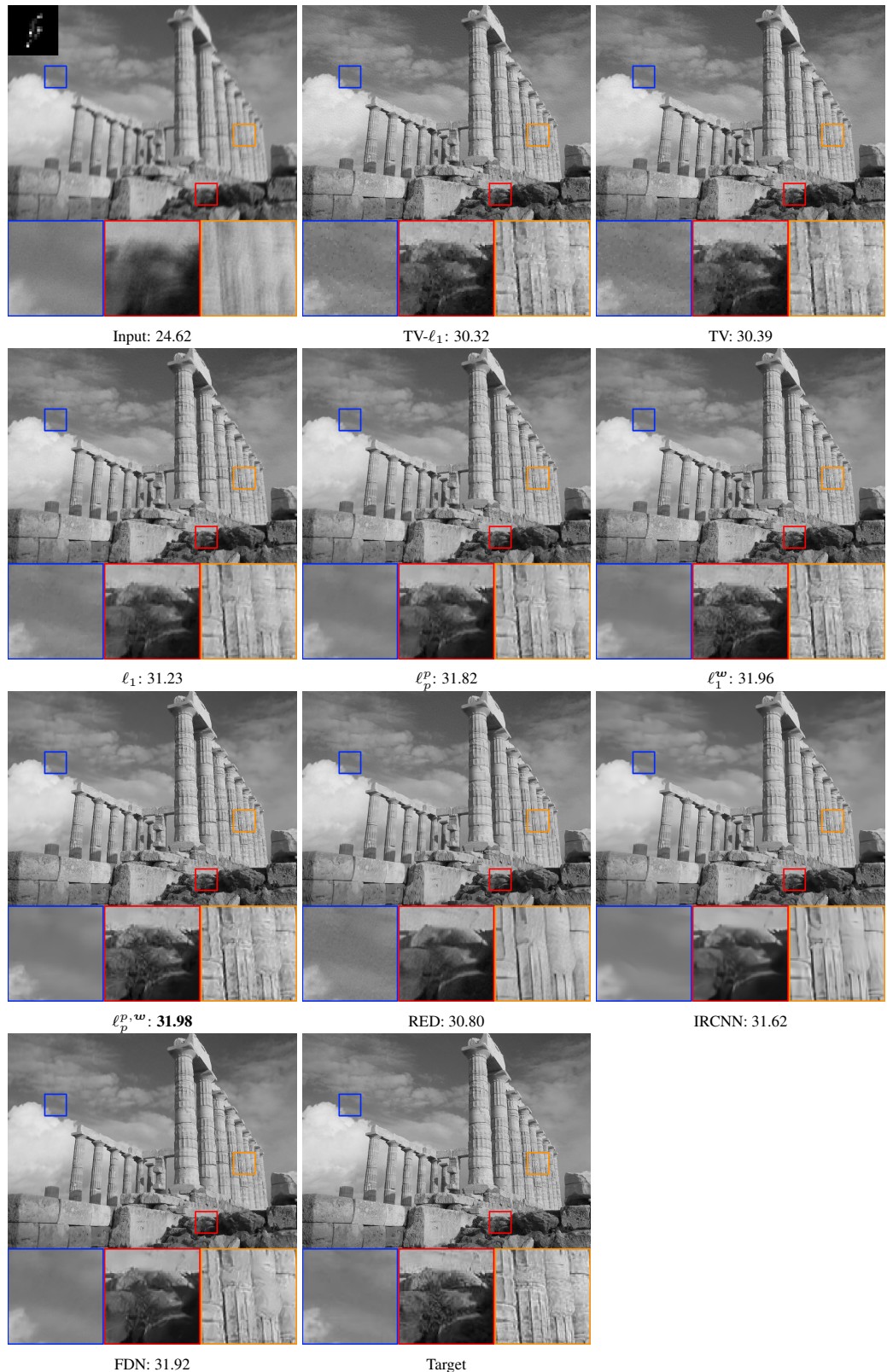

Figure 10: Visual comparisons among several methods on a synthetically blurred image from the Sun et al. (2013) dataset. For each reconstructed image its PSNR value is provided in dB.

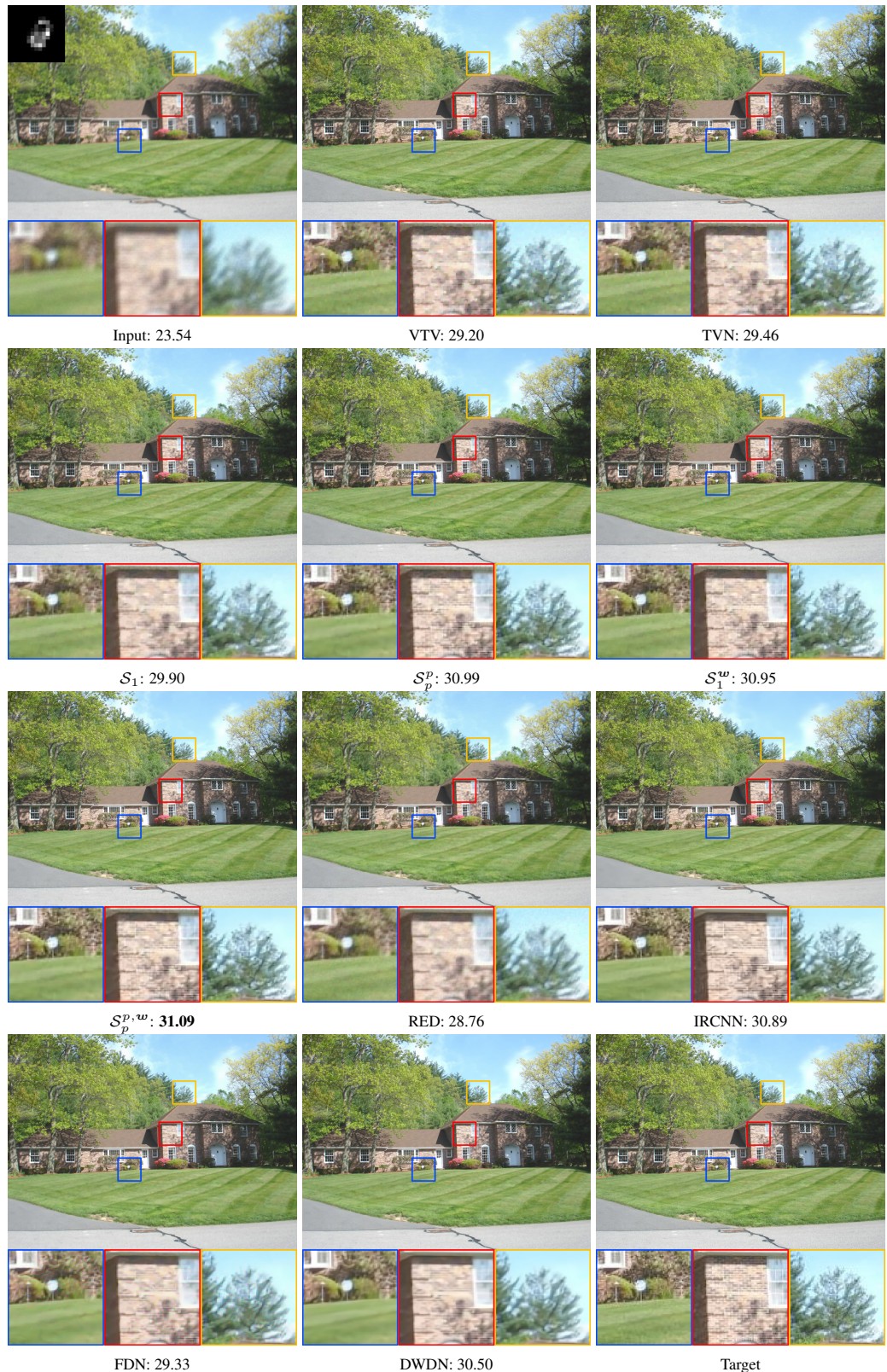

Figure 11: Visual comparisons among several methods on a synthetically blurred image from the Sun et al. (2013) dataset. For each reconstructed image its PSNR value is provided in dB.

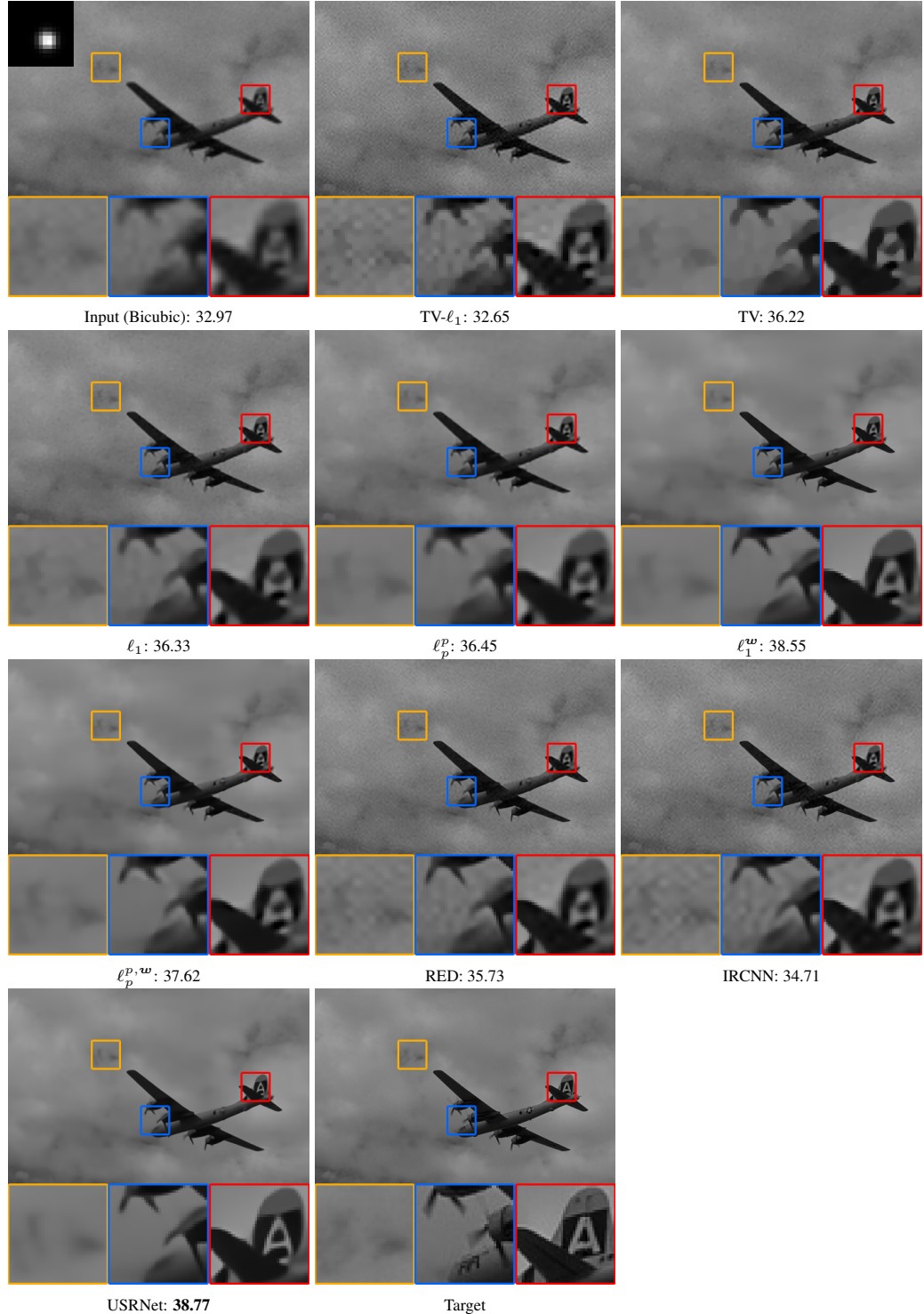

Figure 12: Visual comparisons among several methods on x3 synthetically downscaled image with 1% noise from the BSD100RK dataset. For each reconstructed image its PSNR value is provided in dB.

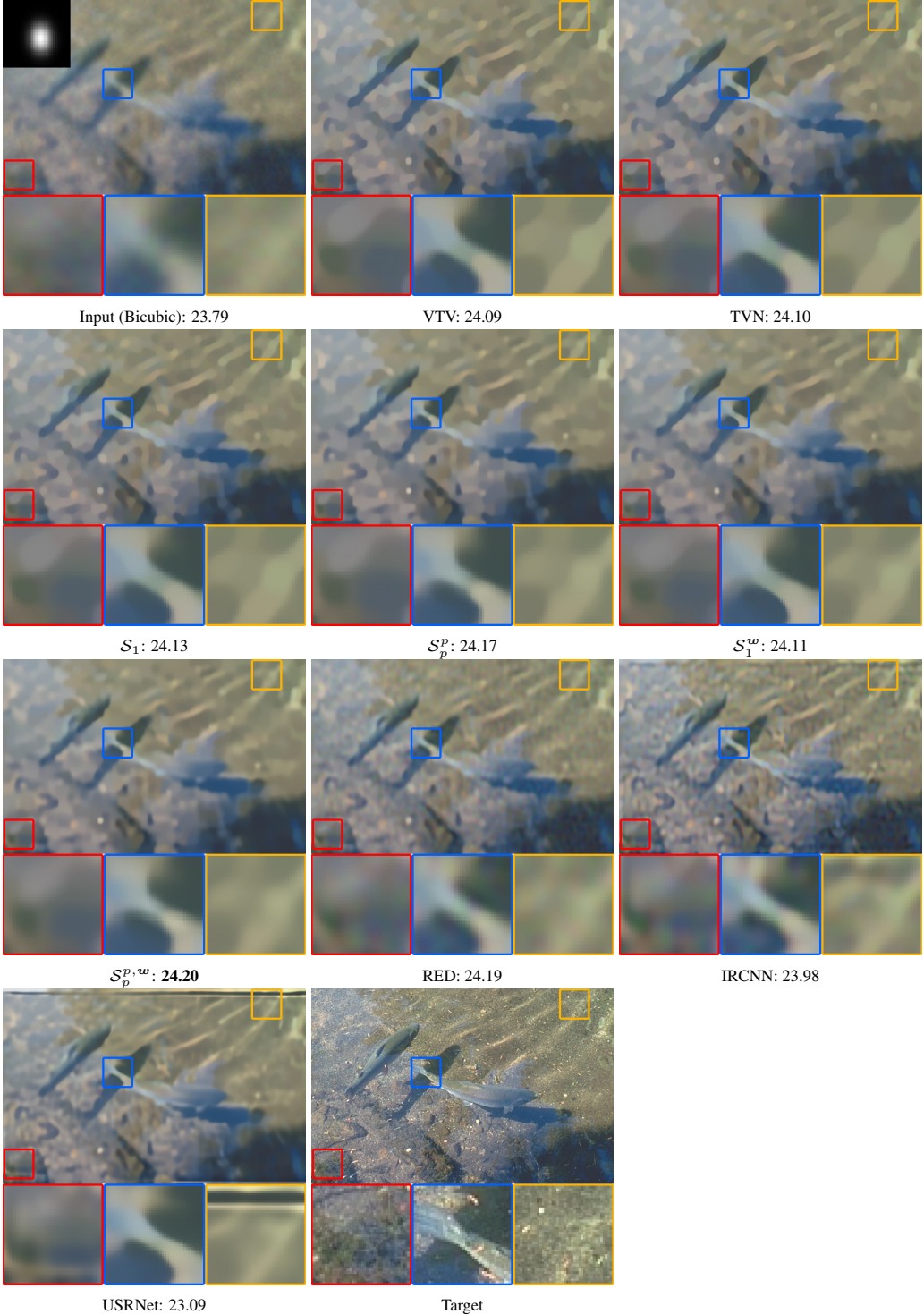

Figure 13: Visual comparisons among several methods on x4 synthetically downscaled image with 1% noise from the BSD100RK dataset. For each reconstructed image its PSNR value is provided in dB.

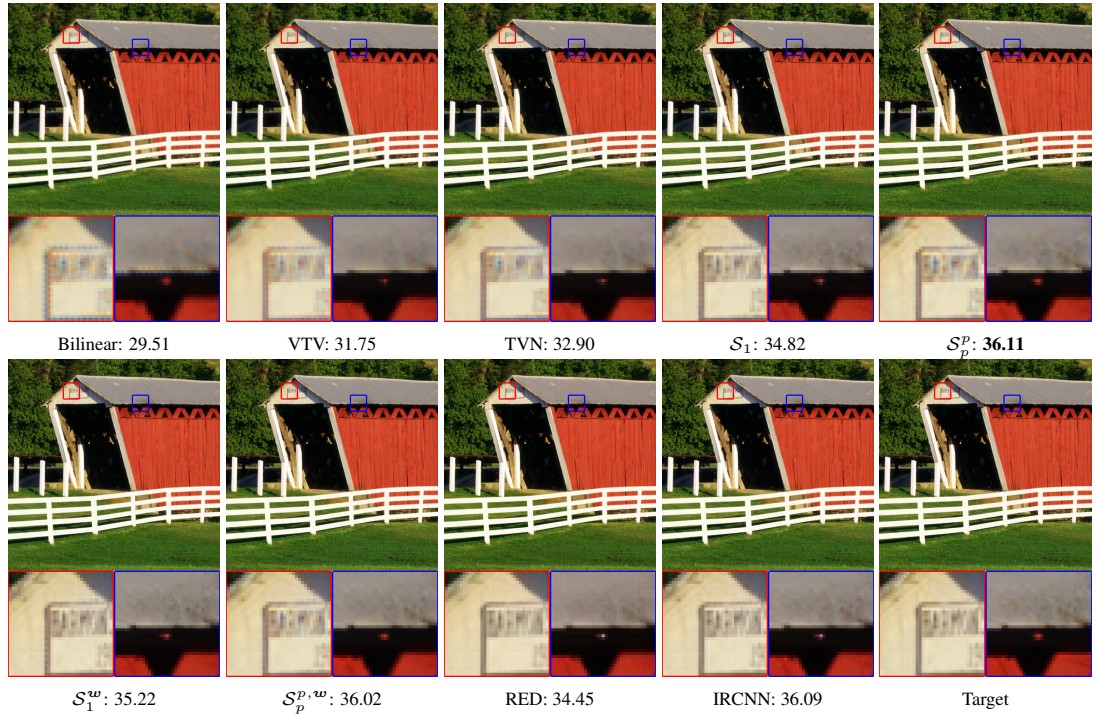

Figure 14: Visual comparisons among several methods on a mosaicked image without noise. For each demosaicked image its PSNR value is provided in dB.

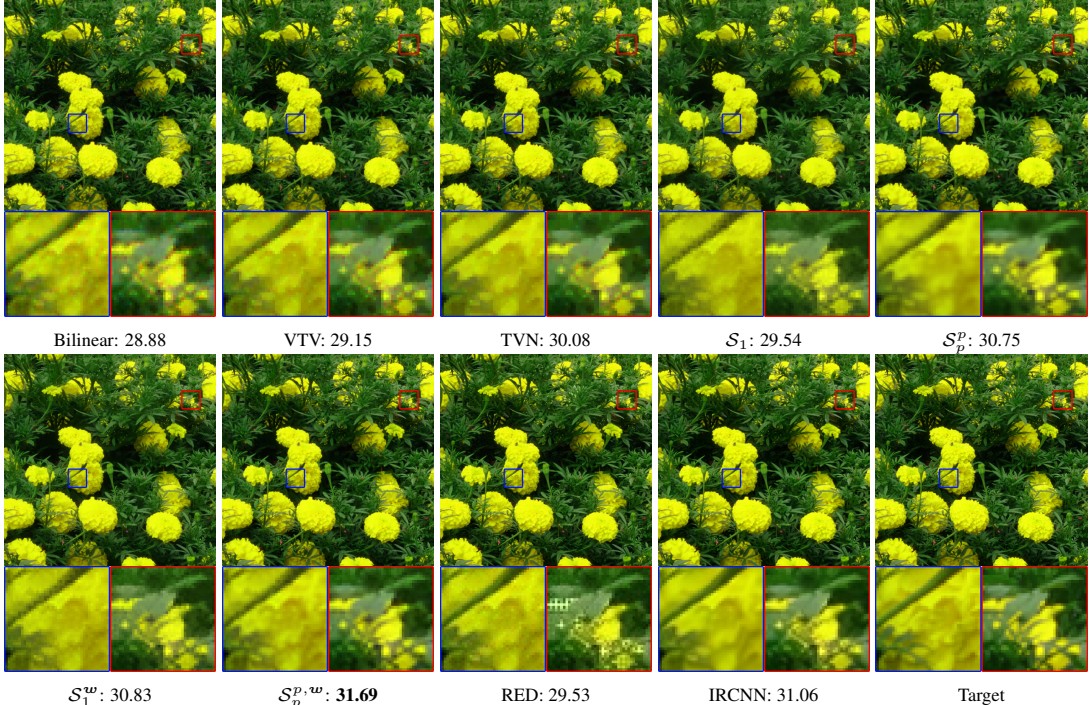

Figure 15: Visual comparisons among several methods on a mosaicked image with 3% noise. For each demosaicked image its PSNR value is provided in dB.

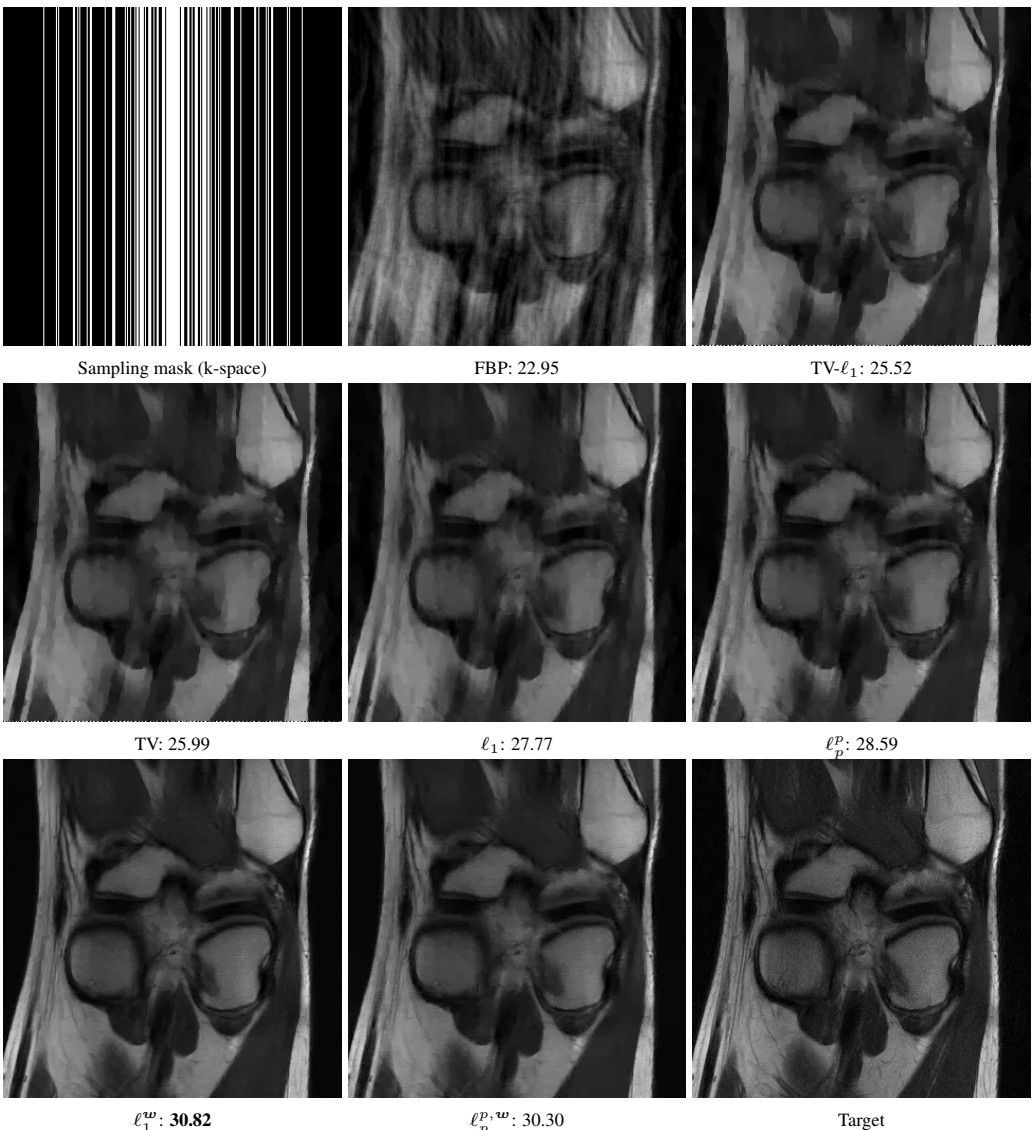

Figure 16: Visual comparisons among several methods on a simulated MRI with x4 acceleration. For each reconstructed image its PSNR value is provided in dB.

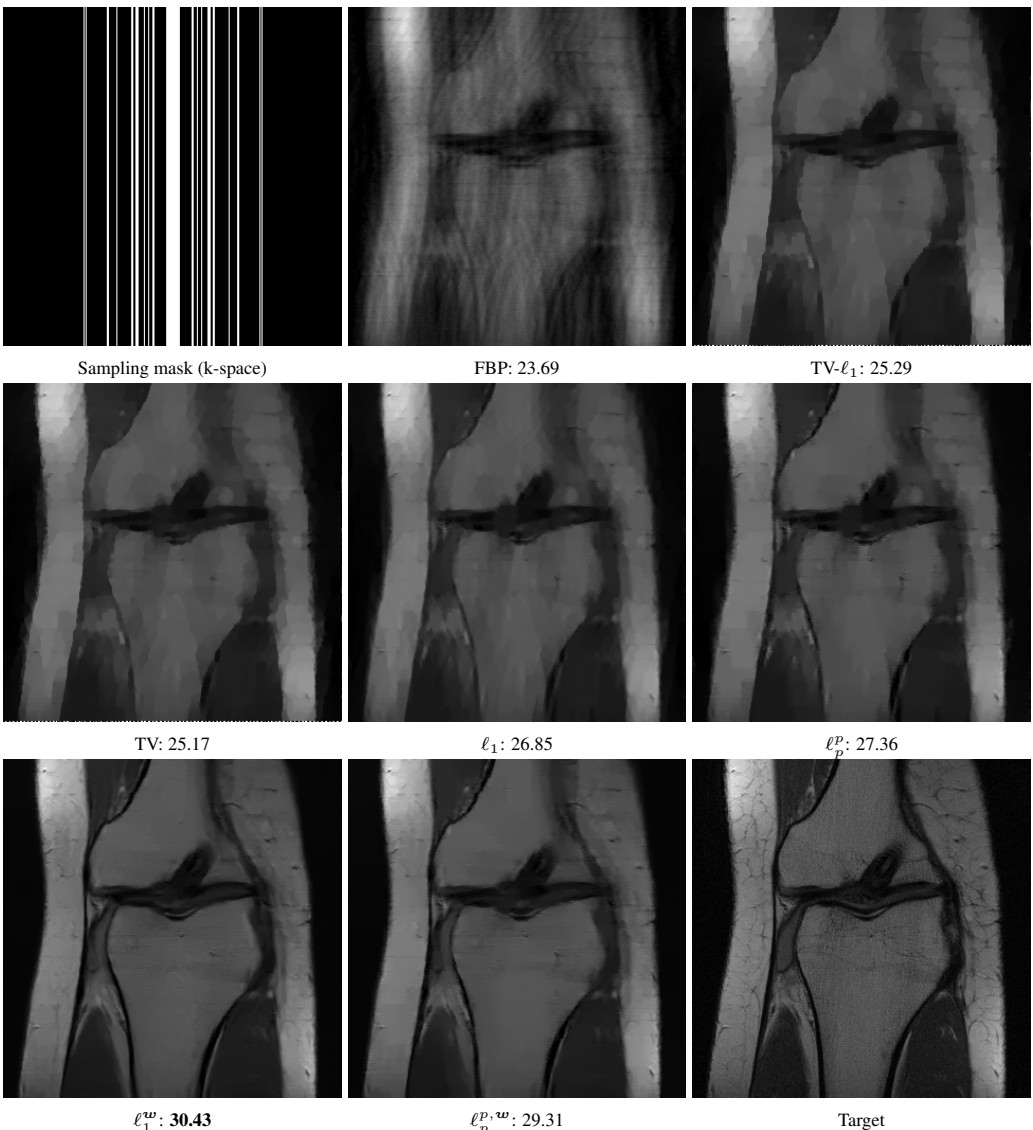

Figure 17: Visual comparisons among several methods on a simulated MRI with x8 acceleration. For each reconstructed image its PSNR value is provided in dB.

