# OpenReview forum: "Learning Sparse and Low-Rank Priors for Image Recovery via Iterative Reweighted Least Squares Minimization"
_ICLR.cc/2023/Conference — ICLR 2023 poster_

### Official Review · Reviewer_vB8U · 2022-10-17

**Confidence:** 4
**Correctness:** 4
**Technical Novelty And Significance:** 3
**Empirical Novelty And Significance:** 3
**Recommendation:** 6

**Clarity, Quality, Novelty And Reproducibility:**

# Clarity

The paper is rather clear. Only part where the neural network is introduced are a little bit confused. For example, I cannot see the number of layers used for the experiment in the text.

# Quality

This is good quality work.

# Novelty

The framework by itself is new and compares favorably against unfolding methods that have been used for such tasks (see https://arxiv.org/abs/2108.06637).

# Reproducibility

As it is some part remains unclear as the number of layers. However the authors put most of the parameters in the paper and the data sets are public, so most of the framework could be reproduced.



**Strength And Weaknesses:**

# Strength

The proposed method is a complete framework to solve inverse problem for image processing. It relies on regularizers that are well known to be effective for such problems and leads to pretty good results. The biggest asset of the framework is the construction of a layer which relies on the IRLS with learned filters for regularizations. Using in some way the best of two worlds it leads to a fairly strong neural network that is able to solve inverse problems.

# Weakness

They are several weakness in the proposed framework.
1. The image formation model (with a Gaussian additive noise) is very general, but fails on some non-additive noise like Poisson noise in low-luminosity setting. As changing the noise change the loss function, the construction of a surrogate may become delicate.
2. The IRLS layer asks for solving a optimization problem. Even if conjugate gradient is a very powerful method it takes times to apply, thus with deep network the learning process may become untrackable...
3. While the results are among the best, both quantitative and qualitative results show little improvements.
4. The results are computed for the specific level of noise. It would have been interesting to try a blind version, where the noise level is random during the training.
5. The reproduction of the methods could be difficult (see below).

**Summary Of The Paper:**

This paper proposes a new image processing method based on combination of several technics. First the authors introduce a generalization of the iterative reweigthed least square (IRLS) optimization scheme. Based on the majorization-minimization scheme, it solved a classical optimization problem build from a least square loss function with regularizers. Second, they proposed to learn the filters used in the regularization using deep learning method, for that they proposed to use the previous method as a layer of a recurrent network. The results show the effectiveness of the framework on different inverse problems in image processing.

**Summary Of The Review:**

To sum up the paper, it is good paper that introduce a  image processing method. The two main contribution are the generalization of the IRLS  with a extended set of regularizers and a new neural network layer based on the optimization framework. While some part still need to be clarified, the full method is exposed. The results are among the best, but shows minor improvement compare to state-of-art.

---

> ### Author Response · Authors · 2022-11-10
> **Response to reviewer's 1st comment**
>
> We thank the reviewer for carefully studying our manuscript and for his constructive comments.
>
> > They are several weakness in the proposed framework.
> > 1. The image formation model (with a Gaussian additive noise) is very general, but fails on some non-additive noise like Poisson noise in low-luminosity setting. As changing the noise change the loss function, the construction of a surrogate may become delicate.
>
> The reviewer correctly points out that the image formation model considered in this work assumes that the noise obeys the Gaussian distribution and this might not be an accurate model for all image acquisition scenarios. Nevertheless, we would like to point out that one of the main contributions of this work is not about the data fidelity term, which accounts for the noise statistics, but about the learnable sparsity- and low-rank promoting regularizers and their efficient training. While the problem of reconstruction under low-light conditions is out of the scope of the current work, it is indeed an interesting future research direction. For this reason we provide below some evidence based on prior work which indicate that with some proper modifications, our MM framework can accommodate for Poisson image recovery tasks.
>
> Specifically, the case of Poisson or mix Poisson-Gaussian noise can still be efficiently dealt with by using different strategies while taking advantage of the MM strategy described in this manuscript. One possible way to do so is by first applying a Variance Stabilization Transform (VST) (Anscombe 1948, Zhang et al. 2007, Makitalo & Foi 2012) on the observed data before processing them. The VST “Gaussianizes” the data so that each sample is near-normally distributed with an asymptotically constant variance. It is then possible to proceed with the reconstruction by following the strategy that we describe in our manuscript and obtain the final estimate by inverting the VST on the output of the network. We also note that  Pronina et al. 2020 applied a similar VST-based strategy in the context of microscopy deconvolution using a deep network. Their reported results are very competitive compared to those obtained by methods explicitly designed for a formation model under Poisson noise.
>
> Another way to deal with the Poisson noise statistics, is to follow a MM strategy where apart from considering a quadratic upper bound on the regularizer, as we do in this work, one can also employ a quadratic upper bound on the data fidelity term. Such an upper bound is considered by Stagliano et al. 2011 and Li et al. 2015. This leads to a tight quadratic upper bound of the overall objective function and, thus, an IRLS strategy is directly applicable. We leave this topic as a future research direction.
>
>
>
> ## References
>
> Pronina, Valeriya, et al. "Microscopy image restoration with deep wiener-kolmogorov filters." European Conference on Computer Vision. Springer, 2020.
>
> Stagliano, A., P. Boccacci, and M. Bertero. "Analysis of an approximate model for Poisson data reconstruction and a related discrepancy principle." Inverse Problems 27.12 (2011): 125003.
>
> Zhang, Bo, et al. "Multiscale variance-stabilizing transform for mixed-Poisson-Gaussian processes and its applications in bioimaging." 2007 IEEE International Conference on Image Processing. Vol. 6. IEEE, 2007.
>
> Makitalo, Markku, and Alessandro Foi. "Optimal inversion of the generalized Anscombe transformation for Poisson-Gaussian noise." IEEE transactions on image processing 22.1 (2012): 91-103.
>
> Anscombe, Francis J. "The transformation of Poisson, binomial and negative-binomial data." Biometrika 35.3/4 (1948): 246-254.
>
> Li, Jia, et al. "A reweighted $ \ell^ 2$ method for image restoration with poisson and mixed Poisson-Gaussian noise." Inverse Problems & Imaging 9.3 (2015): 875.

---

> > ### Comment · Reviewer_vB8U · 2022-11-15
> > **Response to reviewer's 1st comment**
> >
> > Thank you for your answers. I agree that VST is a solution to directly extend the model to more kind of noises, but still it is a limitation especially when working with high noise level. I also agree that a generalization is out of the scope of current paper.

---

> ### Author Response · Authors · 2022-11-10
> **Response to reviewer's 2nd and 3rd comments**
>
> > 2. The IRLS layer asks for solving a optimization problem. Even if conjugate gradient is a very powerful method it takes times to apply, thus with deep network the learning process may become untrackable...
>
> We agree with the reviewer that to obtain the output of the IRLS layer we need to solve a large-scale linear problem using an iterative solver such as CG. However we would like to emphasize that in the MM framework we don’t need to exactly solve the corresponding normal equations but it suffices to obtain a solution at the current IRLS iteration that leads to a decrease of the objective function. This can lead to a significant decrease of the necessary CG iterations. In fact, for all the reported results during training, we utilize not more than 150 CG iterations and we have observed that in many cases the maximum required number of CG iterations is significantly less. This is mainly because of the  preconditioning that we employ and our strategy of using the estimate of the previous IRLS iteration as initial solution of CG for the current IRLS step.
>
> In any case, had we followed the naïve training strategy of back propagating through the unrolled CG iterations, then the learning process would have been practically infeasible due to the relative large numbers of the required iterations. However, as we explain in detail in Section 3.1, relying on the fact that our MM strategy is guaranteed to converge to a fixed point, we eliminate completely the need of storing any intermediate results of the IRLS and CG iterations during the forward pass and we don’t have to back-propagate through the unrolled IRLS and CG iterations during the backward pass. Therefore the learning process becomes feasible but at the cost of having to solve the linear problem of Eq. (13) during the backward pass. As a result during the training of all the different LIRLS models we have not experienced any significant difficulties with the practical application of our proposed learning process.
>
> > 3. While the results are among the best, both quantitative and qualitative results show little improvements.
>
> We agree with the reviewer that the results we report are not significantly better than those of other recent SOTA deep learning networks. However, it is also important to highlight that as we already state in both the abstract and inside the manuscript, we manage to obtain equivalent or better results by using only a fraction of the learned parameters. Another important point that we need to emphasize is that for our networks’ training we used very limited set of data as opposed to the competing methods. In particular, for all the tasks that we report except of MRI, we trained our networks using a subset of BSDS500. Please note that USRNet and DWDN have been trained on multiple datasets that consist of images of higher resolutions, whose total number of samples is an order of magnitude more than the one we used.  This is a clear indication that by wisely designing a reconstruction network, relying on classic regularization and optimization techniques, we do not need to employ huge networks and a lot of training data in order to achieve SOTA results.
>
> Another important advantage of our approach is that it allows us to understand better how the network is processing the inputs, to express in an explicit way the desired properties that we expect the solutions to feature, and to be able to interpret in a more concise way what are the factors that contribute to better reconstructions. Finally, we consider the models that we present in this work to be a baseline on top of which other researchers can build better and more powerful reconstruction networks by exploring different and richer parametrizations for the $\boldsymbol{G}$ operator and the WPN network.

---

> > ### Comment · Reviewer_vB8U · 2022-11-15
> > **Response to reviewer's 2nd and 3rd comments**
> >
> > Thank you for your answers. The convergence of the method is clearer now.

---

> ### Author Response · Authors · 2022-11-10
> **Response to reviewer's 4th and 5th comments**
>
>
> > 4. The results are computed for the specific level of noise. It would have been interesting to try a blind version, where the noise level is random during the training.
>
> During training, the inputs of our networks are distorted by Gaussian noise whose standard deviation is chosen randomly but falls within a predefined range which is the same as the one typically considered in the respective literature, i.e. the standard deviation ranges from 0 to either 1% or 3% of the maximum image intensity. The reason of this choice is that it matches the training setup that is used for the networks that we are comparing against.
> It would also be possible to train our networks so that they do not accept as additional input the standard deviation but they instead include a small noise estimation module, whose task would be to estimate the standard deviation directly from the distorted measurements. This would essentially lead to a noise-blind version of our networks. Since there are several noise estimation methods available in the literature (Donoho & Johnstone 1994, Foi 2009, Liu et al. 2013), which are able to predict accurately enough the noise standard deviation from the input data, we didn’t explore such an approach in this work. Additionally, we would like to note that for the real-world results presented in Figs. 2 and 3 we have estimated the noise variance directly from the degraded images using the Wavelet Median Absolute Deviation (WMAD) method originally proposed by Donoho & Johnstone 1994. We have also experimentally observed that using different values of the variance, as long as these values do not differ significantly, does not lead to substantial differences in the quality of the image reconstructions.
>
> > 5. The reproduction of the methods could be difficult (see below). For example, I cannot see the number of layers used for the experiment in the text.
>
> We thank the reviewer for pointing this out. In the updated version of the manuscript, we have added a new section A.6 in the Appendix where we describe in detail the specifics of all the network architectures which were missing. We hope that this addition would make the reproducibility of our framework easier. Moreover, upon acceptance of our paper we plan to publicly release the related code.
>
>
> ## References
> Foi, Alessandro. "Clipped noisy images: Heteroskedastic modeling and practical denoising." Signal Processing 89.12 (2009): 2609-2629.
>
> Liu, Xinhao, Masayuki Tanaka, and Masatoshi Okutomi. "Single-image noise level estimation for blind denoising." IEEE transactions on image processing 22.12 (2013): 5226-5237.
>
> Donoho, David L., and Jain M. Johnstone. "Ideal spatial adaptation by wavelet shrinkage." biometrika 81.3 (1994): 425-455.

---

> > ### Comment · Reviewer_vB8U · 2022-11-15
> > **Response to reviewer's 4th and 5th comments**
> >
> > Thank you for your answers. I agree that there exists several effective noise estimation methods, but blind methods may have the advantage to partially deals with spatially variating noise.

---

> > > ### Author Response · Authors · 2022-11-17
> > > **Response about spatially-varying noise**
> > >
> > > We note, that if the spatially-varying noise is modeled as an additive multivariate zero-mean Gaussian with covariance matrix $\boldsymbol{\Sigma}$, $\boldsymbol{n} \sim \mathcal{N} \left(\boldsymbol{0}, \boldsymbol{\Sigma} \right)$, then the data fidelity term is quadratic and has the form: $\frac{1}{2} || \boldsymbol{y} - \boldsymbol{A}\boldsymbol{x}||^2_{ \boldsymbol{\Sigma^{-1}}}$. Such noise degradation scenario was considered in Zhang et al. 2018. In this case our MM framework is directly applicable, and the main difference will be in the linear system of Eq. 11, whose new form will be:
> > > $$
> > > \boldsymbol x^{k+1} = \left(\boldsymbol A^T\boldsymbol \Sigma^{-1} \boldsymbol A + p\cdot \sum_{i=1}^\ell\boldsymbol G_i^T\boldsymbol W_i^k\boldsymbol G_i + \delta\boldsymbol I\right)^{-1}\left(\boldsymbol A^T\boldsymbol \Sigma^{-1} \boldsymbol y +\delta\boldsymbol x^k\right).
> > > $$
> > >
> > > Based on the above, it would still be possible to design a noise-blind network, consisting of our recurrent strategy combined with a noise covariance estimation module. We identify this as an interesting future research direction.
> > >
> > > ### References
> > > K. Zhang, W. Zuo and L. Zhang, "FFDNet: Toward a Fast and Flexible Solution for CNN-Based Image Denoising," in IEEE Transactions on Image Processing, vol. 27, no. 9, pp. 4608-4622, Sept. 2018, doi: 10.1109/TIP.2018.2839891.

---

> ### Author Response · Authors · 2022-11-17
> **General response to the reviewer**
>
> We thank the reviewer for his feedback. We hope that our responses have adequately addressed all the issues he raised and that he is now convinced that our framework has potentials and is general enough to accommodate a wider range of problems than those considered in this work.

---

### Official Review · Reviewer_gm4f · 2022-10-24

**Confidence:** 5
**Clarity, Quality, Novelty And Reproducibility:** See the above comment.
**Correctness:** 3
**Technical Novelty And Significance:** 3
**Empirical Novelty And Significance:** 3
**Recommendation:** 6

**Strength And Weaknesses:**

# Weakness
The present manuscript has several drawbacks.

## 1. Insufficient Review of Related Works
There are many papers on IRLS with $\ell_p$
regularization/minimization, which are overlooked by the paper (except
Daubechies et al. 2010, Mohan & Fazel 2012). This leads to several
inaccurate claims and developments, e.g.,
1. the paper claims that the proposed algorithm is a generalization
   of IRLS.
2. the paper claims that the bounds in Lemma 1 and Lemma 2 are novel.
3. The paper attempts to learn the weights for IRLS in the experiments.

Please allow me to elaborate on why the two claims are inaccurate.
1. First it should be noted that the proposed algorithm with $\ell_p$
   regularization has been studied in
   - [1] "Improved Iteratively Reweighted Least Squares for Unconstrained
     Smoothed $\ell_q$ Minimization" (SIAM Journal on Numerical
     Analysis, 2013)

   There, Lemma 1 has been proved. In fact, Lemmas 1 and 2 are already
   known by experts who have read Daubechies et al. (2010), and Mohan &
   Fazel (2012). And the reweighting strategy of the paper follows
   directly from Mohan & Fazel (2012).
2. The reweighting strategy derived by Mohan & Fazel (2012) could be
   sub-optimal, but the provably optimal weights have been found
   in a recent paper:
   - [2] "A Scalable Second Order Method for Ill-Conditioned Matrix
     Completion from Few Samples" (ICML 2021)

   Thus it is a bit unclear to me why the paper proposes to learn
   the weights.
3. The paper believes the proposed IRLS variant is a generalization, perhaps
   because Eq. (11) has an additional augmented Lagrangian term, which
   is to make a matrix in (11) invertible. However, this modification
   is problematic for several reasons. First, note that (11) requires
   knowing the variance of the noise, which is typically unavailable.
   Anyway, let us suppose the variance is known. The second issue is
   that, the matrix would be invertible in general. Finally, even if
   it is not invertible, the least-squares problem has infinitely many
   solutions, a natural way to handle this is to take the minimum-norm
   solution, or to project $x^k$ onto the space defined by normal
   equations. With this mysterious regularization term
   $\|x-x^k\|_2^2$, it is theoretically unclear whether the algorithm
   will be convergent (even to stationary points), as the
   least-squares problem is not solved exactly in each iteration.
   Later we will see this creates a lot of issues.

I would recommend reading more on the recent developments of IRLS
since Daubechies et al. (2010), appreciate prior works more, and write
more humbly regarding actual contributions.

## 2. Technical Clarity
The paper has great potential of confusing the readers. Details are
given below:
1. In Section 2.1, the reader would get confused at the very
   beginning, as there is no motivation of having the regularization
   operator $G$ in (3). Is $G$ given? Where does it come from? Why
   should we have it there? Can there be any example of $G$?

2. Section 2.2 should undergo a complete revision, and should be
   turned into a review of IRLS. Note that the
   majorization-minimization interpretation of IRLS has been known in
   the literature, and the two lemmas are not new in my opinion. There
   seems to be no need to emphasize them so much. Instead, this
   section should have an algorithmic listing for IRLS (which is very
   easy to understand but now missing) with an explanation of each
   step. Then write down clearly the objective
   function that IRLS is minimizing; only after this it makes sense to talk
   about stationary points of that objective, and cite some papers
   which prove stationary convergence. (Note that the paper
   keeps talking about stationary points and local minima, but it is not
   clear which optimization problem is referred to.)
3. To my knowledge, Section 3 is the main novelty of the paper. But it
   is poorly written.
   - The first question to which the paper should answer intuitively
     is **why would it be a good idea of learning/unrolling IRLS**? Is it
     because others are doing this kind of thing for different
     optimization algorithms and getting cited? Note that IRLS is a well-defined
     algorithm that converges to some stationary point of the objective
     that it minimizes (which is not the case for the proposed IRLS
     variant, as explained above). It could be the case that learning IRLS
     destroys this guarantee, as the convolution matrix $G$ is
     changing over time (it remains the same only if the gradient is
     zero), and as the weight prediction module does not compute the
     weights in an IRLS way.
   - In Section 3.1, the way that the paper addresses the challenge of
     training is not well justified. It has not been proven that the
     proposed IRLS variant converges to stationary points or some
     fixed points (which are two different concepts but the paper
     confused them at Eq 12), hence (12) is mainly a heuristic choice.
     And it has not been proved that $x^*=f_{\theta}(x^*;y)$.
## 3. Experiments
First of all, my impression is that there are too many manually chosen
factors that impact the performance, and different experiments have
different settings. Hence I will not take this section into serious account.

However, I do want to point out the importance of ablation study,
which seems missing in the paper.

1. The first thing that the paper should convince me is that, **the learned IRLS
   performs significantly much better than IRLS**. Without such an experiment, I would
   not be able to agree that the proposed method is a good idea.
   Please note that the learned IRLS is significantly more complicated than
   IRLS when it comes to tuning parameters. And also note that
   nowadays many of our researchers are using a lot of electricity,
   money, and trees to train deep networks. We have to avoid training
   deep networks for simply this reason, unless absolutely necessary.

2. The second experiment the paper should conduct is to **show that, at
   least empirically, the learned IRLS is actually convergent**, e.g.,
   the optimality condition holds approximately. This is easy to
   detect and plot. Otherwise, no such claim should be made in the paper.


**Summary Of The Paper:**

# Summary of The Paper
The paper considers the problem of image recovery with sparse or
low-rank regularizers. The method considered to solve this problem is
iteratively reweighted least-squares (IRLS), as well as its learned
variant. The methods are implemented and applied to several image
processing problems, e.g., super-resolution, demosaicking, and MRI reconstruction.

**Summary Of The Review:**

The paper has an interesting idea of unrolling IRLS with several applications to image processing.

However, it also has several problems when it comes to related works and technical clarity. In its current form, the paper appears suboptimal for an ICLR publication and a significant revision is needed in my opinion. This is why I intend to give a score of reject.

---

> ### Author Response · Authors · 2022-11-10
> **General response to reviewer**
>
> > The present manuscript has several drawbacks.
> > 1. ### Insufficient Review of Related Works
> > There are many papers on IRLS with  regularization/minimization, which are overlooked by the paper (except Daubechies et al. 2010, Mohan & Fazel 2012). This leads to several inaccurate claims and developments, e.g.,
> >> 1. the paper claims that the proposed algorithm is a generalization of IRLS.
> >> 2. the paper claims that the bounds in Lemma 1 and Lemma 2 are novel.
> >> 3. The paper attempts to learn the weights for IRLS in the experiments.
>
> We agree with the reviewer that there are plenty of papers in the existing literature that use $\ell_p^p/\mathcal{S}_p^p$ regularization in the context of non-supervised reconstruction and many other papers that study the convergence rates of the respective IRLS minimization strategies. However, the primary goal of our work was not to provide a complete overview of the literature on this topic, which would also have been impossible due to the existing space limitations. Instead we decided to refer to the seminal papers of Daubechies et al. 2010 and Mohan & Fazel 2012 which were the first to introduce an IRLS strategy for the minimization of an objective function that involves $\ell_p^p$ and $\mathcal{S}_p^p$ regularization, respectively.
>
> In addition, based on the majority of the reviewer’s comments we are afraid that he has overlooked or did not fully understand some important aspects of our work, which we hope that we will be able to clarify with our responses. Before, responding in detail to all the reviewer’s comments, we would like first to highlight that in our work we go beyond studying the typical $\ell_p^p$/$\mathcal{S}^p_p$
> minimization problems of the form:
> $$\textrm{argmin}_{\boldsymbol{x}}||\boldsymbol{A}\boldsymbol{x}-\boldsymbol{y}||^2 + ||\boldsymbol{x}||^p_p$$ or
>
> $$\textrm{argmin}_{\boldsymbol{x}}||\mathcal{A}\left(\boldsymbol{X}\right)-\boldsymbol{Y}||^2 + ||\boldsymbol{X}||^p_p$$
> which have been the focus of the papers he referred. As we explain in the manuscript and specifically in Eqs. (4), (5) and (6), our regularizers involve **weighted** extensions of the standard $\ell_p^p$ and $\mathcal{S}^p_p$ quasi-norms with $0 < p \le 1$.
>
> The inclusion of the weights $\boldsymbol{w}$ can potentially promote sparse and low-rank solutions in a more efficient manner. The reason for this is that it has been argued in many papers in the past that typically noise affects more the transform-domain coefficients, $\boldsymbol{z}=\boldsymbol{G}\boldsymbol{x}$, whose values are close to zero than those that have high values. So by using the weights $\boldsymbol{w}$ we can take into account this fact and focus on better removing the noise from the transform coefficients that are affected the most. Please note that without the use of such weights, the processing of all the transform-domain coefficients $\boldsymbol{z}$ would be uniform. Another important difference is the presence of the operator $\boldsymbol{G}$ inside the definition of our regularizers, which is missing in the above objective functions, and we will address its presence in one of our next responses to a related comment of the reviewer.
>
> It also seems that the reviewer has confused the weights $\boldsymbol{w}$ to be the same with the IRLS weights $\boldsymbol{W}_i^k$, which  appear in Eqs. (10) and (11) and are updated in each IRLS iteration. Naturally, based on Lemmas 1 & 2 the weights $\boldsymbol{w}$ indeed appear inside the definitions of $\boldsymbol{W}_i^k$ but the two quantities **are not the same**. Moreover, in our LIRLS networks we are not trying to learn $\boldsymbol{W}_i^k$ as the reviewer mentions in several of his comments, but rather we only learn $\boldsymbol{w}$, which **remains fixed** throughout all the IRLS iterations.

---

> > ### Author Response · Authors · 2022-11-10
> > **Response to reviewer's 1st and 2nd comments on Insufficient Review of Related Works**
> >
> > > Please allow me to elaborate on why the two claims are inaccurate.
> > > 1. First it should be noted that the proposed algorithm with  regularization has been studied in
> > > - [1] "Improved Iteratively Reweighted Least Squares for Unconstrained Smoothed  Minimization" (SIAM Journal on Numerical Analysis, 2013)
> > >
> > > There, Lemma 1 has been proved. In fact, Lemmas 1 and 2 are already known by experts who have read Daubechies et al. (2010), and Mohan & Fazel (2012). And the reweighting strategy of the paper follows directly from Mohan & Fazel (2012).
> >
> > As we have explained earlier the reviewer misses the important point which is the presence of the weights $\boldsymbol{w}$ in the definitions of our **weighted extensions** of the $\ell_p^p$ and $\mathcal{S}^p_p$ regularizers defined in Eqs. (4) and (5). The upper-bounds that he is referring to consider the standard $\ell_p^p$ and $\mathcal{S}^p_p$ regularizers and not the weighted extensions that we study in this work. Regarding Lemma 1 that provides the tight upper-bound of the weighted $\ell_p^p$ function, in fact we state clearly in our proof in the appendix A.1 that its derivation is straightforward. For this exact reason we refrained from stating in our manuscript that Lemma 1 is novel, despite the fact that we have been unable to find any existing paper making use of it. Regarding Lemma 2 that provides the tight upper-bounds for the weighted $\mathcal{S}^p_p$ regularizers we insist that, to the best of our knowledge, these bounds are novel and their derivation is not trivial since they rely on Ruhe’s trace inequality which appears to be not widely known in the field of matrix regularization.
> >
> > > 2. The reweighting strategy derived by Mohan & Fazel (2012) could be sub-optimal, but the provably optimal weights have been found in a recent paper:
> > > - [2] "A Scalable Second Order Method for Ill-Conditioned Matrix Completion from Few Samples" (ICML 2021)
> > >
> > > Thus it is a bit unclear to me why the paper proposes to learn the weights.
> >
> > This comment is directly related to our previous statement about the possible confusion of the reviewer regarding which weights we are learning in our network. We need to emphasize again that we are not trying to learn the IRLS weights $\boldsymbol{W}_i^k$ which are updated in every iteration based on the solution of the previous iteration. In fact these weights have the exact forms specified in Lemmas 1 & 2. The only parameters we are learning are the weights $\boldsymbol{w}$ which are part of our regularizers. The weights $\boldsymbol{w}$ are obtained as the output of a Weight Prediction Network (WPN) that accepts as input the features $\boldsymbol{z}_0$, which are computed from an initial solution of the problem $\boldsymbol{x}_0$. Then during the IRLS minimization the weights $\boldsymbol{w}$ **remain fixed**, unlike the IRLS weights $\boldsymbol{W}_i^k$ which are updated based on $\boldsymbol{x}^k$. To make sure that this issue is clarified adequately, in the updated manuscript we have updated Fig. 1 to reflect more accurately the architecture of our proposed LIRLS networks.

---

> > > ### Author Response · Authors · 2022-11-10
> > > **Response to reviewer's 3rd comment on Insufficient Review of Related Works**
> > >
> > > > 3. The paper believes the proposed IRLS variant is a generalization, perhaps because Eq. (11) has an additional augmented Lagrangian term, which is to make a matrix in (11) invertible. However, this modification is problematic for several reasons. First, note that (11) requires knowing the variance of the noise, which is typically unavailable. Anyway, let us suppose the variance is known. The second issue is that, the matrix would be invertible in general. Finally, even if it is not invertible, the least-squares problem has infinitely many solutions, a natural way to handle this is to take the minimum-norm solution, or to project  onto the space defined by normal equations. With this mysterious regularization term
> > > , it is theoretically unclear whether the algorithm will be convergent (even to stationary points), as the least-squares problem is not solved exactly in each iteration. Later we will see this creates a lot of issues.
> > >
> > > The reason that our proposed minimization strategy is a generalization of the IRLS algorithm is not because of the additional term appearing in Eq. (11). The real reason is that our minimization strategy boils down to solving a sequence of re-weighted least squares problems with the weights $\boldsymbol{W}^k$ being updated based on the solution of the previous iteration $\boldsymbol{x}^k$. In the simplest case where we fix the weights $\boldsymbol{w}$, which appear in the definitions of the regularizers in Eqs. (4), (5) and (6), and set all their elements equal to one, then we end up with the IRLS algorithms introduced by Daubechies et al. 2010 and Mohan and Fazel 2012. In this sense our IRLS is a generalization because it can deal with the more generic case of $\boldsymbol{w} \ne \boldsymbol{1}$.
> > >
> > > Regarding the presence of the extra term $\alpha\boldsymbol{I}$ in Eq. (11), as we explain in the manuscript it appears due to the use of an augmented majorizer of the form:
> > > $$\tilde{Q}(\boldsymbol{x}; \boldsymbol{x}^k)  =  Q_{reg}(\boldsymbol{x}; \boldsymbol{x}^k) + \frac{\alpha}{2}||\boldsymbol{x}-\boldsymbol{x}^k||^2$$
> > >
> > > According to the MM framework, this is still a perfectly valid majorizer since it satisfies both of the required properties, that is:
> > > 1. $\tilde{Q}(\boldsymbol{x}^k; \boldsymbol{x}^k) = R(\boldsymbol{x}^k)$ and 2. $\tilde{Q}(\boldsymbol{x}; \boldsymbol{x}^k) \ge R(\boldsymbol{x}^k),\forall  \boldsymbol{x}, \boldsymbol{x}^k$
> > >
> > > Based on the above, the presence of the term $\frac{\alpha}{2}||\boldsymbol{x}-\boldsymbol{x}^k||^2$ is not mysterious at all, but instead it serves the purpose of ensuring that the system matrix in Eq. (11) is always non-singular and a unique solution of the normal equations exists. Otherwise this would be impossible to ensure in general. Besides that, another motivation of using this term is that it ensures that the estimate of the $k+1$-th iteration does not differ significantly from the estimate of the previous iteration. Please also note that had we not pursued this strategy, the training of the networks could have become unstable and the learning process would be compromised.
> > >
> > > Regarding the knowledge of the variance, as we also explained to reviewer vB8U, we can estimate it quite accurately from the input data. We have also experimentally observed on real data that the quality of the reconstructions is not significantly affected when using different values of the variance when these values do not differ significantly to each other.  Moreover, since the addition of the term $\frac{\alpha}{2}||\boldsymbol{x}-\boldsymbol{x}^k||^2$ still leads to a valid majorizer, the convergence of our minimization strategy is guaranteed even in the case that the normal equations are not solved exactly, as long as it holds $\tilde{Q}(\boldsymbol{x}^{k+1}; \boldsymbol{x}^k) \le \tilde{Q}(\boldsymbol{x}^{k}; \boldsymbol{x}^k)$. The proof is rather simple and is provided in Hunter & Lange 2004. We hope that our provided explanations have convinced the reviewer that our choice is not ad-hoc but theoretically justified and it doesn’t create any issues with the overall minimization strategy.
> > >
> > > > I would recommend reading more on the recent developments of IRLS since Daubechies et al. (2010), appreciate prior works more, and write more humbly regarding actual contributions.
> > >
> > > As we explained in the beginning of our response, our primary goal was never to provide an extensive review on IRLS but rather identify the connection of our strategy to the original IRLS algorithm introduced by Daubechies et al. 2010 and Mohan & Fazel 2012.  However, we are more than happy to include the two references that the reviewer has indicated.

---

> ### Author Response · Authors · 2022-11-10
> **Response to reviewer's 1st and 2nd comment on Technical Clarity**
>
> >  ### 2. Technical Clarity
> > The paper has great potential of confusing the readers. Details are given below:
> > 1. In Section 2.1, the reader would get confused at the very beginning, as there is no motivation of having the regularization operator  in (3). Is  given? Where does it come from? Why should we have it there? Can there be any example of ?
>
> As we explain in Section 2.1 of the manuscript the regularization operator $\boldsymbol{G}$ is an essential part of the majority of regularization functionals designed for image reconstruction tasks.  Among the most prominent examples are the Total Variation and Sobolev regularizers, where the operator $\boldsymbol{G}$ corresponds to the gradient operator while the potential function $\phi$ corresponds to the module and the squared module, respectively, of $\nabla\boldsymbol{x}$. Other very popular regularization operators are wavelets which lead to sparse responses (Figueiredo et al. 2007). The reasoning behind employing such regularization operators is that the underlying signal $\boldsymbol x$ that we seek to recover is not necessarily sparse or low-rank. However, the transformed version of $\boldsymbol x$ by some operator $\boldsymbol{G}$, $\boldsymbol{z}= \boldsymbol{G}\boldsymbol{x}$, can possibly be sparse or low-rank. Note that this is essentially one of the underlying ideas of sparse coding, which has been popularized in the past decade (Elad 2010).
> > 2. Section 2.2 should undergo a complete revision, and should be turned into a review of IRLS. Note that the majorization-minimization interpretation of IRLS has been known in the literature, and the two lemmas are not new in my opinion. There seems to be no need to emphasize them so much. Instead, this section should have an algorithmic listing for IRLS (which is very easy to understand but now missing) with an explanation of each step. Then write down clearly the objective function that IRLS is minimizing; only after this it makes sense to talk about stationary points of that objective, and cite some papers which prove stationary convergence. (Note that the paper keeps talking about stationary points and local minima, but it is not clear which optimization problem is referred to.)
>
> As we have clearly stated in the introduction of Section 2 and in the beginning of Section 2.2 of our manuscript our primal goal is to derive an optimization algorithm that can be used to efficiently minimize the objective function $\mathcal{J}(\boldsymbol{x})$ defined in Eq. (2) under any of the two sparsity- and low-rank promoting regularizers defined in Eq. (6). We also explicitly state that because in general the family of these regularizers are non-convex we cannot find the global minimum of the overall objective $\mathcal{J}(\boldsymbol{x})$ but only aim for finding a stationary point.
> Furthermore, while our strategy turns out to be a generalization of IRLS, as we have clearly explained earlier, the existing IRLS algorithms, including the ones the reviewer referred to, are not applicable in our case. Therefore, we strongly believe that in order the reader to better understand how we end up with the proposed iterative scheme, it is important to present our strategy in connection to the MM framework and provide the upper-bounds presented in Lemmas 1 & 2. Otherwise, the use of the specific IRLS weights $\boldsymbol W^k$ would appear as ad-hoc choices and the paper would be disconnected. In addition, please note that without referring to the MM framework and the derived majorizers, we would have still to prove convergence of our iterative method, which now arises naturally from the MM theory.
>
> Regarding the reviewer’s suggestion of including an algorithmic listing of IRLS, unfortunately due to space limitations we are unable to include it the main part of the manuscript but we have included it in the section A.4 of the Appendix. In addition, we have updated Fig. 1 to reflect in an exact way all the involved steps during the forward pass of LIRLS, and now this figure can serve the same purpose as the algorithmic listing.
>
> Finally, in order to emphasize that our minimization strategy can be used to solve a superset of the problems solved by existing IRLS methods, we also provide the following table with the objective functions that have been considered in the literature. In this table we further indicate the type of methods that have been utilized for their minimization. From the provided table it is clear that the objective functions that we minimize cover all the rest of the objective functions as special cases.
>
> ## References
> Elad, Michael. Sparse and redundant representations: from theory to applications in signal and image processing. Vol. 2. No. 1. New York: springer, 2010.

---

> > ### Author Response · Authors · 2022-11-10
> > **Table with comparisons of different objective functions**
> >
> > | Paper| Objective function | Optimization method |
> > | ----------- | --------------------- |----------- |
> > | Xie et al., (2016) | $\mathcal{J}\left(\boldsymbol{X}\right) = \tfrac{1}{2 \boldsymbol \sigma_{\boldsymbol n}^2} \lVert \boldsymbol Y - \boldsymbol X \rVert_F^2 + \sum_{i=1}^r w_i \left(\boldsymbol \sigma_j\left(\boldsymbol X\right)\right)^p$ | Iterative GST |
> > | Liu et al., (2014) | $\mathcal J\left(\boldsymbol X\right) =\tfrac{1}{2\sigma_{\boldsymbol n}^2} \lVert\boldsymbol y- \boldsymbol A\textrm{vec}\left(\boldsymbol X\right)\rVert_2^2 + \sum_{i=1}^r \left(\boldsymbol \sigma_j\left(\boldsymbol X\right)\right)^p$| MM-based |
> > |Gu et al., (2014)| $\mathcal J\left(\boldsymbol X\right) =\tfrac{1}{2\sigma_{\boldsymbol n}^2} \lVert \boldsymbol Y - \boldsymbol X \rVert_F^2 + \sum_{i=1}^r w_i \boldsymbol \sigma_j\left(\boldsymbol X\right)$ | Iterative GST |
> > |Daubechies et al., (2010)| $\mathcal J\left(\boldsymbol x\right) =\tfrac{1}{2\sigma_{\boldsymbol n}^2} \lVert \boldsymbol y-\boldsymbol A\boldsymbol x \rVert_2^2 + \sum\limits_{i=1}^\ell \left(\boldsymbol x_i^2+\gamma\right)^{\frac{p}{2}} $ | IRLS |
> > |Mohan and Fazel (2012)| $\mathcal J\left(\boldsymbol X\right) =\tfrac{1}{2\sigma_{\boldsymbol n}^2} \lVert \boldsymbol y-\mathcal A(\boldsymbol X) \rVert_2^2 + \sum_{i=1}^r\left(\boldsymbol \sigma_j^2\left(\boldsymbol X\right) + \gamma\right)^{\frac{p}{2}}$ | IRLS |
> > |Lai et al., (2013)| $\mathcal J\left(\boldsymbol x\right) =\tfrac{1}{2\sigma_{\boldsymbol n}^2} \lVert \boldsymbol y-\boldsymbol A\boldsymbol x \rVert_2^2 + \sum\limits_{i=1}^\ell \left(\boldsymbol x_i^2+\gamma\right)^{\frac{p}{2}} $ | IRLS |
> > |Lai et al., (2013)| $\mathcal J\left(\boldsymbol X\right) =\tfrac{1}{2\sigma_{\boldsymbol n}^2} \lVert \boldsymbol y-\mathcal A(\boldsymbol X) \rVert_2^2 + \sum_{i=1}^r\left(\boldsymbol \sigma_j^2\left(\boldsymbol X\right) + \gamma\right)^{\frac{p}{2}}$ | IRLS |
> > | Ours | $\mathcal J\left(\boldsymbol x\right) =\tfrac{1}{2\sigma_{\boldsymbol n}^2} \lVert \boldsymbol y-\boldsymbol A\boldsymbol x \rVert_2^2 + \sum\limits_{i=1}^\ell \sum_{j=1}^d\boldsymbol w_{i,j}\left(\left[\boldsymbol z_i\right]_j^2+\gamma\right)^{\frac{p}{2}}, \ \boldsymbol z_i = \boldsymbol G_i \boldsymbol x$ | IRLS-type |
> > | Ours | $\mathcal J\left(\boldsymbol x\right) =\tfrac{1}{2\sigma_{\boldsymbol n}^2} \lVert \boldsymbol y-\boldsymbol A\boldsymbol x \rVert_2^2 + \sum\limits_{i=1}^\ell \sum_{j=1}^r\boldsymbol w_{i,j}\left(\boldsymbol \sigma_j^2\left(\boldsymbol Z_i\right) + \gamma\right)^{\frac{p}{2}}, \ \textrm{vec}\left(\boldsymbol Z_i\right)=\boldsymbol G_i\boldsymbol x$  | IRLS-type |
> >
> > ## References
> > Lai, Ming-Jun, Yangyang Xu, and Wotao Yin. "Improved iteratively reweighted least squares for unconstrained smoothed $\ell_q$ minimization." SIAM Journal on Numerical Analysis 51.2 (2013): 927-957.

---

> ### Author Response · Authors · 2022-11-10
> **Response to reviewer's 3rd comment on Technical Clarity**
>
> > ### 2. Technical Clarity
> > 3. To my knowledge, Section 3 is the main novelty of the paper. But it is poorly written.
> > - The first question to which the paper should answer intuitively is why would it be a good idea of learning/unrolling IRLS? Is it because others are doing this kind of thing for different optimization algorithms and getting cited? Note that IRLS is a well-defined algorithm that converges to some stationary point of the objective that it minimizes (which is not the case for the proposed IRLS variant, as explained above). It could be the case that learning IRLS destroys this guarantee, as the convolution matrix  is changing over time (it remains the same only if the gradient is zero), and as the weight prediction module does not compute the weights in an IRLS way.
>
> Once again we believe that the question about the reason of learning/unrolling IRLS has as its main source the confusion of the reviewer about the exact parameters that we are learning with our networks. As explained earlier we don’t aim to learn the IRLS weights $\boldsymbol{W}_i^k$ but instead the weights $\boldsymbol{w}$ that appear in the studied regularizers. While both these quantities can be interpreted as “weights” we believe that there is a clear distinction between them in the manuscript. Moving past this misconception we fully agree with the reviewer that trying to learn the IRLS weights destroys the convergence guarantees and we stress once more **that this is not what we do in this work**. Furthermore, the learned $\boldsymbol{G}$ operator does not change over the IRLS iterations but remains fixed. In addition, the weights w are computed from the WPN based on the features extracted from the initial solution $\boldsymbol{x}_0$ and also remain fixed during the IRLS steps. Regarding the training process we also need to highlight that both G and the trainable parameters of WPN (which outputs $\boldsymbol{w}$) remain fixed within each batch of training data and are updated after IRLS has converged for this specific batch and the backpropagation for this batch has been performed according to strategy described in Section 3.1
>
> > - In Section 3.1, the way that the paper addresses the challenge of training is not well justified. It has not been proven that the proposed IRLS variant converges to stationary points or some fixed points (which are two different concepts but the paper confused them at Eq 12), hence (12) is mainly a heuristic choice. And it has not been proved that $x^\ast=f_{\theta}(x^\ast, y)$.
>
> Based on all the previous explanations we believe that the reviewer is now convinced that our IRLS strategy is indeed guaranteed to reach a fixed point and thus our proposed training strategy is entirely valid and well-justified. We also thank the reviewer for pointing out the typo (using “stationary” instead of “fixed” point) prior to Eq. (12), which we have corrected.

---

> ### Author Response · Authors · 2022-11-10
> **Response to reviewer's comments on Experiments**
>
> > ### Experiments
> > First of all, my impression is that there are too many manually chosen factors that impact the performance, and different experiments have different settings. Hence I will not take this section into serious account.
>
> Since we are not sure to which manually chosen factors the reviewer is referring to, we would really appreciate if he could be more specific and explicitly state them. Throughout all the reported experiments we have considered identical setups for the training of our different network variants and the only real difference is between the architecture of the WPN that we used for the MRI reconstruction task and the rest of the problems. Apart from that, our networks don’t involve any parameters that need manual fine-tuning. To be precise, all the parameters that otherwise would need to be manually chosen $\boldsymbol \theta = \left(\boldsymbol G, \boldsymbol w, \boldsymbol p \right )$ in our case are learned.
>
> > However, I do want to point out the importance of ablation study, which seems missing in the paper.
> > 1.	The first thing that the paper should convince me is that, **the learned IRLS performs significantly much better than IRLS.** Without such an experiment, I would not be able to agree that the proposed method is a good idea. Please note that the learned IRLS is significantly more complicated than IRLS when it comes to tuning parameters. And also note that nowadays many of our researchers are using a lot of electricity, money, and trees to train deep networks. We have to avoid training deep networks for simply this reason, unless absolutely necessary.
>
> In Tables 1-5 we compare the performance of our learned models with TV, TV-$\ell_1$, VTV, TVN where the $\boldsymbol G$ operator corresponds to the gradient (Jacobian for TVN and VTV) and the potential function $\phi$ to the $\ell_2$ and $\ell_1$, Frobenius and nuclear norms, respectively. Please note that for all TV regularizers we have employed the proper iterative reweighted least squares strategies in order to obtain their solutions. These comparisons match exactly the comparisons between IRLS and LIRLS that the reviewer suggests. From these comparisons it is evident that our learned sparse- and low-rank promoting regularizers consistently perform significantly better across all considered recovery tasks. In fact showing that has been one of the main goals of this work.
>
> > 2. The second experiment the paper should conduct is to **show that, at least empirically, the learned IRLS is actually convergent,** e.g., the optimality condition holds approximately. This is easy to detect and plot. Otherwise, no such claim should be made in the paper.
>
> While we believe that our previous explanations are sufficient to ensure that our IRLS strategy is convergent, in the Appendix we have included the section A.5, where we provide empirical results that further confirm these claims.

---

### Official Review · Reviewer_FciL · 2022-10-25

**Confidence:** 3
**Correctness:** 4
**Technical Novelty And Significance:** 3
**Empirical Novelty And Significance:** 3
**Recommendation:** 8

**Clarity, Quality, Novelty And Reproducibility:**

Novelty: The general idea of majorization minimization and unrolling are not new, but they are combined in a novel way.

Clarity: The paper is clearly written. The proofs look correct, although I did not check every details.

Quality: The overall quality of the paper is satisfying.

**Strength And Weaknesses:**

Strength:

1. The idea of the paper is interesting and very natural. Many existing unrolling networks require a large number of parameters and data, whereas the proposed network only learns a few number of functions as components of IRLS framework. This is a nice and natural way to combine neural networks with a regularized least squares, where the regularizers lie in some restricted spaces that can be efficiently represented by neural networks. Moreover, the majorizors that the authors developed enables incorporating sparse or low rank prior for deep image recovery by combining IRLS with an "unrolling" strategy.

2. The paper is overall well-written. The idea is clearly motivated and easy to follow. I really enjoyed reading this paper.

3. The experiments look impressive. The proposed method achieves top performance on a variety of datasets for different image recovery tasks.

Weaknesses and questions:

1. There seems to be no regularizer that consistently outperforms others, which makes sense. However, it raises another question that how to choose the "searching space" of regularizers for different datasets. Could author(s) add some discussions on whether fixing $p$ or not, and whether promote sparsity or low-rankness? The authors should also explain the reason why sometimes fixing $p=1$ (instead of learning $p$) can be more advantageous, which is somewhat counterintuitive (is that a case of overfitting?).

2. In some image recovery and matrix recovery problems, the ground truth could be both low rank and sparse. Is there anyway to jointly emphasize the sparsity and low-rankness?

3. In the literature review, the authors missed some related works in unrolling networks, and I name a few:

[1] Algorithm Unrolling: Interpretable, Efficient Deep Learning for Signal and Image Processing

[2] Deep Algorithm Unrolling for Blind Image Deblurring



**Summary Of The Paper:**

This work introduces an algorithm for incorporating a learned sparse and low-rank penalty for image recovery tasks. A majorization bound of the penality functions (as quadratic functions) are derived for efficient optimization by Iteratively Reweighted Least
Squares (IRLS). Due to the recurrent nature of the algorithm, the authors propose to learn the prior function $G_i$, the weights and $p$ by
a recurrent network, while the other parts follow the IRLS framework. The proposed algorithm is shown to achieve competitive performance on several image recovery tasks.

**Summary Of The Review:**

Overall, this paper has made a solid contribution to the area of deep image recovery. It uses a novel combination of unrolling networks and the majorization minimization framework to efficiently incorporate sparsity and low-rank prior. In the meantime, I encourage the authors to include more discussions on the choices of regularizers and $p$.

---

> ### Author Response · Authors · 2022-11-10
> **Answer to reviewer's 1st question**
>
> We thank the reviewer for carefully studying our manuscript, for clearly identifying the strengths of our work and for his overall very positive review.
>
> > 1. There seems to be no regularizer that consistently outperforms others, which makes sense. However, it raises another question that how to choose the "searching space" of regularizers for different datasets. Could author(s) add some discussions on whether fixing  or not, and whether promote sparsity or low-rankness? The authors should also explain the reason why sometimes fixing  (instead of learning ) can be more advantageous, which is somewhat counterintuitive (is that a case of overfitting?).
>
> As the reviewer correctly points out, according to our reported results there is not a learned regularizer that consistently outperforms the rest for all the recovery tasks under study. This can be attributed to the fact that different training data were used for different recovery tasks. However, we do observe that for all grayscale reconstruction tasks the $\ell_1^{\boldsymbol{{w}}}$ and $\ell_p^{p, \boldsymbol{{w}}}$ LIRLS models consistently perform better than the $\ell_1$ and $\ell_p^p$ models. This is a strong indication that the presence of the learned weights $\boldsymbol{{w}}$ lead to more powerful sparsity-promoting regularizers and have a positive impact on the reconstruction quality. Moreover, we also observe that the best performance among all the models is accomplished by the $\ell_1^{\boldsymbol{{w}}}$, which can be somewhat counter-intuitive since in theory the choice of $p < 1$ should promote sparse solutions better. A possible explanation for this is that the $\ell_1^{\boldsymbol{{w}}}$ regularizer is a convex one and, thus, is amenable to efficient minimization and LIRLS will converge to the global minimum of the objective function $\mathcal{J}\left(\boldsymbol{{x}}\right)$ in Eq. (2) of the manuscript. On the other hand, the $\ell_p^{p, \boldsymbol{{w}}}$ regularizer is non-convex, which means that the stationary point reached by LIRLS can be sensitive to the initialization and might be far from the global minimum. This is something we have identified in the conclusions section and we plan to investigate in detail in the future.
>
> Regarding the color reconstruction tasks and the learned low-rank promoting regularizers that we have considered, we observe that the best performance on average is achieved by
> the $\mathcal{S}^p_p$ and $\mathcal{S}_p^{p, \boldsymbol{{w}}}$ LIRLS models with $p < 1$. To better interpret these results, we need to have in mind that the only convex regularizer out of this family is the $\mathcal{S}_1$ (nuclear norm). Given that the nuclear norm is not as expressive as the rest of our low-rank promoting regularizers, it is expected that this LIRLS model will be the least performing. Also note that the weighted nuclear norm, $\mathcal{S}_1^{\boldsymbol{{w}}}$, unlike to the $\ell_1^{\boldsymbol{{w}}}$, is non-convex and thus it does not benefit from the convex optimization guarantees. Based on the above, and having in mind the results we report in our manuscript, it turns out that the choice of a $p < 1$ in this case plays a more important role in the reconstruction quality than the learned weights $\boldsymbol{{w}}$.
>
> Regarding the very interesting point that the reviewer makes about whether promoting sparse or low-rank solutions is better, in this work mainly due to the space limitations we considered only the case of promoting sparse solutions for grayscale/single-channel reconstruction tasks and low-rank solutions for the color reconstruction tasks. However, this is a very interesting question that we plan to address in the near future. Indeed, in the literature of single-channel reconstruction tasks, there are regularizers which employ matrix-valued operators such as the Hessian and the Structure Tensor are utilized, see for example Lefkimmiatis et al. 2013 & 2015, and low-rank solutions in the corresponding domains are sought. Likewise, for color reconstruction tasks it is entirely possible to enforce sparsity on the responses of a regularization operator instead of low-rankness.
>
> Finally, regarding the question about fixing $p=1$, the main reason for considering it, is that in certain cases the resulting regularizers are convex functionals, which benefit by certain optimization guarantees. Additionally, we have conducted some experiments but not report them due to space limitations, where we fixed $p$ to values less than 1 and only learned the $\mathbf{G}$ operator. We have observed that this strategy led to worse reconstruction performance than jointly learning $p < 1$ and $\mathbf{G}$.
>
> In the Appendix of the updated manuscript we have added a new section A.3 with this discussion.

---

> > ### Comment · Reviewer_FciL · 2022-11-25
> > **Response to Authors**
> >
> > Thanks for the detailed response, and it addressed all of my concerns. I also read the comments and responses from other reviewers and they do not change my opinion that this is a solid paper. Therefore, I prefer to keep my current score.

---

> ### Author Response · Authors · 2022-11-10
> **Answer to reviewer's 2nd and 3rd questions/comments**
>
> > 2. In some image recovery and matrix recovery problems, the ground truth could be both low rank and sparse. Is there anyway to jointly emphasize the sparsity and low-rankness?
>
> This is a valid point and the answer is positive. If we would like to promote jointly both sparse and low-rank solutions we can definitely do so by forming a composite regularizer of the form:
> $$R(\mathbf{\mathcal{x}}) = R_{sp}(\mathbf{\mathcal{x}}) + \lambda R_{lr}(\mathbf{\mathcal{x}})$$
>  where the $\lambda$ parameter balances the contribution of the sparsity and low-rank promoting regularizers to the solution.
> In such case the machinery for learning the involved parameters of the composite regularizer and minimizing the overall objective function $\mathcal{J}{(\mathbf{\mathcal{x}}})$ is exactly the same as described in Sections 2 and 3. Specifically, we can use the upper bounds derived in Lemmas 1 and 2 to obtain a tight quadratic upper bound for the composite regularizer $R(\mathbf{\mathcal{x}})$, which would be of the form:
> $$Q(\mathbf{\mathcal{x}}; \mathbf{\mathcal{x}}^k) = Q_{sp}(\mathbf{\mathcal{x}}; \mathbf{\mathcal{x}}^k) + \lambda Q_{lr}(\mathbf{\mathcal{x}}; \mathbf{\mathcal{x}}^k)$$
>
> Based on this upper bound we can construct a Learned IRLS network and proceed with the training and inference in the same way as we did for the cases that we examine in the manuscript.
>
>
> > 3. In the literature review, the authors missed some related works in unrolling networks, and I name a few:
> > [1] Algorithm Unrolling: Interpretable, Efficient Deep Learning for Signal and Image Processing
> > [2] Deep Algorithm Unrolling for Blind Image Deblurring
>
> We thank the reviewer for the relative literature, which we have included in the updated manuscript.

---

### Comment · Reviewer_gm4f · 2022-11-11
**Reply to Authors' Rebuttal**

Dear authors,

thanks for your rebuttal and justification. It is lengthy and in great detail, and requires significant efforts in this short period of time. I really appreciate it. I am using a new page, as the original one is too long.

My major concerns come from the fact that I confused the two notations of the weights. I re-read the paper and all the reviews and rebuttals. Much of my concerns have been addressed. But some of them remain, and new concerns arise (naturally from the rebuttal). Let me describe them next.

# 1. Clarity
- The algorithm name (learned IRLS) is quite confusing. What the paper actually do is to learn an objective/prior (parametrized by its weights, exponent p, and transformation G). The way it was done is via a specific training procedure that includes IRLS iterations. However, the phrase (learned IRLS) suggests something like "let us learn how to perform iteratively reweighted least-squares using deep networks".

- The second point that confuses me is about the difference between training and inference. During inference, I think the inference phase is basically to perform IRLS over the learned prior/objective. Of course, in this situation, convergence to stationary points is expected. However, during training, the prior/objective keeps changing. In this situation, why does differentiating through the fixed point equation (12) make sense?

- The authors suggest that $x$ itself might not be sparse or low-rank, but its transformed version Gx could be, so let us learn G. However, if we set G=0, Gx is the sparest and the most low-rank. Therefore I challenge the author with the question of how we could prevent learning a trivial prior G=0.

- I think it would be great if the discussion on the augmented term is added to the paper, otherwise, the current paper looks like this: (i) let us review this MM strategy, (ii) ok, IRLS converges to stationary points, (iii) let us forget the MM strategy and add another term in the objective with an augmented term.

- A minor point that I forgot to mention in my initial review: In the paragraph below (7), the authors wrote "we will reach a local minimum (stationary point)". It should be noted that only convergence to stationary points is guaranteed, not local minimizers.

# 2. Lemmas 1 and 2
- It should be made very clear that "The paper is a generalization of ... since the objective has some weights".
- Even with the extra weight, Lemma 1 is straightforward. One can simply absorb the weight inside the parenthesis, and apply existing results. This will give Lemma 1.

- For the matrix case (Lemma 2), since the objective is not separable, I am not entirely sure whether the above technique would still work. The authors are encouraged to justify why this trivial approach does not work, and having this justification in the paper would make Lemma 2 more well-motivated and convincing.


# 3. Experiments and Ablation Study
- Come back to the point where I insisted on some ablation study on IRLS VS learned IRLS. This is no longer important as the learned IRLS performs IRLS during inference. The difference between TV and learned IRLS in the tables is the objectives.
- In the paper, the authors did not introduce how TV, TV-$\ell_1$ are minimized. I read it twice during reviewing but I did not find that they are solved by IRLS.

In summary, I think the authors' rebuttal has helped me understand the paper. Conditioned on further improvements in clarity, I increased my score.

---

> ### Author Response · Authors · 2022-11-12
> **Response to the reviewer's 1st and 2nd comment on Clarity**
>
> > Comment:
> > Dear authors,
> > thanks for your rebuttal and justification. It is lengthy and in great detail, and requires significant efforts in this short period of time. I really appreciate it. I am using a new page, as the original one is too long.
> >My major concerns come from the fact that I confused the two notations of the weights. I re-read the paper and all the reviews and rebuttals. Much of my concerns have been addressed. But some of them remain, and new concerns arise (naturally from the rebuttal). Let me describe them next.
>
> We thank the reviewer for his timely feedback on our response. We are glad that with our explanations we have managed to clarify in a satisfactory way some important aspects of our work. Below we provide our responses to all reviewer's comments.
>
> > ### 1. Clarity
> > - The algorithm name (learned IRLS) is quite confusing. What the paper actually do is to learn an objective/prior (parametrized by its weights, exponent p, and transformation G). The way it was done is via a specific training procedure that includes IRLS iterations. However, the phrase (learned IRLS) suggests something like "let us learn how to perform iteratively reweighted least-squares using deep networks".
>
> In order to eliminate all sources of possible confusion on this topic, we have renamed Section 3. The new title reflects in a more clear way the strategy we are following with our proposed recurrent network. While we have used the same acronym (LIRLS) to refer to our trained models, we have also updated the text to avoid any misconceptions.
>
> > - The second point that confuses me is about the difference between training and inference. During inference, I think the inference phase is basically to perform IRLS over the learned prior/objective. Of course, in this situation, convergence to stationary points is expected. However, during training, the prior/objective keeps changing. In this situation, why does differentiating through the fixed point equation (12) make sense?
>
> The reviewer is correct about the exact steps that take place during the inference stage. Regarding training, his concern is valid. Indeed,
> if the prior/objective is changing during the IRLS iterations for a single minibatch, then the learning process will be compromised. Let us now describe in detail why this does not happen with our training procedure. Since we employ stochastic training with backpropagation, each training act consists of a minibatch forward + backward pass (note that we refer to it as training “act” and not training “step” in order to avoid confusion with IRLS steps). Let us consider the training act at a timestamp $t$. At this moment we have already trained to some extent the parameters of our regularizer $\boldsymbol \theta^{t-1} = \left( \boldsymbol G^{t-1}, \boldsymbol w^{t-1}, p^{t-1} \right)$. At the beginning of the training act $t$ a new minibatch of data $\boldsymbol d^t = \left( \boldsymbol y^t, \boldsymbol \sigma^t_{\boldsymbol n }, \boldsymbol A^t\right)$ arrives. In order to proceed, we firstly **fix** the regularizer at the current state with its parameters $\boldsymbol \theta = \boldsymbol \theta^{t-1} = \left( \boldsymbol G^{t-1}, \boldsymbol w^{t-1}, p^{t-1} \right)$. Then, we perform a forward pass through our IRLS network for all elements inside the minibatch in parallel. For each one of these elements, the convergence to a fixed point is guaranteed, given that the regularizer parameters $\boldsymbol \theta$ remain fixed to their values $\boldsymbol \theta^{t-1}$ throughout all IRLS iterations. Justifications on convergence are exactly the same, as those correctly pointed out by reviewer for the case of the inference phase, since in fact each forward pass during training is just the inference of IRLS on a minibatch of data with the fixed regularizer parameters $\boldsymbol \theta$. Upon reaching the fixed point $\boldsymbol x^\ast$ for the minibatch, we calculate the loss value between it and the corresponding ground-truth measurements $\boldsymbol x^{\textrm{gt}}$, using the loss function $\mathcal L\left(\boldsymbol \theta, \boldsymbol x^\ast, \boldsymbol x^{\textrm{gt}}\right)$. Now we are able to calculate the gradients of this loss function w.r.t. parameters $\boldsymbol{\theta}$ at the point $\theta^{t-1}$ according to Eqs.(12) and (13): $\left[\nabla_{\boldsymbol \theta} \mathcal L\right]|_{\boldsymbol \theta = \boldsymbol \theta^{t-1}}$. Only then, after both forward and backward passes have been performed, the parameters of regularizers are updated based on the computed gradients, using a stochastic gradient-based optimizer:
>
> $$\boldsymbol \theta^t \leftarrow \textrm{optimizer} \left(\boldsymbol \theta^{t-1}, [\nabla_{\boldsymbol \theta} \mathcal L]|_{\boldsymbol \theta = \boldsymbol \theta^{t-1}} \right).$$
>
> We believe that our explanation above sheds more light to the validity of our proposed learning strategy.

---

> > ### Author Response · Authors · 2022-11-12
> > **Response to the reviewer's 3rd, 4th and 5th comments on Clarity**
> >
> > > ### 1. Clarity
> > > The authors suggest that $\boldsymbol x$ itself might not be sparse or low-rank, but its transformed version $\boldsymbol G \boldsymbol x$ could be, so let us learn $\boldsymbol G$. However, if we set $\boldsymbol G=\boldsymbol 0$, $\boldsymbol G \boldsymbol x$ is the sparest and the most low-rank. Therefore I challenge the author with the question of how we could prevent learning a trivial prior $\boldsymbol G=\boldsymbol 0$.
> >
> > The reviewer raises a valid concern regarding the proper safeguards that can ensure that the trivial prior $\boldsymbol{G}=\boldsymbol{0}$ will not be learned. This indeed would be a real issue if we were training our networks using a loss function that measured the sparsity or rank of the features $\boldsymbol{z^\ast}=\boldsymbol{G}\boldsymbol{x^\ast}$, which are extracted from the network output $\boldsymbol{x^\ast}$. However, in our case the network parameters are trained in such a way so that the negative PSNR loss function between $\boldsymbol{x^\ast}$ and the ground-truth $\boldsymbol{x^{gt}}$ is minimized and we don't use any other loss that explicitly enforces sparsity of the features $\boldsymbol{z^\ast}$. In this case, by learning $\boldsymbol{G}=\boldsymbol{0}$ the network would essentially solve the reconstruction task by relying explicitly on the data fidelity term, which quantifies the consistency of the solution to the observation model, and disregarding entirely any prior knowledge from the training data. While we cannot provide a formal proof for our claims, we believe that it is clear that such reconstructions cannot achieve the best possible PSNR and thus we expect that the network parameters $\boldsymbol{G}$, unless they are initialized to this particular value, will never get equal to $\boldsymbol{G}=\boldsymbol{0}$. It is also worth noting that in all our experiments we have never witnessed such an issue.
> >
> > > I think it would be great if the discussion on the augmented term is added to the paper, otherwise, the current paper looks like this: (i) let us review this MM strategy, (ii) ok, IRLS converges to stationary points, (iii) let us forget the MM strategy and add another term in the objective with an augmented term.
> >
> > Following the reviewer’s suggestion, we have added a Section in the Appendix A.1.1 where we explain in detail why the augmented majorizer we are using is theoretically justified.
> >
> > > A minor point that I forgot to mention in my initial review: In the paragraph below (7), the authors wrote "we will reach a local minimum (stationary point)". It should be noted that only convergence to stationary points is guaranteed, not local minimizers.
> >
> > We thank the reviewer for noticing this typo, which we have corrected.

---

> > ### Comment · Reviewer_gm4f · 2022-11-12
> > **Thanks for clarification on the training process**
> >
> > Thanks for the reply. It is clear to me now. That corresponds to the algorithmic listing in the appendix, and I overlooked the backward pass.
> >
> > I was confused because Figure 1 only indicates the forward pass. Perhaps some clarification on this can be added to the paper. Specifically, the paragraph starting from "To overcome this problem ..." jumps a bit too much and the reader could not follow. The sentence there, **upon convergence** ... "$x^*$ is the network's output" is ambiguous:
> > - **upon convergence** could mean **the convergence of training the networks**. What the authors actually meant here is **upon convergence of IRLS**.
> > - And the definition of $x^*$ should be more specific, as in the algorithmic listing.
> >
> > If space allows, I would recommend moving the algorithmic listing into the main paper, that could kill a lot of potential confusion and misunderstanding caused by the texts.

---

> > > ### Author Response · Authors · 2022-11-12
> > > **Response to the reviewer**
> > >
> > > > Thanks for the reply. It is clear to me now. That corresponds to the algorithmic listing in the appendix, and I overlooked the backward pass.
> > > >
> > > >I was confused because Figure 1 only indicates the forward pass. Perhaps some clarification on this can be added to the paper. Specifically, the paragraph starting from "To overcome this problem ..." jumps a bit too much and the reader could not follow. The sentence there, upon convergence ... " is the network's output" is ambiguous:
> > > >
> > > > - upon convergence could mean the convergence of training the networks. What the authors actually meant here is upon convergence of IRLS.
> > > > - And the definition of $x^\ast$ should be more specific, as in the algorithmic listing.
> > >
> > > We thank the reviewer once again for his suggestions that help us to improve the clarity of the manuscipt. We modified this particular paragraph in the manuscript and we now explicitly refer to the convergence of IRLS and also refer to Eq. 11 about the definition of $\boldsymbol{x}^\ast$.
> > >
> > > >If space allows, I would recommend moving the algorithmic listing into the main paper, that could kill a lot of potential confusion and misunderstanding caused by the texts.
> > >
> > > Unfortunately, the manuscript in its current form is very dense and to be able to include the algorithmic listing in the main text, it would require from us to remove other substantial parts that involve essential information. However, in the revised version of the manuscript in the paragraph following Eq. (14), we refer explicitly to the Algorithm 1 of Section A.4 for the implementation details of the forward and backward passes of our network.

---

> ### Author Response · Authors · 2022-11-12
> **Response to reviewer's comments on Lemmas 1 and 2 and on Experiments**
>
> > ### 2. Lemmas 1 and 2
> > It should be made very clear that "The paper is a generalization of ... since the objective has some weights".
>
> We have emphasized this at the end of Section 2.2.
>
> > Even with the extra weight, Lemma 1 is straightforward. One can simply absorb the weight inside the parenthesis, and apply existing results. This will give Lemma 1.
>
> We agree with the reviewer that the weights $\boldsymbol{W}_{\boldsymbol{y}}$ derived from Lemma 1, could also be inferred by applying a similar strategy  from Daubechies et al. 2010. We have added a small paragraph discussing this in the section A. 1 of the Appendix. where we provide the proof of Lemma 1.
>
> > For the matrix case (Lemma 2), since the objective is not separable, I am not entirely sure whether the above technique would still work. The authors are encouraged to justify why this trivial approach does not work, and having this justification in the paper would make Lemma 2 more well-motivated and convincing.
>
> Similar to the reviewer we also think that there is not a straightforward way to adapt prior existing methods such as the one in Mohan and Fazel 2012, to derive the weights of Lemma 2. Unfortunately, the same strategy as in the $\ell_p^p$ case doesn’t seem to apply . The reason is that the weighted $\mathcal{S}^p_p$  don’t apply directly on the matrix $\boldsymbol{X}$ but instead on its singular values. Specifically, it holds that:
> $$\phi_{lr} (\boldsymbol{X}; \boldsymbol{w}, p) = \sum_{i=1}^r \boldsymbol{w}_i (\boldsymbol{\sigma}^2_i (\boldsymbol{X}) + \gamma)^{p/2}=\textrm{tr}(\boldsymbol{W}(\boldsymbol{X})(\boldsymbol{X}\boldsymbol{X}^T+\gamma \boldsymbol{I})^{p/2})$$.
>
> Unlike the previous case, here the weights $\boldsymbol{w}$ are included in the matrix $\boldsymbol{W} (\boldsymbol{X})$, which directly
> depends on $\boldsymbol{X}$ as: $\boldsymbol{W} (\boldsymbol{X}) = \boldsymbol{U} (\boldsymbol{X}) \textrm{diag} (\boldsymbol{w}) \boldsymbol{U}^T (\boldsymbol{X})$, with $\boldsymbol{U} (\boldsymbol{X})$ being the left singular vectors
> of $\boldsymbol{X}$. Therefore it is unclear to us how the approach by Mohan & Fazel (2012) would apply in this case
> and how the convergence of IRLS can be established
>
> > ### 3. Experiments
> > - In the paper, the authors did not introduce how TV, TV-$\ell_1$ are minimized. I read it twice during reviewing but I did not find that they are solved by IRLS.
>
> We thank the reviewer for pointing this out. We have updated the manuscript to include this information.
>
> > In summary, I think the authors' rebuttal has helped me understand the paper. Conditioned on further improvements in clarity, I increased my score.
>
> We appreciate that the reviewer has increased his original score and found our clarifications to be positive in his further understanding of our work.  We hope that with our latest responses we have managed to clarify the rest of the reviewer's concerns and this will further increase his confidence about the quality of  our work.

---

### Comment · Reviewer_gm4f · 2022-11-12
**Summary of My Review and Score**

Hello everyone,

I have had several rounds of discussions with the authors.

There were some clarity issues in the initial submission, but I think many of them have been fixed. Also, in authors' rebuttal, most concerns of mine have been addressed.

In my opinion, the contribution of the paper consists of the following:
- a quadratic bound for the matrix case (Lemma 2), which allows for a majorization-minimization strategy
- a recurrent network that enables the training with IRLS via differentiating through the fixed point equation. The network has fewer parameters than several other related deep networks.
- multiple applications in image processing

My current score is 6 with a confidence of 5. I would like to give a score of 7 if there were such an option. Overall I think this is a solid and relevant work and deserves publication to ICLR.

---

> ### Comment · Reviewer_vB8U · 2022-11-30
> **Summary of My Review and Score**
>
> Hello everyone,
>
> I completely agree with this comments. The authors answers to my concerns and I can update the score I give to 7. Yes this is solid work that can be publish at ICLR.

---

### Decision · Program_Chairs · 2023-01-20

**Decision:**

Accept: poster

**Justification For Why Not Higher Score:**

As described in the meta review, the paper clearly deserves acceptance. Reviewers converged to a solid accept decision (6,6,8, which reviewers asserted should be read as 7,7,8). The paper's contributions sit within a fairly rich existing literature on IRLS and learned optimization; the paper shows how a well-motivated combination of these ideas, with a novel approach to deep training, can obtained very good empirical performance on a range of tasks.

**Justification For Why Not Lower Score:**

This is a solid paper, with a sufficiently novel formulation, well-motivated training scheme and experimental performance competitive with state-of-the-art networks with significantly more parameters. The paper deserves acceptance.

**Metareview: Summary, Strengths And Weaknesses:**

The paper develops an approach to image recovery based on a combination of (i) iterative reweighed least squares (IRLS) for rank / sparsity minimization, (ii) learned regularization (i.e., an analysis sparsity formulation with a learned analysis operator, and (iii) learned optimization (i.e., training both the analysis operator, weights and other parameters on data). It shows that networks / algorithms developed through this combination of ideas obtain surprisingly strong performance on benchmark datasets in low-level image processing (deblurring, demosaicing, superresolution), while involving far fewer trainable parameters, and requiring less, and less-specific, training data than the state-of-the-art neural networks for these tasks. Deep training is facilitated by using the optimality conditions of the associated optimization problem, to avoid needing to train through large numbers of unfolded iterations.

After several rounds of interaction with the authors, the reviewers converged to an evaluation of the work as a solid paper, deserving of acceptance. In particular, this interaction clarified several key points about what is learned and where the contributions of the paper reside. The key contribution of the paper is less on IRLS per se, than on developing a generalization of IRLS to weighted rank / sparsity formulations for which both weights and regularization parameters (e.g., analysis operators G) can be learned from data, to develop well-motivated deep training strategies, and to demonstrate competitive experimental performance.

**Note From Pc:**

if the above contains the word "oral" or "spotlight" please see: "oral" presentation means -> notable-top-5% and "spotlight" means -> notable-top-25%. As stated in our emails, we are disassociating presentation type from AC recommendations